# Provable Sample Efficiency of Curriculum Post-Training for Transformer Reasoning

**Dake Bu** [1 2 3]  **Wei Huang** [2 4]  **Andi Han** [5]  **Atsushi Nitanda** [3 6]  **Hau-San Wong** [1]  **Qingfu Zhang** [1]  **Taiji Suzuki** [7 2]

## Abstract

Recent curriculum techniques in the post-training stage of LLMs have been empirically observed to outperform non-curriculum approaches in improving reasoning performance, yet a principled understanding of their effectiveness and limitations remains incomplete. To bridge this gap, we develop an abstract theoretical framework and identify sufficient conditions under which curriculum post-training yields exponential improvements in sample complexity. To substantiate this framework, we model the base model's Chain-of-Thought generation as a state-conditioned autoregressive reasoning tree, and formalize curriculum subtasks as either depth-increasing curricula that progressively extend reasoning horizons or hint-decreasing curricula that gradually remove partial hints. Our analysis shows that reinforcement learning finetuning with both curriculum strategies achieves high accuracy with polynomial sample complexity, whereas non-curriculum counterpart encounters an exponential complexity bottleneck. We further establish analogous guarantees for test-time scaling. Empirical simulations support our theoretical findings. Code is available at https://github.com/DakeB U/Curriculum-Post-training.

## 1. Introduction

Transformer-based Large Language Models (LLMs) have recently demonstrated strong emergent capabilities in reasoning and general problem solving. To further improve base models in mathematical domains, various Reinforcement Learning (RL) fine-tuning techniques (Xin et al., 2025; Guo et al., 2025) and test-time scaling methods (Lightman et al., 2024; Snell et al., 2025) with a binary $(0/1)$ outcome verifier have attracted great attention for their effectiveness. Among these approaches, *curriculum-style post-training* (Parashar et al., 2025; Liu et al., 2025a; Lee et al., 2025; Bae et al., 2025)—which encourages Chain-of-Thought (CoT) generation to evolve from easy to hard—has shown strong empirical promise for improving sample efficiency and learning outcomes in reasoning (Meng et al., 2025; Wen et al., 2025b; Zhang et al., 2025a; Zhou et al., 2025; Zhang et al., 2025b).

Despite this progress, the value of curricula in *post-training* remains largely intuitive and lacks rigorous theoretical grounding. Curriculum learning has been extensively studied in *pre-training* or train-from-scratch settings (Bengio et al., 2009; Graves et al., 2017; Narvekar, 2017; Wang et al., 2022; Soviany et al., 2022), with theoretical guarantees established for specialized function classes such as convex regression and binary classification (Weinshall et al., 2018; Weinshall & Amir, 2020), parity (Abbe et al., 2023b; Panigrahi et al., 2025), teacher–student perceptrons with sparse features (Saglietti et al., 2022), and $k$-fold function composition (Wang et al., 2025). However, these results are inherently problem-specific: their notions of "difficulty" and "performance," algorithmic constructions, and proof techniques are tightly tailored to train-from-scratch regimes. Consequently, they do *not* directly explain transformer-based post-training for reasoning, which starts from a strong pretrained base model and targets explicit CoT generalization. This motivates the central question:

> *When, why, how, and in what sense can curriculum strategies in post-training theoretically improve performance compared to direct post-training without curricula?*

Our analysis begins with an intrinsic property of post-training for reasoning. "Performance" (pass@K) is measured by the probability of generating a correct CoT, while "difficulty" is commonly quantified by the *success rate*, i.e., the probability that the base model generates a correct CoT (Tong et al., 2024; Parashar et al., 2025). Equivalently,

---
[1]City University of Hong Kong [2]Center for Advanced Intelligence Project, RIKEN [3]CFAR and IHPC, Agency for Science, Technology and Research (A*STAR) [4]The Institute of Statistical Mathematics [5]University of Sydney [6]Nanyang Technological University [7]The University of Tokyo. Correspondence to: Atsushi Nitanda <atsushi_nitanda@a-star.edu.sg>, Hau-San Wong <cshswong@cityu.edu.hk>.

*Proceedings of the 43rd International Conference on Machine Learning*, Seoul, South Korea. PMLR 306, 2026. Copyright 2026 by the author(s).

task difficulty is governed by the *rarity* of correct reasoning trajectories under the base model. In sparse-reward post-training, this rarity directly drives the amount of sampling required to obtain informative learning signals. Correspondingly, a curriculum can be viewed as learning a sequence of subtasks whose correct CoTs evolve from common to rare.

To formalize this notion, we adopt the *coverage coefficient* (Foster et al., 2025), which measures how rarer a correct CoT is under the base policy than under an optimal target policy, and hence inversely controls the success rate. Standard rejection sampling makes this connection precise by linking rarity to the expected number of trials needed to observe a correct CoT (Block & Polyanskiy, 2023). This yields a sharp necessary-and-sufficient characterization: curriculum post-training improves sample efficiency *if and only if* the cumulative difficulty of all curriculum stages is no larger, up to constants, than the difficulty of the final target relative to the base policy.

This characterization provides a unifying lens for understanding existing curriculum strategies. In practice, *hint-decreasing curricula* (Liu et al., 2025b; Amani et al., 2025) define subtasks by completing the remaining CoT from increasingly shorter correct prefixes, while *depth-increasing curricula* arise naturally in tasks such as parity or countdown by extending short-to-long reasoning steps. More broadly, many graph-structured reasoning problems admit shorter correct CoTs that can be composed into longer ones for harder tasks (Huang et al., 2025c; Ran-Milo et al., 2026). While such empirical curricula do not necessarily guarantee an exact relation between stage-wise and final difficulty, the subtask difficulty typically increases in a controlled and moderate manner, with success rates degrading *gradually rather than catastrophically*.

The characterization also highlights a particularly transparent *sufficient* regime for an *exponential* improvement: the existence of a *$L/p$-th-root curriculum*, where $L$ denotes the number of curriculum stages and $0 < p \ll L$ is a constant. In this regime, the difficulty at each stage is bounded by a $L/p$-th-root level of the target task's difficulty. As a consequence, the total cost of curriculum post-training scales linearly with the number of stages, while direct post-training suffers from the full difficulty of the target, whose correct CoT is very rare under the base policy.

To substantiate this sufficient condition within transformer architectures, we study a state-conditioned autoregressive reasoning process (Def. 2) that subsumes a broad class of reasoning tasks (Kim et al., 2025c; Nichani et al., 2025; 2024; Gandhi et al., 2024). We show that $K$-th-root curricula arise naturally when the pretrained base model assigns comparable probabilities to child nodes in a reasoning tree. Under this condition, we establish exponential-to-polynomial reductions in sample complexity for RL fine-

tuning (Thm. 2) and in reward-oracle or computational complexity for test-time scaling (Thm. 3).

**Contributions.** Our main contributions are:

- We formalize curriculum post-training and establish a general bottleneck corollary (Cor. 1) characterizing sufficient conditions under which stepwise curricula yield exponential improvements over direct post-training.
- We instantiate the theory using an autoregressive reasoning tree (2S-ART) and its transformer realization, covering a broad class of graph-based and compositional reasoning tasks.
- We prove that curriculum strategies in RL fine-tuning and test-time scaling yield exponential-to-polynomial reductions in sample complexity and oracle-query complexity separately, and corroborate the theory with empirical simulations.

We defer additional related work to App. A.

## 2. Theoretical Framework for Curriculum Post-training on reasoning trees

**Preliminaries and Notations**. For each prompt $x$, a policy $\pi$ is a conditional probability measure $\pi(\cdot \mid x)$ over the output space $\mathcal{O}$, where $o \in \mathcal{O}$ denotes a Chain-of-Thought (CoT) trajectory. For two policies $\pi$ and $\pi'$, if $\pi'(\cdot \mid x) \ll \pi(\cdot \mid x)$ for all $x$, we write $\|\frac{\pi'}{\pi}\|_\infty := \sup_{x \in \mathcal{X}} \|\frac{d\pi'(\cdot|x)}{d\pi(\cdot|x)}\|_{L^\infty(\pi(\cdot|x))} = \sup_{x \in \mathcal{X}} \operatorname{ess\,sup}_{o \sim \pi(\cdot|x)} \frac{d\pi'(\cdot|x)}{d\pi(\cdot|x)}(o)$. Landau symbols $\Omega(\cdot), \Theta(\cdot), O(\cdot)$ follow standard conventions, and variants such as $\widetilde{\Theta}(\cdot)$ hide logarithmic factors. For $\mathbf{a} \in \mathbb{R}^m$, $\operatorname{softmax}(\mathbf{a})_i := \exp(a_i)/\sum_j \exp(a_j)$, and with temperature $\beta > 0$ we write $\operatorname{softmax}(\mathbf{a}/\beta)$.

For any $\varepsilon > 0$, let $N_\varepsilon(\pi_2 \mid \pi_1)$ denote the $\varepsilon$-precision sample complexity[1] required to reach the optimal policy $\pi^2$ starting from a policy $\pi_1$. Consider learning the optimal policy $\pi^\star$ from a base policy $\pi_{\mathrm{ref}}$ via a step-wise curriculum consisting of $L$ intermediate subtask policies

$$\pi_0^\star = \pi_{\mathrm{ref}}, \quad \pi_1^\star, \quad \ldots, \quad \pi_K^\star := \pi^\star.$$

By definition, curriculum-style post-training is more sample-efficient than direct post-training if and only if the cumulative sample complexity of the intermediate stages is smaller than that of directly learning $\pi^\star$ from $\pi_{\mathrm{ref}}$, namely,

$$\sum_{\ell \in [L]} N_\varepsilon(\pi_\ell^\star \mid \pi_{\ell-1}^\star) < N_\varepsilon(\pi^\star \mid \pi_{\mathrm{ref}}).$$

---

[1]The concrete notion of $\varepsilon$-precision sample complexity depends on the learning setting. For example, it may correspond to the sample complexity of RL fine-tuning algorithms (Thm. 2), or to the reward-oracle query complexity in test-time scaling (Thm. 3).

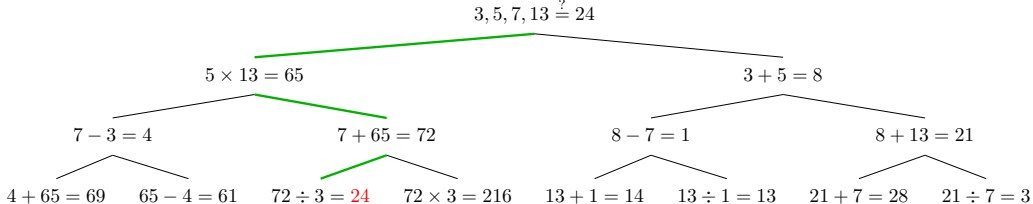

*Figure 1.* An illustration in Liu et al. (2025b) of the Chain-of-Thought for Countdown game, where the goal is to obtain 24 by applying basic arithmetic operations $(+, -, \times, \div)$ $(\Phi_\ell(\cdot, \cdot)$ in our Def. 2) between the current step's number (e.g., 13, 65 or 72) and some unused number (e.g., 5, 7 or 3) in $\{3, 5, 7, 13\}$, targeting 24 as the final outcome. Per (Parashar et al., 2025), the difficulty measure of Countdown is the number of arithmetic operations required to solve an instance.

This condition is exact but abstract. Notably, a transparent sufficient regime under which curriculum post-training yields an *exponential* improvement could be directly derived under the $K$-th root stage-wise sample complexity assumption, per summarized below.

**Corollary 1** (Sufficient Condition for Curriculum Exponential Improvement). *For any $\varepsilon > 0$, consider a base policy $\pi_{\mathrm{ref}} = \pi_0^\star$ and a target optimal policy $\pi^\star \neq \pi_{\mathrm{ref}}$. Suppose there exists a curriculum of $L$ subtask optimal policies $\pi_1^\star, \ldots, \pi_L^\star := \pi^\star$ such that*

$$N_\varepsilon(\pi_\ell^\star \mid \pi_{\ell-1}^\star) = \Theta\big(\sqrt[K]{N_\varepsilon(\pi^\star \mid \pi_{\mathrm{ref}})}\big), \qquad (1)$$

*for all $\ell \in [L]$. Then, the ratio between the sample complexity of direct estimation $N_\varepsilon^{\mathrm{direct}}$ and that of step-wise curriculum estimation $N_\varepsilon^{\mathrm{curriculum}}$ satisfies*

$$\frac{N_\varepsilon^{\mathrm{direct}}}{N_\varepsilon^{\mathrm{curriculum}}} = \frac{N_\varepsilon(\pi^\star \mid \pi_{\mathrm{ref}})}{\sum_{\ell \in [L]} N_\varepsilon(\pi_\ell^\star \mid \pi_{\ell-1}^\star)} = \Theta\Big(\frac{(C^\star)^L}{LC^\star}\Big),$$

*where $C^\star := \sqrt[L]{N_\varepsilon(\pi^\star \mid \pi_{\mathrm{ref}})} > 1$.*

**Exponential–Polynomial Gaps under Relaxed Conditions.** Even if the exact equalities above are relaxed to admit polynomial exponents,

$$N_\varepsilon(\pi_{\ell+1}^\star \mid \pi_\ell^\star) = \Theta\big((C^\star)^{p_\ell}\big), \qquad 1 \leq p_\ell \ll L, \quad (2)$$

the overall curriculum complexity remains

$$N_\varepsilon^{\mathrm{curriculum}} = \Theta\big(L(C^\star)^{p_{\max}}\big) \ll N_\varepsilon^{\mathrm{direct}} = (C^\star)^L, \quad (3)$$

where $p_{\max} := \max_\ell p_\ell$.

**Success Rate and Sample Complexity.** As discussed in the introduction, a natural formalization of success rate is the *coverage coefficient*[2] $\|\frac{\pi^\star}{\pi_{\mathrm{ref}}}\|_\infty$ (Foster et al., 2025), which inversely controls the probability of generating a correct CoT and therefore serves as a natural measure of difficulty. Standard rejection sampling links coverage directly to sampling

cost: sampling *one* correct CoT from $\pi_\ell^\star$ using $\pi_{\mathrm{ref}}$ requires $\Theta\big(\|\frac{\pi_\ell^\star}{\pi_{\mathrm{ref}}}\|_\infty \log(\delta^{-1})\big)$ trials with confidence $1 - \delta$ (Block & Polyanskiy, 2023). More generally, distinguishing policies at a $\|\frac{\pi^\star}{\pi_{\mathrm{ref}}}\|_\infty$-scale margin typically incurs complexity $\widetilde{\Theta}\big(\|\frac{\pi^\star}{\pi_{\mathrm{ref}}}\|_\infty^2\big)$, linking success rate, difficulty, and learning cost in a unified manner.

**Related Theoretical Context.** The assumptions underlying Cor. 1 closely mirror those error accumulation assumptions of approximate policy iteration configurations in Parashar et al. (2025); a detailed comparison is provided in App. D. Moreover, in linearly realizable Markov Decision Processes, by leveraging the spanner-sampling framework of Foster et al. (2025), we establish an analogous exponential improvement in inference-time computational complexity (Thm. 4); see App. D for details. Besides, the analysis of graph traversal tasks in Ran-Milo et al. (2026) can be viewed as a concrete instantiation of Cor. 1: allocating nontrivial probability mass to short-CoT (easy) instances enables rapid convergence to policies which can length-generate, whereas training exclusively on long-CoT (hard) instances leads to exponentially vanishing gradient signals.

**Roadmap.** In the subsequent sections, we show that the conditions of Cor. 1 naturally arise in the transformer tree-reasoning, when the base policy assigns comparable probability mass to the children at each parent state. In this setting, curriculum subtasks correspond to *prefix reasoning traces* (Parashar et al., 2025) or *hint-assisted reasoning* (Liu et al., 2025b; Amani et al., 2025). We further establish a representation theorem showing that transformers can subsume the underlying autoregressive reasoning tree, and prove that the relaxed bound in Eq. (3) holds for both RL fine-tuning dynamics (Thm. 2) and test-time scaling (Thm. 3), in terms of sample complexity and oracle-query complexity.

### 2.1. Transformer Reasons over a States-Conditioned Reasoning Tree

Building on prior work that views post-training as reweighting over a pre-trained reasoning tree (Snell et al., 2025; Yue et al., 2025; AI et al., 2025; Gandhi et al., 2025; Liu

---

[2]i.e., the $\ell_\infty$-norm of the Radon–Nikodym derivative of $\pi^\star$ with respect to $\pi_{\mathrm{ref}}$. If $\mathcal{X} \times \mathcal{Y}$ is discrete, then $\|\frac{\pi^\star}{\pi_{\mathrm{ref}}}\|_\infty = \sup_{x \in \mathcal{X}, y \in \mathcal{Y}} \frac{\pi^\star(y|x)}{\pi_{\mathrm{ref}}(y|x)} > 1$.

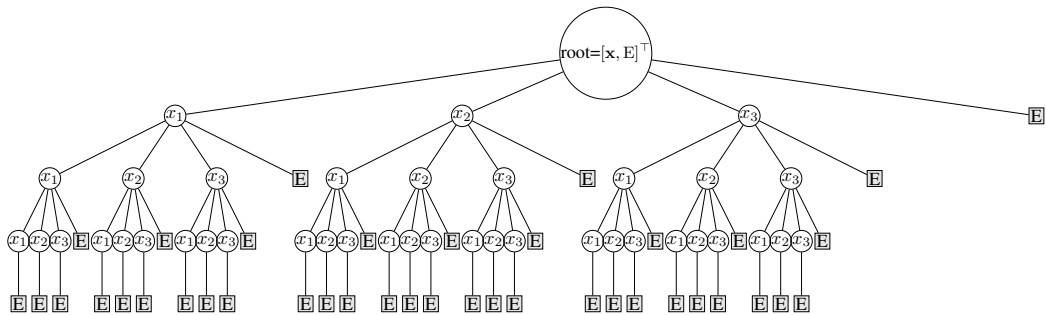

*Figure 2.* Reasoning tree for parity problems with $d = L = 3, V = 2$ and input $x_1, x_2, x_3$, EOS, where $1, 2$ of vocabulary represents $0, 1$. The nodes on the 2nd–4th levels denote hypotheses about which index is the current secret index, corresponding to $x_{i_1}, x_{i_2}$, and $x_{i_3}$, respectively. In our parity CoT class $f \in \mathcal{F}_{\text{2S-ART}}^{\text{parity}}$, each step actually consists of two actions: (i) choose the next secret index $i_t$; (ii) after the choice, apply an XOR over $z_{t-1}$ and $x_{i_t}$ as $z_t = \Phi_l(z_{t-1}, x_{i_t}) := z_{t-1} \oplus x_{i_t}$, as formalized in Eq. (6). For visual clarity, the tree only displays the index-selection branches and omits the explicit XOR updates. E denotes the EOS token. For parity tasks, there are illegal children $\notin \mathcal{I}_\ell$ for each parent that violate the "legal" criteria: non-repeating indices and strictly increasing order $(i_1 < i_2 < i_3 < \cdots)$; thus, for any parity problem, CoTs with duplicate variables or decreasing index order are illegal.

et al., 2025b), we model the reasoning from a tree-causal, low-order autoregressive perspective, per formalized below.

**Definition 1** (2-States Conditioned Autoregressive Reasoning Tree (2S-ART))**.** *A 2S-ART, denoted as $\mathcal{F}_{\text{2S-ART}}$ $(\{\Phi_\ell\}_{\ell \leq L}, \{\mathcal{I}_\ell\}_{\ell \leq L})$ is a class of autoregressive tasks. Given input $(\mathbf{x}, \text{EOS})$ where $\mathbf{x} = (x_1, \ldots, x_d) \in [V]^d$ is followed by a EOS token indexed by $V + 1$, a task $f_{S^k} \in \mathcal{F}_{\text{2S-ART}}$ generates a CoT $(z_1, \ldots, z_k, \text{EOS})$ deterministically according to its $(k+1)$-size index path $S^k = (i_1, \ldots, i_k, d + 1)$ where $k < L$: at step $\ell \in [k]$, $f_{S^k}$ chooses an index $i_\ell \leq d + l$ from a legal set $\mathcal{I}_\ell$ with size $\Theta(d)$, reads the corresponding element $v_{i_\ell}$ from the current sequence, and updates its reasoning state by a two-states map $z_\ell = \Phi_\ell(z_{\ell-1}, v_{i_\ell})$, with $z_0 := \text{EOS}$, $i_{k+1} = d + 1$ and $\Phi_{k+1}(z_k, \text{EOS}) = \text{EOS}$.*

*The **function value** $f_{S^k}(\mathbf{x})$ of task $f_{S^k}$ is defined by the pre-EOS token, i.e.*

$$f_{S^k}(\mathbf{x}) := \Phi_k(z_{k-1}, v_{i_k}).$$

*The associated curriculum subtask family is then characterized by the EOS-ended prefix paths $S^l = (i_1, \ldots, i_\ell, d+1)$ with function values $f_{S^\ell}(\mathbf{x}) := \Phi_k(z_{\ell-1}, v_{i_\ell}), \ell < k$.*

Fig. 2 is an example of parity with $d = L = V + 1 = 3$, and Fig. 1 is an instance of Countdown with $d = 4, V = \mathbb{Z}^+, L = 3$ without illustrating explicit EOS. Here EOS is the end-of-sequence token, and $z_\ell$ is the $\ell$-th reasoning state in the CoT. Depending on the task, $z_\ell$ may be a single token (e.g., in Eq. (6) where $\Phi_\ell(z_{\ell-1}, v_\ell(i_\ell)) = z_{\ell-1} \oplus v_\ell(i_\ell)$) or a short expression (e.g., in the countdown game where $\Phi_\ell(z_{\ell-1}, v_\ell(i_\ell))$ applies $(+, -, \times, \div)$ to $z_{\ell-1}$ and $v_\ell(i_\ell)$; see Fig. 1). The legal index set $\mathcal{I}_\ell$ encodes real-world selection rules, such as not reusing indices or numbers in parity or Countdown. The 2S-ART also subsumes prior abstractions of reasoning behavior:

- Markov-chain reasoning (Kim et al., 2025c) $(\Phi_\ell(z_{\ell-1}, v_\ell(i_\ell)) = \phi(z_{\ell-1})$ for some $\phi$);

- Induction-head for associative recall (Nichani et al., 2025) $(\Phi_\ell(z_{\ell-1}, v_\ell(i_\ell)) = v_\ell(i_\ell))$;

- causal-graph reasoning (Nichani et al., 2024) $(\Phi_\ell(z_{\ell-1}, v_\ell(i_\ell)) = \phi(v_\ell(i_\ell))$ for some $\phi$).

A detailed version of Def. 1 (Def. 3) appears in App. G. To rule out ambiguities where different index paths represent the same task, we adopt the uniqueness assumption below.

**Assumption 1** (Uniqueness)**.** *For $\forall f_{S_\star}, f_{S'_\star} \in \mathcal{F}_{\text{2S-ART}}$ with $|S_\star|, |S'_\star| \leq L + 1$, if $|S_\star| \neq |S'_\star|$, then $f_{S_\star} \neq f_{S'_\star}$.*

A natural way to define a base model with general capability over the task class $\mathcal{F}_{\text{2S-ART}}$ is to suggest that, at each depth $l \in [L]$, the model assigns uniform probability to all children in the legal branch (Liu et al., 2025b) as below.

**Definition 2** (Probabilistic 2S-ART Base Model (`PART`))**.** *Consider a 2S-ART $\mathcal{F}_{\text{2S-ART}}$ $(\{\Phi_\ell\}_{\ell \leq L}, \{\mathcal{I}_\ell\}_{\ell \leq L})$ in Def. 2 with $L \ll d$, a `PART` samples $S_\star$ by uniformly drawing $i_\ell \in \mathcal{I}_\ell(\mathbf{CoT}_{\ell-1}), \forall \ell \in [L]$, i.e. $P(i_\ell \mid z_{<\ell}) = \frac{1}{|\mathcal{I}_\ell(\mathbf{CoT}_{\ell-1})|}$.*

This uniform selection rule in the 2S-ART base model leads directly to an exponential decay in *success-rate* of tasks with depth $\ell$, as summarized below.

**Corollary 2** (Exponential Decay of Success Probability with Depth)**.** *Consider a 2S-ART sampler defined in Def. 1. For a fixed target $f_{S_\star} \in \mathcal{F}_{\text{2S-ART}}$ with associated curriculum subtask family $\mathcal{F}_{S_\star} = \{f_{S_\star^1}, \ldots, f_{S_\star^{k_\star}}\}$, the probability that `PART` samples the legal CoT for $f_{S_\star^\ell}$ is $\Theta(d^{-(\ell+1)})$.*

Consider the optimal policy $\pi_{S^\ell_\star}$ for $f_{S^\ell_\star}$: given any $\mathbf{x} \in [K]^d$, $\pi_{S^\ell_\star}$ samples correct $i_1^\star, \ldots, i_\ell^\star$ in the correct order almost surely, thereby generating correct prefix $z_1^\star, \ldots, z_\ell^\star$

autoregressively, and from step $\ell + 1$ onward copies the behavior of PART. Denote $\pi_{\text{PART}}$ as the policy of PART. It is straightforward to verify that $\|\frac{\pi_{S_\star^{\ell+1}}}{\pi_{S_\star^\ell}}\|_\infty = \Theta(d)$ and $\|\frac{\pi_{S_\star^\ell}}{\pi_{\text{PART}}}\|_\infty = \Theta(d^{\ell+1})$. In the next step, we show that a standard transformer architecture can faithfully replicate the PART reasoning process.

**Transformer ($\text{TF}(\cdot; \mathbf{W})$) as Learning Model.** Let $\mathbf{U} = [\boldsymbol{\mu}_1, \ldots, \boldsymbol{\mu}_V, \boldsymbol{\mu}_{\text{EOS}}] \in \mathbb{R}^{d_{\text{X}} \times (V+1)}$ denote the embedding vocabulary, where the $v$-th column corresponds to the embedding of token $v$, and $\boldsymbol{\mu}_{\text{EOS}} = \boldsymbol{\mu}_{V+1}$ denotes the embedding of EOS. Following Li et al. (2025), we adopt mutually orthogonal positional encodings $\mathbf{P} := [\mathbf{p}_1, \ldots, \mathbf{p}_{d+1}]$ satisfying $\mathbf{p}_i \perp \mathbf{U}$. Token embeddings and positional encodings are concatenated as $\mathbf{E}[x_m] = [\boldsymbol{\mu}^{x_m \top}, \mathbf{p}_m^\top]^\top$, $\mathbf{E}[\text{EOS}] = [\boldsymbol{\mu}_{\text{EOS}}^\top, \mathbf{p}_{d+1}^\top]^\top$, and $\mathbf{E}[z_\ell] = [\boldsymbol{\mu}^{z_\ell \top}, \mathbf{p}_{i_\ell}^\top]^\top$, for all $m \in [d]$, $\ell \in [L+1]$, and $i_\ell \in [d+1]$. Given the embedded inputs $\mathbf{E}[x_1], \ldots, \mathbf{E}[x_d], \mathbf{E}[\text{EOS}] \in \mathbb{R}^{d_{\text{E}}}$. For notational convenience, we define $\mathbf{E}[z_{-d}] := \mathbf{E}[x_1], \ldots, \mathbf{E}[z_{-1}] := \mathbf{E}[x_d]$, and $\mathbf{E}[z_0] := \mathbf{E}[\text{EOS}]$. The transformer performs autoregressive next-token prediction via

$$\text{TF}(\mathbf{E}[z_{-d}], \ldots, \mathbf{E}[z_{\ell-1}]; \mathbf{W}) = \mathbf{E}[z_\ell], \tag{4}$$

where $\mathbf{W}$ denotes the model parameters. Throughout the reasoning process, the original input embeddings remain fixed, while the CoT states $\mathbf{E}[z_\ell]$ are generated autoregressively. Specifically, the forward computation at step $\ell$ is

$$
\begin{aligned}
\hat{\mathbf{p}}_\ell &= \sum_{j=-d}^{\ell-2} \mathbf{V}\Big(\mathbf{E}[z_j]\,\text{softmax}\big(\mathbf{E}[z_j]^\top \mathbf{K}^\top \mathbf{Q} \mathbf{E}[z_{\ell-1}]\big)\Big) \in \mathbb{R}^{d_{\text{X}}}, \\
\mathbf{p}_{i_\ell} &\sim \text{softmax}(\hat{\mathbf{p}}_l^\top \mathbf{P}/\beta), \\
\hat{\boldsymbol{\mu}}^{z_\ell} &= \text{FFN}_\ell(\boldsymbol{\mu}^{x_{i_\ell}}, \hat{\boldsymbol{\mu}}^{z_{\ell-1}}), \quad \mathbf{E}[z_\ell] = [\hat{\boldsymbol{\mu}}^{z_\ell \top}, \mathbf{p}_{i_\ell}^\top]^\top \in \mathbb{R}^{d_{\text{E}}}.
\end{aligned}
\tag{5}
$$

The algorithm terminates when $\mathbf{p}_{d+1}$, the positional embedding associated with EOS, is sampled. Here, $\beta > 0$ denotes a temperature parameter. Causal masking is enforced by setting $\mathbf{E}[z_j]^\top \mathbf{K}^\top \mathbf{Q} \mathbf{E}[z_\ell] \leftarrow -\infty$ whenever $j \geq \ell$ or $\ell \leq 0$. The vector $\boldsymbol{\mu}^{x_{i_\ell}}$ represents the depth-specific token selected at step $\ell$, while $\hat{\boldsymbol{\mu}}^{z_\ell}$ denotes the resulting reasoning state produced by the depth-specific nonlinear feedforward module $\text{FFN}_\ell(\cdot, \cdot)$. Each $\text{FFN}_\ell$ is designed to implement a task-specific reasoning primitive $\Phi_\ell(\cdot, \cdot)$ as defined in Def. 1, such as the XOR operation realized by $h^\top \text{ReLU}(W[\mathbf{E}[z_{\ell-1}]_{:d_{\text{X}}}; \boldsymbol{\mu}^{x_{i_\ell}}])$ in Wen et al. (2025a), the parity function $\phi(\cdot)$ in Kim & Suzuki (2025), or the arithmetic update used in each reasoning step of the countdown game (Fig. 1) (Liu et al., 2025b).

Importantly, our transformer architecture is a sampling variant of the deterministic ones in Kim & Suzuki (2025); Wen et al. (2025a). For theoretical convenience, we adopt the same treatment: attention serves to select crucial token index, while FFN is responsible for basic atomic skills—such as XOR operation, or other logical or arithmetic calculation (i.e., $+, -, \times, \div$)—that are assumed to be already present in the base model and remain stable under fine-tuning.

The following results show that, under this formulation, $\text{TF}(\cdot; \mathbf{W})$ can faithfully replicate the PART behavior.

**Theorem 1** (Base Model as PART ($\text{TF}(\cdot; \mathbf{W}_{\text{base}})$))**.** *Fix any 2S-ART ($\{\Phi_\ell\}_{\ell \leq L}, \{\mathcal{I}_\ell\}_{\ell \leq L}$) from Def. 1 and its probabilistic base model (PART) in Def. 2. Assume that for each depth $\ell$ there exists the map $\text{FFN}_\ell$ that replicates the target operation on embeddings, i.e., $\boldsymbol{\mu}^{z_\ell} = \text{FFN}_\ell(\boldsymbol{\mu}^{v_l(i_\ell)}, \mathbf{E}[z_{\ell-1}]_{:d_{\text{X}}})$ whenever $z_\ell = \Phi_\ell(z_{\ell-1}, v_l(i_\ell))$. Then there exists a parameterization of a transformer in Eq. (5) to copy the PART behavior in Def. 2.*

The assumption that $\text{FFN}_\ell$ exists for each depth $\ell$ is justified by the universal approximation theorem (Cybenko, 1989; Hornik, 1991). This establishes that feedforward networks can approximate any continuous function on compact subsets of $\mathbb{R}^n$ to arbitrary precision. Since continuous functions are dense in $L^p$ spaces for $1 \leq p < \infty$, this extends to approximating most practically relevant functions. Recent advances show that ReLU networks of width $d + 3$ and arbitrary depth can approximate any scalar continuous function of $d$ variables (Lu et al., 2017), while Suzuki (2019); Suzuki & Nitanda (2021) demonstrates optimal approximation rates for functions in Besov spaces, encompassing broader function classes beyond continuity.

0-1 **Outcome Signals**. In real-world mathematical reasoning and program-like tasks, supervision is often available only at the outcome level (correct vs. incorrect). We therefore adopt 0-1 outcome-only supervision that evaluates only the *final pre-EOS prediction*. We use two oracles:

- $\mathbf{R}_{\mathbf{x}}^{f_{S_\star}}(\cdot)$: returns 1 if and only if, immediately before sampling EOS, $\text{TF}(\cdot; \mathbf{W})$ outputs a token whose first $d_{\text{E}}$ coordinates of the embedding equal $\boldsymbol{\mu}^{f_{S_\star}(\mathbf{x})}$; otherwise returns 0.

- $\mathbf{R}_{\mathbf{x}}^{\mathcal{F}_{S_\star}}(\cdot, \ell)$: for the curriculum subtask family $\mathcal{F}_{S_\star} = \{f_{S_\star^1}, \ldots, f_{S_\star^{k^\star}}\}$ defined in Def. 1, returns 1 if and only if the pre-EOS token has an embedding whose first $d_{\text{E}}$ coordinates equal $\boldsymbol{\mu}^{f_{S_\star^\ell}(\mathbf{x})}$ for the queried depth $\ell \in [k^\star]$; otherwise returns 0.

These outcome-only definitions are agnostic to any specific case study; in the parity case discussed in Sec. 2.1.1, outcome reward specializes to whether the result is correct per Eq. (6); in the multi-step language translation task studied in Abedsoltan et al. (2025) where the sampled word is translated to different language at different reasoning steps according to a fixed order, the outcome reward denotes whether the translation is correct for the task family.

**Challenge: Inherent Reward Hacking**. Outcome-only supervision on the final pre-EOS token allows many spurious CoTs to be accepted for a fixed input $\mathbf{x}$, because the oracle **only checks the pre-EOS token rather the correctness of CoT** (i.e., the internal index-selection process of the 2S-ART ($\{\Phi_\ell\}_{\ell \leq L}, \{\mathcal{I}_\ell\}_{\ell \leq L}$)). For instance, in parity, choosing any wrong index at some depth can still yield the correct final bit with probability $1/2$ under $\mathbf{x} \sim \text{Unif}(\{0,1\}^d)$.

### 2.1.1. CONCRETE EXAMPLE: PARITY PROBLEM

**Sparse Parity Problem.** We remark that the Sparse Parity Problem can be seen as a $V = 2$ subcase of Def. 1, with $\mathbf{U} = \{0,1\}$ and the deterministic kernel $\Phi_\ell(z_{\ell-1}, x_{i_\ell})$ defined by the XOR operation $z_{\ell-1} \oplus x_{i_\ell}$ ($l \geq 2$), as well as the convention

$$z_0 = \text{EOS}, \ \Phi_\ell(\text{EOS}, z) = z, \ \Phi_\ell(z, \text{EOS}) = \text{EOS}, \ \forall z \in \mathbf{U}.$$

Formally, given a $d$-dimensional binary input vector $\mathbf{x} = (x_1, \ldots, x_d) \sim \text{Unif}\{0,1\}^d$, the size-$k+1$ index path $S = S^k := (i_1, ..., i_k, d+1)$ is defined by a size-$k$ secret index set $S \backslash \{d+1\}$ plus the index of EOS token (i.e., $d+1$). Here, $S$ define a class of Boolean functions, where each function computes the parity (XOR-sum) of the input components specified by $S = S^k$: $f_{S^k}(\mathbf{x}) = \bigoplus_{i \in S} x_i = x_{i_1} \oplus x_{i_2} \oplus \cdots \oplus x_{i_k}$, where $\oplus$ denotes XOR and $i_1 < i_2 < \cdots < i_k$ without loss of generality. The output satisfies $f_S(\mathbf{x}) = 1$ if $\sum_{i \in S} x_i$ is odd, and $f_S(\mathbf{x}) = 0$ otherwise. The function class containing all such functions is denoted by $\mathcal{P}_{d,k} = \text{Parity}(d,k)$ with cardinality $|\mathcal{P}_{d,k}| = \binom{d}{k}$.

**CoT and Curriculum Subtask Family for Parity**. Per Def. 1, the resulting XOR-based CoT (Wen et al., 2025a; Abedsoltan et al., 2025) is:

$$z_1 = x_{i_1}, \ z_2 = x_1 \oplus x_{i_2}, \ \ldots, \ z_k = z_{k-1} \oplus x_{i_k} = f_S(\mathbf{x}), \quad (6)$$

where $i_1 < i_2 < ... < i_k$. In this case, the curriculum subtask family is the prefix-index family $\mathcal{F}_S = \{f_{S^1}, \ldots, f_{S^{k-1}}\}$ with $S^{k'} = (i_1, \ldots, i_{k'}, d+1)$ and $f_{S^{k'}}(\mathbf{x}) = z_{k'}$ for $k' \in [k]$; early termination by EOS follows Def. 1. In particular, the legal sets $\{\mathcal{I}_\ell\}$ is defined to satisfy $i_1 < i_2 < \cdots < i_k$.

**Transformer** ($\text{TF}(\cdot; \mathbf{W})$) **as Learning Model**. We follow Eq. (4), Eq. (5), set the vocabulary as $\mathbf{U} = [\boldsymbol{\mu}^0, \boldsymbol{\mu}^1, \boldsymbol{\mu}_{\text{EOS}}] = [\boldsymbol{e}_1, \boldsymbol{e}_2, \boldsymbol{e}_3]$ for $0, 1, \text{EOS}$, and the $\text{FFN}_l$ is instantiated for XOR operation:

$$\text{FFN}_l = \mathbf{W}_2 \text{ReLU}[\mathbf{W}_1(\hat{\boldsymbol{\mu}}^{x_{i_l}} + \mathbf{E}[z_{\ell-1}]_{:d_{\text{X}}})] \quad (7)$$

where

$$\mathbf{W}_1 = \begin{bmatrix} \frac{1}{2} & \frac{1}{2} & \frac{1}{2} \\ 0 & 1 & 0 \\ -\frac{1}{2} & \frac{1}{2} & -\frac{1}{2} \end{bmatrix}, \quad \mathbf{W}_2 = \begin{bmatrix} 1 & -1 & 2 \\ 0 & 1 & -2 \\ 0 & 0 & 0 \end{bmatrix}. \quad (8)$$

It can be checked directly that this ensure our $\text{FFN}_l$ replicates the $\Phi_\ell(z_{\ell-1}, v_l(i_\ell)) = z_{\ell-1} \oplus v_l(i_\ell)$.

### 2.1.2. OUTCOME SIGNAL-BASED RL FINETUNINGS BY GRADIENT DESCENT

For mathematics benchmarks, the conventional REINFORCE objective is used to increase the probability that sampled CoTs yield correct answers (Xiong et al., 2025; Setlur et al., 2025). In our setting, the training objective is

$$\mathcal{J}_{\text{REINFORCE}}^{k^\star}(\mathbf{W}^{k^\star}) = \mathbb{E}\left[R^{k^\star}(\text{TF}(\cdot; \mathbf{W}))\right], \quad (9)$$

where $R^{k^\star}(\cdot) \in \{R_{\mathbf{x}}^{f_{S_\star}}(\cdot), R_{\mathbf{x}}^{\mathcal{F}_{S_\star}}(\cdot)\}$, and $\mathbf{W} \in \mathbb{R}^{(d_{\text{E}}-d_{\text{X}})^2}$ is the only matrix we consider trainable in Thm. 1:

$$\mathbf{K}^\top \mathbf{Q} = \begin{pmatrix} \mathbf{0}_{d_{\text{X}} \times d_{\text{X}}} & \mathbf{0}_{d_{\text{X}} \times (d_{\text{E}}-d_{\text{X}})} \\ \mathbf{0}_{(d_{\text{E}}-d_{\text{X}}) \times d_{\text{X}}} & \mathbf{W} \end{pmatrix}, \ \mathbf{V} = \begin{pmatrix} \mathbf{0} & \mathbf{I}_{d_{\text{E}}-d_{\text{X}}} \end{pmatrix}, \quad (10)$$

where the $\mathbf{0}, \mathbf{V}$, as well as the feedforwad map $\text{FFN}_\ell$, is considered fixed during finetuning. This type of reparametrization is common in the transformer optimization literature to enable tractable analysis (Zhang et al., 2024; Huang et al., 2024; Mahankali et al., 2024; Kim & Suzuki, 2025).

In mathematical datasets, the difficulty measure *success-rate* often coincides with reasoning length of CoT (Parashar et al., 2025): Blocksworld uses plan length (Valmeekam et al., 2023), Countdown counts the number of operations (Park et al., 2025), parity utilizes number of XOR operation, and for graph problems, shorter CoTs' atomic skills could be generalized to longer CoTs for harder problems (Cheng et al., 2025; Ran-Milo et al., 2026). Building on these observations, a natural curriculum for $\mathcal{F}_{S_\star}$ is to gradually increase reasoning depth from shallow to deep. Separately, Liu et al. (2025b); Amani et al. (2025) demonstrated the benefit of providing hints (partial CoT prefixes) and progressively shortening them so that the model completes longer suffixes. Therefore, we consider two categories of curriculum finetuning under the oracle $\mathbf{R}_{\mathbf{x}}^{\mathcal{F}_{S_\star}}$ at the $\ell$-th stage:

- **Depth-increasing Curriculum** (Parashar et al., 2025): the algorithm truncate CoTs from $\text{TF}(\cdot, \mathbf{W}^{(t)})$ with EOS to ensure length $\ell + 1$.

- **Hint-decreasing Curriculum** (Liu et al., 2025b): the algorithm provides a CoT prefix of length $k^\star + 1 - \ell$, letting $\text{TF}(\cdot, \mathbf{W}^{(t)})$ generate the remaining steps.

The following theorem proves the exponential improvement of curriculum post-training within our transformer autoregressive reasoning tree setting, which serves as a concrete case study of our Cor. 1 with $C^\star = d^2$.

**Theorem 2** (Curriculum RL Finetuning Avoid Exponential Bottleneck). *Let $\mathbf{U}$ be an orthogonal matrix with $d_{\text{X}} = \Theta(K), d_{\text{E}} = \Theta(d+L)$, $\text{TF}(\cdot; \mathbf{W}_{base})$ per Thm. 1 with trainable $\mathbf{W}$, and a target $f_{S_\star} \in \mathcal{F}_{\text{2S-ART}}$ with $|S_\star| = k^\star+1$. For any $\varepsilon \in (0,1)$, using the RL objective in Eq. (9) and*

*one gradient step per stage (a single step for no-curriculum; $k^\star+1$ online steps for curricula), with probability no less than $1 - \delta$, there exist learning-rate choices $\eta$ such that*

1. *No-curriculum ($R_{\mathbf{x}}^{f_{S\star}}$ as oracle, one-shot update with $\eta = \tilde{\Theta}(d^{k^\star+1})$): the sample complexity to achieve $\mathbb{E}_{\mathbf{x}\sim\mathrm{Unif}([K]^d)}\big[R_{\mathbf{x}}^{f_{S\star}}(\mathrm{TF}(\mathbf{x};\mathbf{W}))\big] \geq 1 - \varepsilon$ is at least $N_\varepsilon \geq \tilde{\Omega}(d^{2k^\star+2})$.*

2. *Depth-increasing Curriculum and Hint-decreasing Curriculum ($\mathbf{R}_{\mathbf{x}}^{\mathcal{F}_{S\star}}$ as oracle, $k^\star+1$ online updates with $\eta = \tilde{\Theta}(d^2)$): the sample complexity to achieve $\mathbb{E}_{\mathbf{x}\sim\mathrm{Unif}([K]^d)}\big[R_{\mathbf{x}}^{f_{S\star}}(\mathrm{TF}(\mathbf{x};\mathbf{W}))\big] \geq 1 - \varepsilon$ is at most $N_\varepsilon \leq \tilde{O}((k^\star + 1)d^2)$.*

*Here $\tilde{\Theta}(\cdot), \tilde{\Omega}(\cdot)$ and $\tilde{O}(\cdot)$ hide polylogarithmic factors in $d, 1/\delta, 1/\varepsilon, \beta$ as well as task-dependent spurious-acceptance parameters.*

**Remark**. Item (1) is not minimax-tight in certain cases of 2S-ART (for example, the minimax rate for the parity class up to EOS is $d^{k^\star}$). The gap arises because our analysis is based on vanilla gradient descent over the empirical estimate of outcome rewards Eq. (9), which disallows cross-sample referencing or structure-exploiting techniques such as Gaussian elimination (Raz, 2018; Abbe & Sandon, 2020). Nevertheless, our result already *suffices*: Item (2) establishes that curriculum post-training can reduce the sample complexity to polynomial order. The specific $d^2$ dependence stems from distinguishing margins of order $\Theta(d^{-1})$ in parallel attempts, which information-theoretically requires inverse-square costs. More sophisticated noisy SGD algorithms that enables cross-sample reference may improve this rate at the cost of a $\mathrm{poly}(d)$ time complexity (Cornacchia & Mossel, 2023; Abbe et al., 2023b).

It is worth noting that one- or few-shot convergence under large learning rates has been widely discussed in prior optimization work, such as Cornacchia & Mossel (2023); Kim & Suzuki (2025). As mentioned in the introduction and Sec. A, there are also theoretical investigations of curriculum benefits in pre-training (train-from-scratch) settings for the parity function class (Cornacchia & Mossel, 2023; Abbe et al., 2023b; Panigrahi et al., 2025). In particular, Cornacchia & Mossel (2023) and Abbe et al. (2023b) construct curricula by mixing data distributions, where "difficulty" is characterized by the density of Hamming weight (fewer 1s than 0s are easier), while Panigrahi et al. (2025) study a teacher–student setup in which "difficulty" is defined by the signal strength of checkpoints provided by the teacher. Their analyses focus on 2-layer ReLU networks or MLPs trained by carefully designed stage-wise or layer-wise gradient descent algorithms. In contrast, our perspective builds on the difficulty measure *success-rate* and its connection

to the inherently probabilistic, tree-like CoT generation behavior in post-training (Yue et al., 2025). By Cor. 2, this leads naturally to the depth-increasing and hint-decreasing curriculum, and we consider GD updates of transformer mirroring the post-training scenarios.

*Proof Sketch.* By Cor. 1, the remaining task is to show Eq. (2) in our case. The sample complexity is determined by sample size needed to reliably distinguish the gradient margin between true signals and spurious signals. From detailed gradient calculations, the expected margin at step $\ell$ is $\Theta(\beta^{-1}d^{-(k^\star+2-\ell)})$, and its absolute value is lower bounded by $\Theta(\beta^{-1}d^{-1})$. To ensure confidence radius no larger than half of the margin, Bernstein-type bounds with a union argument yield $n_\ell = \tilde{\Theta}(d^{2k^\star+2-\ell})$. The first step dominates with $n_1 = \tilde{\Theta}(d^{2k^\star+2})$, giving total sample complexity $N_\varepsilon^{\mathrm{direct}} \geq \tilde{\Omega}(d^{2k^\star+2})$. In contrast, under curriculum strategies, the expected margin at each step is at most $\Theta(\beta d^{-1}/2)$, so the required samples per step are at most $\tilde{\Theta}(d^2) \leq \sqrt[k^\star]{N_\varepsilon^{\mathrm{direct}}}$, and thus satisfies Eq. (2).

### 2.1.3. OUTCOME SIGNAL-BASED TEST-TIME SCALING

In this section, we study *test-time scaling*, where the goal is not merely to output the correct pre-EOS token, but to *identify the correct reasoning path* in the presence of outcome-based reward hacking (e.g., in parity, using a wrong index still yield the correct label with $1/2$ probability).

Following Foster et al. (2025), we measure the sample cost $N_\varepsilon$ of test-time scaling along two axes. Let $T_{\mathrm{data}}$ denote the *reward-oracle query complexity*, i.e., the total number of evaluations of $\mathbf{R}_{\mathbf{x}}^{f_{S\star}}$ or $\mathbf{R}_{\mathbf{x}}^{\mathcal{F}_{S\star}}$, and let $T_{\mathrm{comp}}$ denote the *model-sampling complexity*, i.e., the total number of token emissions from $\mathrm{TF}_{\mathrm{base}}$. The theorem below formalizes an exponential bottleneck for $\mathbf{R}_{\mathbf{x}}^{f_{S\star}}$-only test-time scaling, and shows that curriculum-style oracle ($\mathbf{R}_{\mathbf{x}}^{\mathcal{F}_{S\star}}$,) access provably eliminates this bottleneck.

**Theorem 3** (Curriculum Test-time Scaling Avoid Exponential Bottleneck). *Let $\mathrm{TF}_{base} := \mathrm{TF}(\cdot;\mathbf{W}_{base})$ per definedbe as in Thm. 1, and consider a target $f_{S\star} \in \mathcal{F}_{\text{2S-ART}}$ with $|S_\star| = k^\star + 1$. Then, for identifying the ground-truth path $S_\star$ with confidence $1 - \delta$:*

1. *Using only $\mathbf{R}_{\mathbf{x}}^{f_{S\star}}$, any adaptive procedure (including ones that force visible $x$-tokens by rejection sampling and then roll out) requires $T_{data}, T_{comp} \geq \tilde{\Omega}(d^{2k^\star})$.*

2. *Using curriculum queries adaptively to $\mathbf{R}_{\mathbf{x}}^{\mathcal{F}_{S\star}}$, there exists a procedure with $T_{data} \leq \tilde{O}((k^\star + 1)d^2)$ and $T_{comp} \leq \tilde{O}((k^\star + 1)d^3)$.*

*Here $\tilde{\Omega}(\cdot)$ and $\tilde{O}(\cdot)$ hide polylogarithmic factors in $d$, $1/\delta$ and spurious-acceptance parameters.*

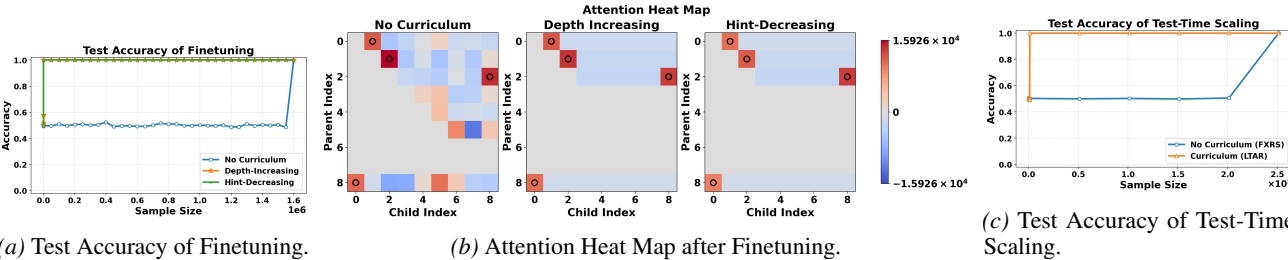

*(a)* Test Accuracy of Finetuning.  *(b)* Attention Heat Map after Finetuning.  *(c)* Test Accuracy of Test-Time Scaling.

*Figure 3.* Dynamics of Finetuning and Test-time Scaling on the parity task ($d = 8$, $k^\star = 3$, $S^\star = [0, 1, 2]$). (a) Test accuracy evolution of finetuning along sample size; (b) attention heatmap after finetuning; (c) test accuracy evolution of test-time scaling along reward oracle query complexity. For finetuning, both curriculum algorithms (Alg. 1 & Alg. 2) converge at sample size $n = 76$, where no-curriculum counterpart (Alg. 1) converges at $n = 1600076$; Attention learning of curriculum algorithms focus on crucial parent-children maps, while no curriculum counterpart is contaminated with spurious corelations; For test-time scaling, curriculum algorithm (LTAR in Alg. 6) converges at $n = 1410$ while its no curriculum counterpart (Alg. 4) converges at $n = 251410$.

Akin to the remark of Thm. 2, without structure-exploiting procedure (Raz, 2018; Abbe et al., 2023a), our results match the information-theoretical bound.

*Proof Sketch.* Again the remaining task is proving Eq. (1) by Cor. 1. We illustrate the idea using the parity class with $1/2$ spurious success chance when picking wrong indices. Consider correct reasoning till step $\ell$, the correct index $j_\star$ and any wrong index $j \neq j_\star$ share probability $\Theta(d^{-1})$ to be selected. If $j_\star$ is chosen and the process continues until EOS, the final CoT obtains reward 1 with probability $\Theta(d^{-(k^\star + 1 - \ell)}) + (1 - \Theta(d^{-(k^\star + 1 - \ell)}))/2$ by Cor. 2, whereas if $j \neq j_\star$, the reward is $1/2$. Hence the probability gap is $\Theta(d^{-(k^\star + 1 - \ell)}/2)$. Distinguishing such gaps information-theoretically requires $\tilde{\Theta}(d^{2k^\star + 2 - 2\ell})$ trials, and summing over $\ell$ yields $\tilde{\Omega}(d^{2k^\star})$ reward queries in total (i.e., $N_\varepsilon = \tilde{\Omega}(d^{2k^\star})$). By contrast, with curriculum (Alg. 6), the algorithm apply $\mathbf{R}_\mathbf{x}^{\mathcal{F}_{S_\star}}(\cdot, \ell)$ on the $\ell + 1$-length truncated CoT. Then, if $j_\star$ is chosen, the chance of reward 1 is $\Theta(d^{-1}) + (1 - \Theta(d^{-1}))/2$, while if $j \neq j_\star$, it is $1/2$. The resulting gap is $\Theta(d^{-1}/2)$, requiring only $\tilde{O}(d^2) \leq \sqrt[k^\star]{N_\varepsilon}$ trials to resolve, which proved Eq. (2) with $C^\star = d^2$.

## 3. Experimental Validations

### 3.1. Synthetic Experiments

We empirically validate the parity-case predictions of our theory. For fine-tuning, all methods are evaluated under the same parity setting: dimension $d = 8$, secret size $k^\star = 3$, and secret indices $S^\star = [0, 1, 2]$. The learning rate $\eta$ is set to $1e^8$ for no-curriculum training and $1e^5$ for both Depth-Increasing and Hint-Decreasing curriculum fine-tuning. All methods use the same transformer architecture with $\beta = 1$ and the same evaluation protocol. Each evaluation is performed on a freshly sampled test set of size $n_{\text{test}} = 8192$, generated on the fly with independent random seeds and no overlap with training data. We use the same parity and transformer setting for test-time scaling experiments, where

complexity is measured by reward-oracle query complexity. Fig. 3 supports our main theoretical predictions. As shown in Fig. 3(a,c), both curriculum strategies substantially outperform direct fine-tuning and require much lower sample/query complexity. To interpret Fig. 3(b), recall that in our transformer realization of 2S-ART, *attention* selects the relevant earlier token/state, while the *FFN* performs the corresponding atomic operation. In the parity case, the FFN implements the XOR update, while attention must recover the correct parent-child dependency along the secret chain.

For the example in Fig. 3(b), the target task has $d = 8$ and secret indices $[0, 1, 2]$, so the intended reasoning chain is $8 \to 0 \to 1 \to 2 \to 8$, where 8 denotes the final EOS token in the input. Starting from EOS, the model should attend to the secret indices step by step and terminate the CoT by outputting $x_1 \oplus x_2 \oplus x_3$ followed by EOS. The attention map therefore directly reveals whether the model has recovered the intended reasoning chain. Both curriculum strategies concentrate sharply on the correct transition chain, with little leakage to spurious edges. By contrast, direct fine-tuning spreads substantial mass over incorrect parent-child links even with significantly more samples, reflecting the intrinsic difficulty of direct training under outcome-only feedback.

Taken together, the accuracy curves in Fig. 3(a,c) and the attention maps in Fig. 3(b) corroborate Thm. 2 and Thm. 3: curriculum-style post-training avoids the exponential bottleneck by recovering the correct dependency structure more efficiently. Additional results, including index-accuracy curves, step-wise heatmap evolution, broader settings ($d = 16, 32$), small-learning-rate experiments, and error bars, are deferred to App. B.

### 3.2. Large-scale Experiments

Beyond the synthetic parity experiments, we evaluate whether curriculum-style post-training remains beneficial on larger-scale reasoning benchmarks. We consider **Countdown**, **MATH**, and **Blocksworld**, all fine-tuned from the

*Table 1.* Pass@K summary on Countdown, MATH, and Blocksworld for the base model (Qwen2.5-1.5B-Instruct), no-curriculum fine-tuning (GRPO), Depth-Increasing Curriculum, and Hint-Decreasing Curriculum. All values are reported in %. Within each dataset block, the best method is highlighted in **bold**. The final block reports macro-averages across the three datasets. Difficulty columns show the mean pass@K over $K \in \{1, \dots, 256\}$. For Countdown and MATH, Levels 1–5 correspond to Trivial/Easy/Medium/Hard/OOD per Parashar et al. (2025); for Blocksworld, they correspond to Levels 0–4 (Parashar et al., 2025).

| Dataset | Method | $K=1$ | $K=2$ | $K=4$ | $K=8$ | $K=16$ | $K=32$ | $K=64$ | $K=128$ | $K=256$ | Avg. | Level 1 | Level 2 | Level 3 | Level 4 | Level 5 |
|---|---|---|---|---|---|---|---|---|---|---|---|---|---|---|---|---|
| Countdown | Base Model | 0.3 | 0.5 | 2.1 | 6.2 | 11.3 | 21.9 | 31.0 | 42.3 | 50.7 | 18.5 | 45.1 | 31.2 | 12.0 | 4.2 | 0.8 |
| | No-Curriculum | 5.2 | 9.2 | 18.8 | 28.5 | 37.8 | 46.3 | 52.6 | 59.5 | 66.3 | 36.0 | 68.8 | 61.1 | 32.6 | 11.9 | 3.8 |
| | Depth-Increasing | **9.4** | **14.2** | **24.4** | **33.8** | 42.8 | 49.9 | 56.2 | 63.6 | 70.6 | 40.5 | **79.0** | **66.5** | 36.4 | 14.3 | 5.2 |
| | Hint-Decreasing | **9.4** | **14.2** | 20.9 | **33.8** | **43.6** | **53.6** | **62.8** | **69.6** | **75.7** | **42.6** | 77.9 | 62.2 | **41.0** | **22.6** | **9.4** |
| Math | Base Model | 0.5 | 0.7 | 2.2 | 4.8 | 10.0 | 18.9 | 27.2 | 36.9 | 47.3 | 16.5 | 42.9 | 29.5 | 21.5 | 12.5 | 6.0 |
| | No-Curriculum | 7.0 | 13.0 | 20.5 | 29.8 | 37.1 | 44.1 | 52.0 | 58.6 | 64.4 | 36.3 | 74.8 | 57.3 | 44.3 | 30.2 | 14.8 |
| | Depth-Increasing | **9.0** | **15.4** | **24.0** | **31.4** | **39.5** | **46.1** | **52.4** | 59.3 | 66.3 | **38.2** | **75.9** | **58.7** | **44.9** | **30.7** | 14.7 |
| | Hint-Decreasing | 4.3 | 7.0 | 13.0 | 24.6 | 31.9 | 42.7 | 51.7 | **60.5** | **68.3** | 33.8 | 68.3 | 50.9 | 40.5 | 28.1 | **15.4** |
| Blocksworld | Base Model | 0.0 | 0.8 | 1.2 | 1.5 | 3.9 | 7.1 | 7.7 | 10.4 | 13.7 | 5.1 | 34.2 | 12.6 | 0.1 | 0.0 | 0.0 |
| | No-Curriculum | 1.9 | 2.1 | 2.1 | 3.5 | 5.6 | 9.8 | **13.3** | **16.6** | 19.7 | 8.3 | **48.2** | 32.1 | 3.4 | 0.4 | 0.1 |
| | Depth-Increasing | 0.8 | 1.0 | 1.9 | 4.6 | 7.9 | 10.2 | 12.2 | **16.6** | **20.1** | 8.4 | 41.8 | 35.1 | 4.2 | **0.5** | **0.2** |
| | Hint-Decreasing | **3.3** | **6.4** | **7.9** | **8.7** | **10.4** | **10.6** | 11.2 | 12.0 | 12.4 | **9.2** | 28.9 | **54.3** | **6.2** | 0.0 | 0.0 |
| *Macro-average over the three datasets* | | | | | | | | | | | | | | | | |
| Avg. | Base Model | 0.3 | 0.7 | 1.8 | 4.2 | 8.4 | 16.0 | 22.0 | 29.9 | 37.2 | 13.4 | 40.7 | 24.5 | 11.2 | 5.6 | 2.3 |
| | No-Curriculum | 4.7 | 8.1 | 13.8 | 20.6 | 26.8 | 33.4 | 39.3 | 44.9 | 50.1 | 26.9 | 63.9 | 50.2 | 26.8 | 14.2 | 6.2 |
| | Depth-Increasing | **6.4** | **10.2** | **16.8** | **23.3** | **30.0** | 35.4 | 40.3 | 46.5 | **52.3** | **29.0** | **65.6** | 53.4 | 28.5 | 15.1 | 6.7 |
| | Hint-Decreasing | 5.7 | 9.2 | 13.9 | 22.4 | 28.6 | **35.6** | **41.9** | **47.4** | 52.1 | 28.5 | 58.3 | **55.8** | **29.2** | **16.9** | **8.3** |

same base model, **Qwen2.5-1.5B-Instruct**. We compare four methods: the base model without post-training, no-curriculum fine-tuning with GRPO, Depth-Increasing Curriculum, and Hint-Decreasing Curriculum. Table 1 reports Pass@K across datasets and difficulty levels. Overall, both curriculum strategies consistently outperform the base model and generally improve over no-curriculum fine-tuning. On **Countdown**, both curricula are clearly stronger than direct GRPO, with Hint-Decreasing achieving the best overall average and the strongest gains on harder levels. On **MATH**, Depth-Increasing achieves the best overall average, while Hint-Decreasing becomes competitive at larger $K$ and on the hardest level. On **Blocksworld**, the gains are smaller and more curriculum-dependent, suggesting that planning-style tasks are more sensitive to how intermediate subtasks are designed.

These results are consistent with the qualitative message of our theory: curriculum is most helpful when it provides a useful decomposition of the target reasoning task into easier intermediate stages. We do not expect uniform dominance of one curriculum across all practical tasks; rather, the empirical results show that curriculum-based fine-tuning is generally more effective than direct fine-tuning, while the better curriculum type depends on task structure. Full algorithm configurations, hyperparameter settings and per-dataset visualizations are deferred to the Appendix C.

## 4. Conclusion, Limitation, and Future Work

We established Cor. 1, which theoretically formalizes the idea that if one can construct a length-$K$ curriculum in which each stage maintains a moderate difficulty relative to the near-zero success rate of the original task, then step-wise learning can be exponentially faster if the stage-wise complexity scales at most at the $K/p$-th-root level of the original task. We substantiate Cor. 1 by modeling reasoning as an autoregressive tree-like process, in which the effective reason-

ing depth grows with task difficulty. Within this model, we show that commonly used curriculum post-training strategies naturally satisfy the above condition, that the resulting reasoning process can be faithfully replicated by a pretrained transformer, and that both post-training fine-tuning and outcome-based test-time scaling enjoy exponential-to-polynomial reductions in sample complexity.

While our analysis is limited on transformer post-training, the benefit of curricula might be indeed model-agnostic, as highlighted in Cor. 1 and Sec. D. Notably, empirical studies also report the effectiveness of curriculum in diffusion models. For MDLMs, Kim et al. (2025b;a); Bai et al. (2026); Cai & Li (2026) suggest that test-time decoding guided by the model's highest-confidence positions can improve reasoning performance. If confidence is viewed as a proxy for difficulty, then decoding guided by confidence are naturally interpretable as an easy-to-hard inference curriculum: the model first resolves easier subproblems, and then compositionally tackles harder ones. Relatedly, Kim et al. (2026); Bu et al. (2026b) provides a training-time analogue of this idea: its progressive unmasking strategy constructs training states according to model's confidence, which can likewise be viewed as a curriculum such that learning is concentrated on progressively more challenging intermediate states rather than being spread uniformly over an exponentially large space of states.

From this viewpoint, both adaptive decoding and progressive training can be seen as high-level supporting instances of Cor. 1: structuring a sequence of easier-to-harder intermediate subproblems may reduce the effective complexity needed to achieve a target accuracy. Extending the theory more explicitly to DLMs is an exciting direction.

## Acknowledgements

DB and HW are supported in part by the Research Grants Council of the Hong Kong Special Administrative Region

(Project No. CityU 11206622). WH is supported by JSPS KAKENHI (24K20848) and JST BOOST (JPMJBY24G6). TS was partially supported by JSPS KAKENHI (24K02905) and JST CREST (PMJCR2015). This research is supported by the National Research Foundation, Singapore and the Ministry of Digital Development and Information under the AI Visiting Professorship Programme (award number AIVP-2024-004). Any opinions, findings and conclusions or recommendations expressed in this material are those of the author(s) and do not reflect the views of National Research Foundation, Singapore and the Ministry of Digital Development and Information.

## Impact Statement

This paper presents work whose goal is to advance the field of Machine Learning. There are many potential societal consequences of our work, none which we feel must be specifically highlighted here.

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

# A. Additional Related Work

**Curriculum strategies in post-training**. A growing body of work has provided empirical evidence for the effectiveness of curriculum strategies during post-training (Liu et al., 2025a; Lee et al., 2025; Bae et al., 2025; Meng et al., 2025; Shi et al., 2025; Wen et al., 2025b; Zhang et al., 2025a; Amani et al., 2025). Especially, **BREAD** (Zhang et al., 2025b) uses branched rollouts with partial expert prefixes: when the model fails, it is given a short expert hint and trained to complete the remaining reasoning steps. As training progresses, the rollouts become increasingly self-generated. This is a clear instance of the **Hint-Decreasing Curriculum** in our main body, though implemented in a more practically refined way. **CORE** (Gao et al., 2026) is different in form but closely aligned with our perspective. It introduces concept-guided intermediate supervision, so the model first learns important concepts and then reasons from them. In our view, this can be interpreted as constructing an easier intermediate policy $\pi^\star_{\text{concept}}$ before learning the final target policy. From the perspective of Cor. 1, it follows the same general principle: a easier intermediate subtask can reduce the overall learning complexity.

**Theoretical benefits of curriculum in post-training**. Zhang et al. (2025a) offered a theoretical perspective, showing that intermediate-difficulty questions yield higher signal-to-noise ratios for gradient estimates. Closely related, Liu et al. (2025b) modeled LLM reasoning as a search tree (see Figure 1) and proved that uniformly sampling a hint depth (i.e., revealing the CoT up to that depth and letting the LLM complete the rest) reduces the exploration complexity of achieving $\geq 50\%$ pass@1 from exponential to polynomial (see Remark 7). In practice, they adopted a cosine scheduler to provide longer hints in early stages and gradually shorten them during fine-tuning. Similarly, Parashar et al. (2025) analyzed curriculum within Approximate Policy Iteration: under exponentially decaying assumptions on policy-approximation error and performance loss, they proved gains in sample complexity and empirically compared uniform, cosine, and Gaussian schedulers. However, none of these studies explicitly models the difficulty measure in post-training — namely, the *pass-rate* and its dependence on the base model — nor do they consider transformer's states-conditioned tree-like autoregressive reasoning. Their results largely focus on optimization endpoints argument (i.e., $\arg\min$ solutions assuming gradient methods have converged), without analyzing the underlying post-training dynamics or the transformer architecture. In contrast, our framework is built directly on the *pass-rate* and the probabilistic tree structure of post-training reasoning, leading to Cor. 1 and its case studies on transformers.

**Theoretical benefits of curriculum learning in pre-training for parity and beyond**. Curriculum Learning (CL) was first introduced by Bengio et al. (2009), and has since been empirically validated across vision, NLP, and reinforcement learning (Graves et al., 2017; Narvekar, 2017; Wang et al., 2022; Soviany et al., 2022). On the theoretical side, most analyses focus on *pre-training* with highly specific problem classes or architectures. A prominent line of work studies the *parity problem*: while parity can be solved efficiently by Gaussian elimination over $\mathbb{F}_2$ (Abbe & Sandon, 2020), gradient-based neural networks struggle on dense data due to precision barriers (Abbe et al., 2021). To circumvent this, Malach et al. (2021) showed that sparse parities become learnable with a one-layer network augmented by a parity module, while Daniely & Malach (2020) analyzed two-layer fully connected nets where sparse inputs with *leaky labels* help learning. Cornacchia & Mossel (2023) proved that sparse parities can be learned by a two-layer net with one-step gradients but require a large learning rate, and Abbe et al. (2024) gave empirical evidence that presenting sparse-to-dense samples as a curriculum improves generalization, though without formal guarantees. In convex settings, Weinshall et al. (2018); Weinshall & Amir (2020) proved that curriculum can accelerate the convergence of SGD. Building on these observations, Abbe et al. (2023b) analyzed a two-layer ReLU network trained with noisy SGD based on mixed data curriculyum where the sparse (lower hamming weights with fewer $-1$) data is first learned, and proved that a *layer-wise curriculum* starting from sparse samples reduces both time and sample complexity. In contrast to data curricula Panigrahi et al. (2025) considered teacher-student setting where the student would benefit from teacher model's checkpoints, serving as the internal signals for students to learn parity support. Statistically, Mannelli et al. (2024) showed the negative effect of overparameterization for curriculum learning, in the case of a XOR-like Gaussian Mixture problem. Beyond parity, Saglietti et al. (2022) studied a teacher–student model where the target depends on a sparse set of features, and curriculum is defined via the variance of irrelevant features (low variance = easy, high variance = hard). More recently, Wang et al. (2025) investigated curriculum learning for transformers in the *k-fold composition task* setting, where they showed that a carefully staged pre-training schedule—both in terms of data distribution and layer-wise progression—enables efficient learning. Taken together, these works demonstrate that curriculum can provably help in pre-training, but only under tailored data curricula (e.g., sparse input with lower Hamming weight where probability of $-1$ is mild, leaky labels, checkpoints), specialized architectures (e.g., augmented by a parity modules), or non-standard optimization schemes (e.g., noisy SGD, layer-wise training). In contrast, our work studies curriculum in the *post-training* regime, where we leverage the base model's coverage power and pre-trained *chain-of-thought reasoning* ability, *without exploiting handcrafted data correlations* or requiring algorithmic modifications.

**Theoretical Benefit of CoT.** A substantial literature has emerged on the theoretical role of chain-of-thought (CoT) in enhancing transformer models. Several works demonstrate that incorporating polynomial-length intermediate steps expands the expressive capacity of transformers beyond constant-depth architectures (Feng et al., 2023; Li et al., 2024; Merrill & Sabharwal, 2024; Wei et al., 2022). Complementing these positive results, a parallel line of research establishes intrinsic barriers, proving lower bounds or rank-based constraints that necessitate non-trivial reasoning depth (Peng et al., 2024; Barceló et al., 2025; Amiri et al., 2025). Further, statistical and compositional analyses reveal how CoT encourages structured generalization and compositional filtering, clarifying its benefit from a distributional perspective (Prystawski et al., 2023; Li et al., 2023). Kim & Suzuki (2025); Wen et al. (2025a) study the benefit of CoT for parity training from scratch using directly aggregated intermediate supervision signals. Relatedly, Liu et al. (2025b) study the benefit of uniformly sampling hint, their analysis counts the number of nodes explored in a search tree, where each reasoning depth is visited with probability $\Theta(1/K)$. This uniform-hint mechanism resembles settings in which learning signals are distributed evenly across depths, mathematically similar to Kim & Suzuki (2025); Wen et al. (2025a). Their techniques are largely different from us: we establish sufficient structural conditions under which step-wise curricula yield exponential improvements in sample complexity. Also, parity is only a special case of our general autoregressive reasoning-tree (2S-ART) framework, which also encompasses Countdown reasoning (Fig. 1), Markov-chain reasoning (Kim et al., 2025c), induction-head–style associative recall (Nichani et al., 2025), causal-graph reasoning (Nichani et al., 2024), and other structured multi-step reasoning operations (Gandhi et al., 2024). Moreover, Kim & Suzuki (2025) analyze learning via a statistical-query framework with an $O(\varepsilon)$-approximate gradient oracle and an average $L_2$ teacher-forced loss, whereas our model explicitly incorporates dictionary-based sampling, uses standard stochastic gradients computed from model-generated trajectories under outcome-only feedback, and evaluates correctness via pre-EOS validation.

**Training Dynamics of Transformers.** Beyond expressiveness, recent studies examine how transformers trained with CoT evolve during optimization. Early work explored emergent in-context learning and convergence under multi-head attention (Hahn & Goyal, 2023; Huang et al., 2024; Kim & Suzuki, 2024; Yang et al., 2024), while subsequent analyses provide provable characterizations of induction heads and sparse token mechanisms (Chen et al., 2024). Of particular relevance are studies on CoT-specific training dynamics, which analyze nonlinear or single-layer transformers solving structured tasks such as parity (Wen et al., 2025a; Kim & Suzuki, 2025; Li et al., 2025). More recent work extends these insights to multi-step optimization and regular language recognition, uncovering implicit biases and algorithmic behaviors in gradient descent (Huang et al., 2025a;b;c). Wen et al. (2025a); Kim & Suzuki (2025) investigated the learning of transformers on the parity task with CoT, building on which Yin et al. (2025) studied the impact of data shift. Yang et al. (2025) further demonstrated that transformers can implement both forward and backward *tree-structured* symbolic reasoning using two attention heads. However, these works abstract away real-world sampling process during inference, relying instead on *atypical deterministic prediction of internal tokens/sentences*, and they primarily analyze pre-training rather than post-training. Bu et al. (2026a); Ran-Milo et al. (2026) both theoretically show the distribution simplicity bias of post-training. In our case, the simple shortcut might be reward hacking (e.g., for parity task, choosing the wrong index still gives $1/2$ population reward), and thus favoring simple rewarded CoT would be a disaster rather than accelerating, as revealed in our theory and attention map's experiments (Fig. 3), where the no-curriculum counterpart struggles to distinguish informative signals.

## B. Additional Experiments Details on Parity

### B.1. Additional Results of Parity Tasks on Default Algorithms (Alg. 1; Alg. 2; Alg. 3)

B.1.1. $d = 8, k^\star = 3$

**Additional Details of $d = 8, k^\star = 3$.** For the experiments of $d = 8, k^\star = 3, S^\star = [0, 1, 2]$ case discussed in Section 3, we further visualize the accuracies of index selection (i.e., whether the model identify the correct secret index step-by-step), as shown in Fig. 5. Corroborating Fig. 3, it is clear that with few sample complexity, the curriculum finetuning strategies correctly identify the correct index.

In Fig. 4, it is also obvious that the no-curriculum finetuning would suffer from spurious success when the sample complexity is limited ($n = 250000$), and they finally grasp the most crucial patterns in a very large sample complexity $n = 11000000$ regime.

The detailed step-by-step attention heatmaps of curriculum finetuning are also presented in Fig. 6 and Fig. 7, where the correct parent ($z_{i_l}$)-children ($x_{i_{l+1}}$) relationship is successfully learned.

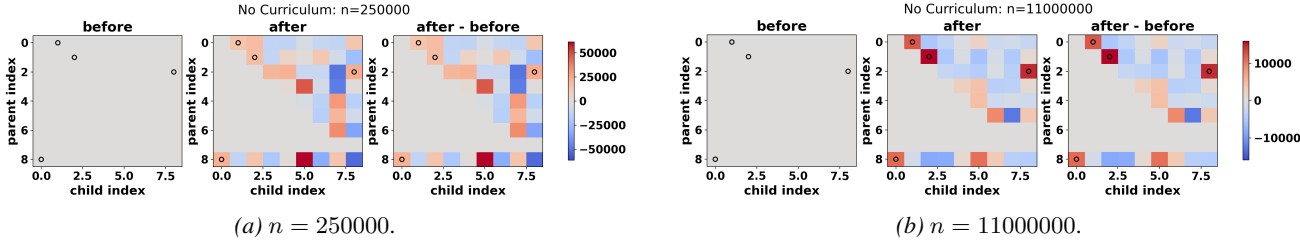

*(a)* $n = 250000$.      *(b)* $n = 11000000$.

*Figure 4.* Attention Heatmaps of No-Curriculum (Direct Learning) with different sample complexity.

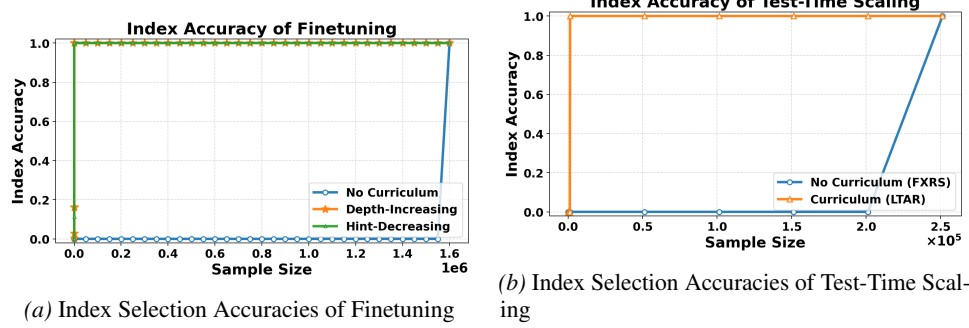

*(a)* Index Selection Accuracies of Finetuning    *(b)* Index Selection Accuracies of Test-Time Scaling

*Figure 5.* Index Selection Accuracies of Finetuning and Test-time Scaling on the parity task ($d = 8$, $k^\star = 3$, $S^\star = [0, 1, 2]$).

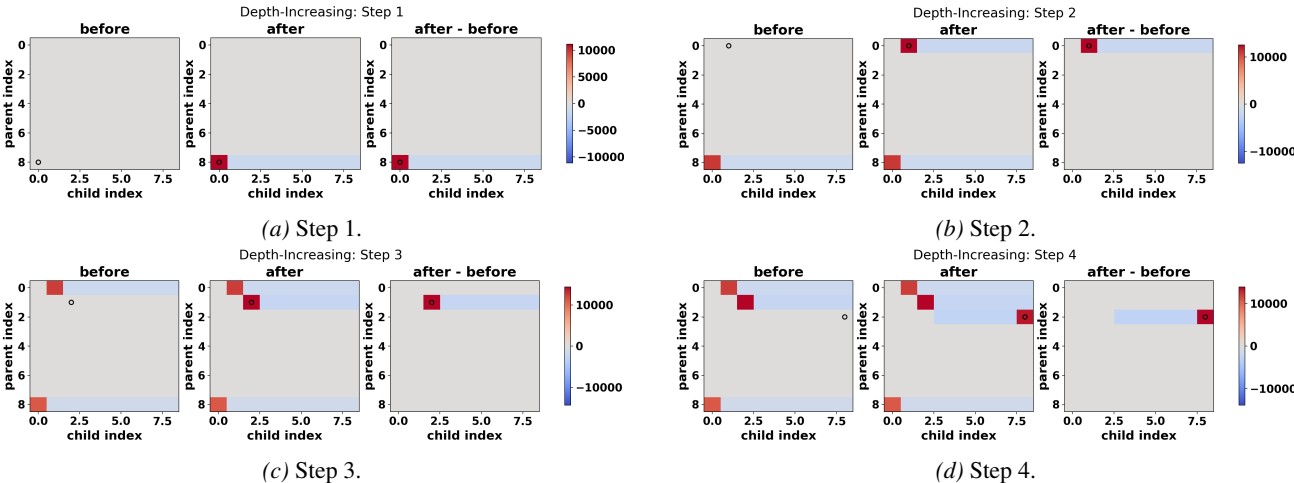

*(a)* Step 1.      *(b)* Step 2.

*(c)* Step 3.      *(d)* Step 4.

*Figure 6.* Step-wise Attention Heatmaps for Depth-Increasing Curriculum ($d = 8$, $k^\star = 3$, $S^\star = [0, 1, 2]$, $n = 100000$).

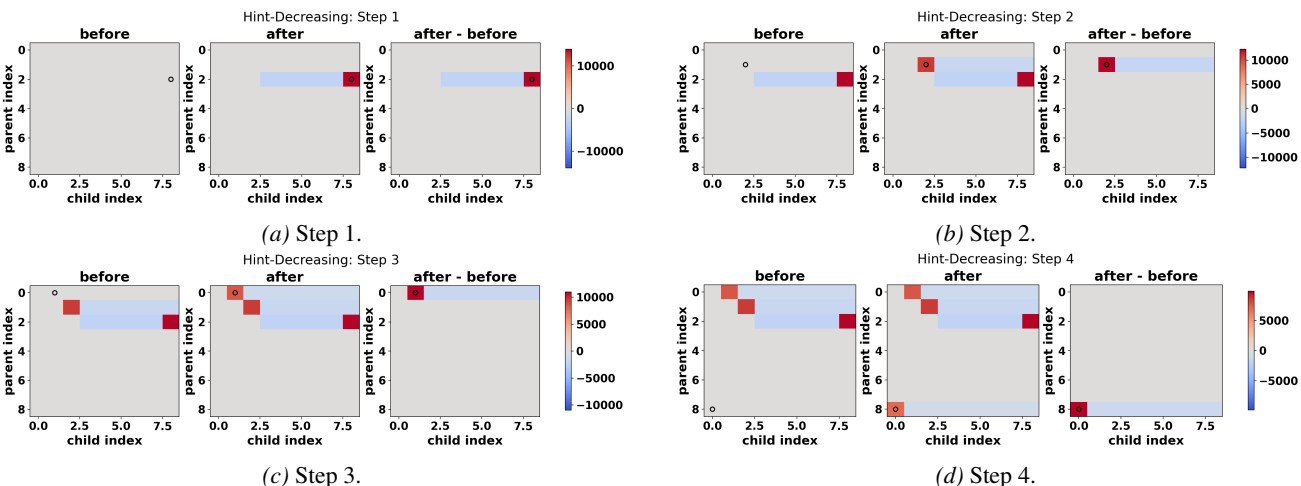

*Figure 7.* Step-wise Attention Heatmaps for Hint-Decreasing Curriculum ($d = 8, k^\star = 3, S^\star = [0, 1, 2], n = 100000$).

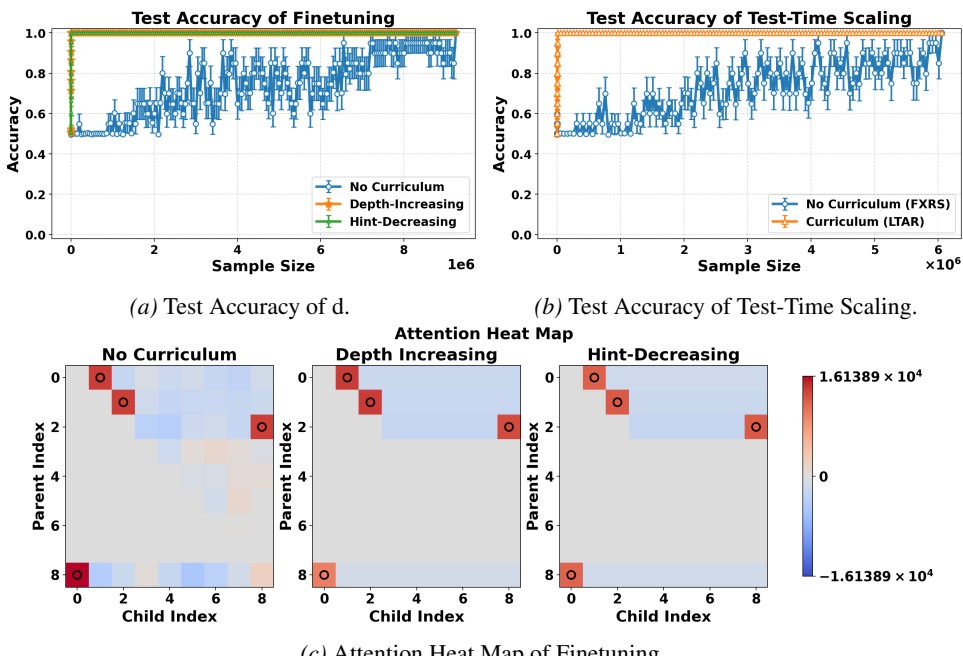

*(a)* Test Accuracy of d.          *(b)* Test Accuracy of Test-Time Scaling.

*(c)* Attention Heat Map of Finetuning.

*Figure 8.* Post-training Dynamics on the parity task with error bar ($d = 8, k^\star = 8, S^\star = [0, 1, 2], \eta_A = 10^8, \eta_B = \eta_C = 10^5$). (a) Test accuracy evolution of finetuning along sample complexity; (b) Test accuracy evolution of Test Time-Scaling along sample complexity; (c) averaged attention heatmap after finetuning.

**Multi-Trial Validation with Error Bars.** To further validate the effectiveness of curriculum strategies and quantify the variance of our results, we conduct extensive multi-trial experiments for both finetuning and test-time scaling procedures. For finetuning experiments, we run $n_{\text{trials}} = 10$ independent trials with base seed 0, where each trial uses a distinct random seed seed $= 0 + t$ for $t = 0, \ldots, 9$. For each trial, we evaluate on a held-out test set of $n_{\text{test}} = 8192$ fresh samples generated with independent randomness, ensuring no overlap between training and test data. We report the mean accuracy across trials along with error bars representing a $1.0\times$ confidence interval (CI $=$ mean $\pm 1.0 \times$ SEM), where SEM $= \sigma/\sqrt{n_{\text{trials}}}$ is the standard error of the mean and $\sigma$ is the sample standard deviation. For test-time scaling experiments, we follow the same multi-trial protocol with $n_{\text{trials}} = 10$, evaluating both Curriculum (LTAR) and No Curriculum (TTS) procedures at each sample complexity $n$. The test evaluation uses the same $n_{\text{test}} = 8192$ fresh samples per trial. Error bars are computed identically as mean $\pm 1.0 \times$ SEM.

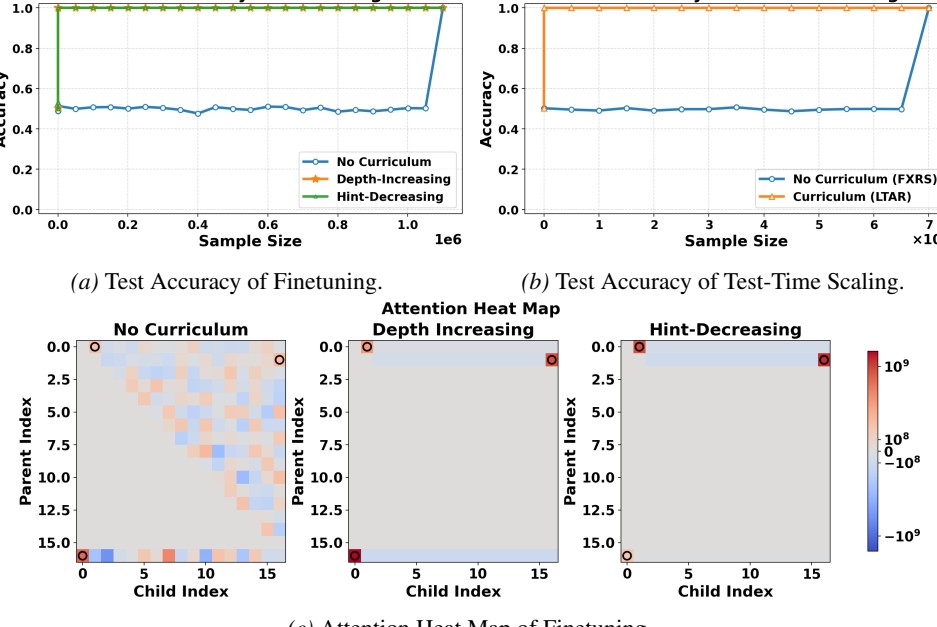

*(a)* Test Accuracy of Finetuning.      *(b)* Test Accuracy of Test-Time Scaling.

*(c)* Attention Heat Map of Finetuning.

*Figure 9.* Post-training Dynamics on the parity task ($d = 16$, $k^\star = 2$, $S^\star = [0, 1]$, $\eta_A = 2 \times 10^{12}$, $\eta_B = \eta_C = 10^{10}$). (a) Test accuracy evolution of finetuning along sample complexity; (b) Test accuracy evolution of Test Time-Scaling along sample complexity; (c) attention heatmap after finetuning. For finetuning, Depth-Increasing and Hint-Decreasing Curriculum Algorithms both converges at $n = 120$, where No Curriculum counterpart converges at $n = 1100120$; For test-time scaling, curriculum algorithm (LTAR in Alg. 6) converges at $n = 5$ while its no curriculum counterpart converges at $n = 700005$.

As shown in Fig. 8, the curriculum-based methods (Depth-Increasing and Hint-Decreasing for finetuning; LTAR for test-time scaling) consistently achieve near-perfect accuracy ($> 99.9\%$) with substantially fewer samples compared to their no-curriculum counterparts. The narrow error bars indicate high statistical significance: for finetuning, No Curriculum requires approximately $n \approx 10^6$ samples to converge, while both curriculum methods converge by $n \approx 200$. For test-time scaling, No Curriculum (TTS) also requires $n \approx 10^6$ samples, whereas Curriculum (LTAR) converges by $n \approx 8 \times 10^3$—an improvement of over $500\times$ in sample complexity. These results strongly corroborate our theoretical findings that curriculum strategies avoid the exponential bottleneck inherent in non-curriculum approaches. **Parameters.** Finetuning experiments use learning rates $\eta_A = 10^8$ for Algorithm A and $\eta_{BC} = 10^5$ for Algorithms B/C, with micro-batch size $6.7 \times 10^6$ (automatically adjusted based on available GPU memory). Test-time scaling experiments use oracle batch size $7.3 \times 10^6$, confidence parameter $\delta = 0.1$, and spurious-success upper bound $\rho_{\text{spur}}^{\text{sup}} = 0.5$.

B.1.2. $d = 16, k^\star = 2$

For parameter settings $d = 16$, $k^\star = 2$, $S^\star = [0, 1]$, $\eta_A = 2 \times 10^{12}$, $\eta_B = \eta_C = 10^{10}$, we also visualize the sample complexity v.s. accuracies plots of test-time scaling as well as finetuning dynamics, along with the attention heatmaps, per shown in Fig. 9. Similar to Fig. 3, the curriculum strategies help grasp the subtask's pattern more effectively, resulting satisfactory test performance with less sample complexity.

B.1.3. $d = 32, k^\star = 2$

Similarly, we have the results for $d = 32$, $k^\star = 2$ in Fig. 10.

**B.2. Additional Results of Parity Tasks under Small Learning Rates**

We further investigate the behavior of our three algorithms in the small learning rate regime. All experiments are conducted on the parity task with depth $d = 8$, $k^\star + 1 = 4$, and $S^\star = [0, 1, 2]$. Training is performed with a small learning rate $\eta = 0.5$ and batch size $B = 256$ for up to 1000 optimization steps. Model performance is evaluated every 2048 samples, and curriculum stage transitions are triggered once the validation accuracy exceeds 0.99. We report results averaged over 10

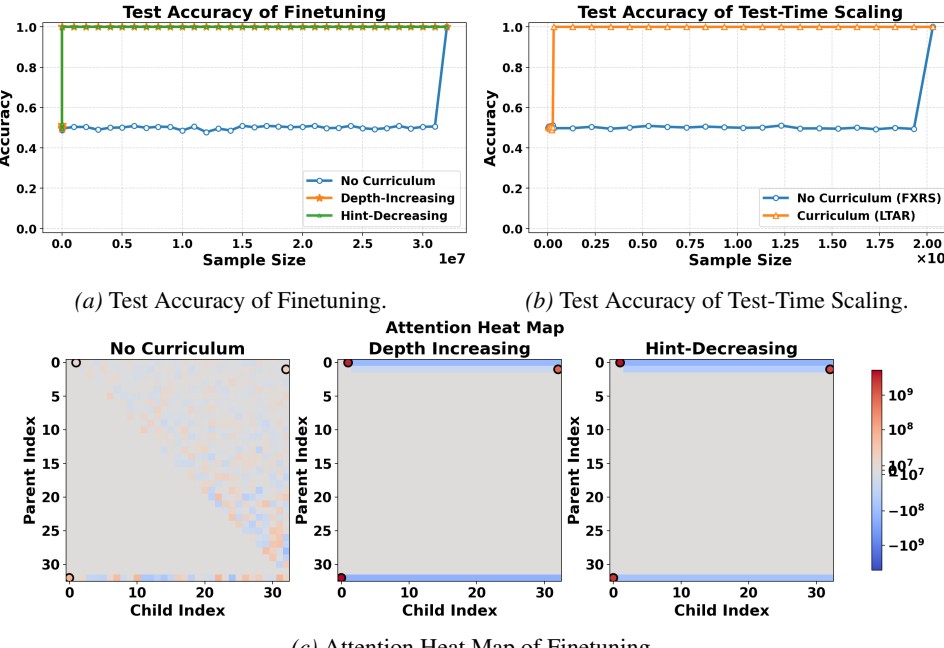

*(a)* Test Accuracy of Finetuning.      *(b)* Test Accuracy of Test-Time Scaling.

*(c)* Attention Heat Map of Finetuning.

*Figure 10.* Post-training Dynamics on the parity task ($d = 32$, $k^\star = 2$, $S^\star = [0, 1]$, $\eta_A = 2 \times 10^{12}$, $\eta_B = \eta_C = 10^{10}$). (a) Test accuracy evolution of finetuning along sample complexity; (b) Test accuracy evolution of Test Time-Scaling along sample complexity; (c) attention heatmap after finetuning. For finetuning, Depth-Increasing and Hint-Decreasing Curriculum Algorithms both converges at $n = 260$, where No Curriculum didn't converge till $n = 32000260$; For test-time scaling, curriculum algorithm (LTAR in Alg. 6) converges at $n = 310000$ while its no curriculum counterpart converges at $n = 20310000$.

independent runs with different random seeds, and compare four optimizers (SGD with momentum 0.9, Adam, Muon with Nesterov momentum, and AdamW), each trained independently under identical settings.

Fig. 11 shows that curriculum-based methods converge substantially faster than the non-curriculum baseline across all optimizers. Interestingly, under the no-curriculum setting, Muon and SGD are still able to make gradual progress, albeit at a much slower rate, whereas Adam and AdamW fail to achieve meaningful improvement. One interesting future direction is to further investigate the underlying rationale of this phenomenon.

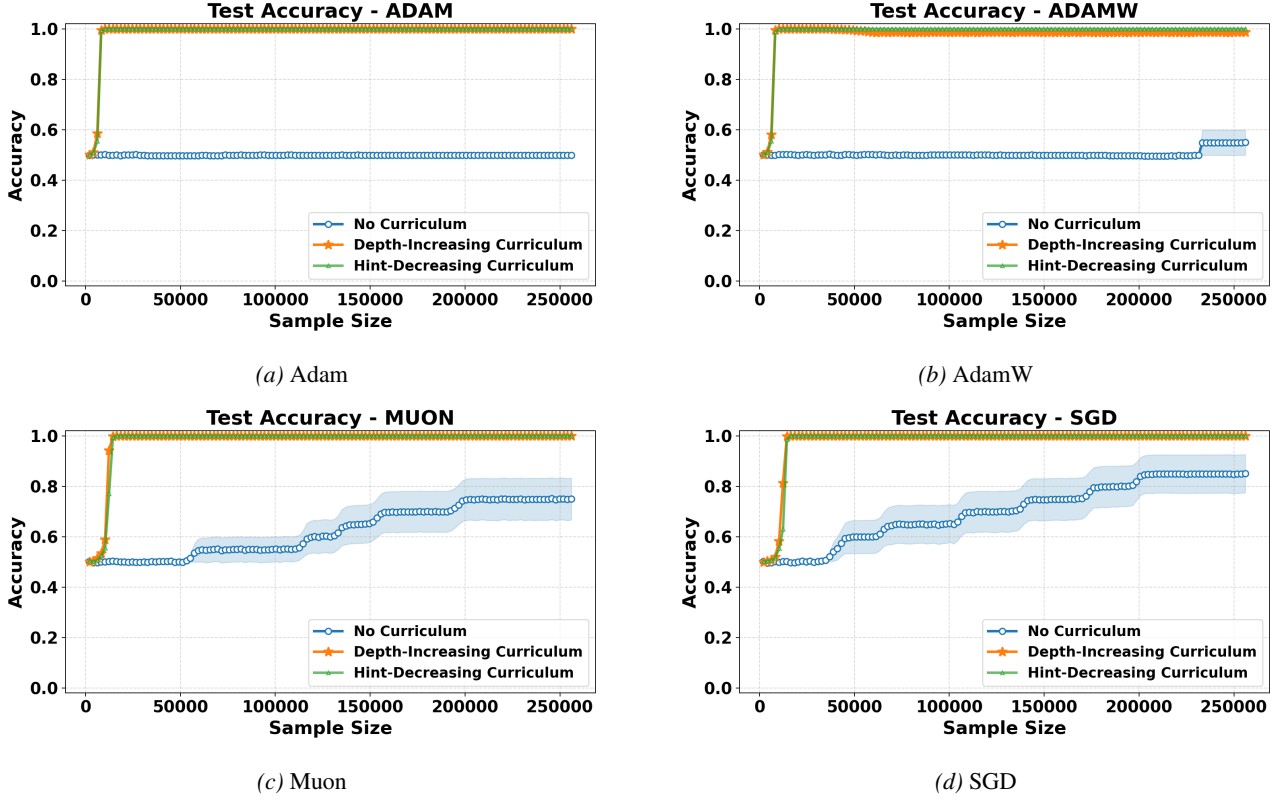

*Figure 11.* Small Learning rate regime ($d = 8, k^\star = 3, S^\star = [0, 1, 2], \eta = 0.5, B = 256$).

## C. Additional Large-Scale Experiments on Realistic Reasoning Datasets

We instantiate two curriculum mechanisms under a unified GRPO backbone: (i) *Depth-Increasing Curriculum (E2H)* (Parashar et al., 2025), and (ii) *Hint-Decreasing Curriculum* in two variants: a practical prefix-SFT interpolation used in our Countdown experiments, and a UFT-aligned objective decomposition used in our Math experiments (Liu et al., 2025b). The optimizer, reward interface, and rollout system are kept fixed; only curriculum-specific data exposure or loss partition changes.

Let

$$\mathcal{D} = \{(x_i, y_i^\star, c_i^\star, d_i)\}_{i=1}^N$$

denote the training set, where $x$ is the prompt, $y^\star$ is the target answer, $c^\star$ is teacher rationale (CoT), and $d_i \in \{1, \ldots, L\}$ is the difficulty level. The base RL objective is

$$\mathcal{L}_{\mathrm{RL}}(t) = \mathcal{L}_{\mathrm{pg}}(t) - \alpha \mathcal{L}_{\mathrm{ent}}(t),$$

with normalized time $p_t = t/T \in [0, 1]$ and shared cosine progress

$$s_t = \frac{1 - \cos(\pi p_t)}{2}.$$

**Depth-Increasing Curriculum (E2H (Parashar et al., 2025)).** At step $t$, the maximal exposed difficulty level is

$$\ell_t = 1 + \lfloor (L - 1)s_t \rfloor,$$

and training samples are drawn from

$$\mathcal{D}_t^{\mathrm{E2H}} = \{(x, y^\star, c^\star, d) \in \mathcal{D} : d \leq \ell_t\}.$$

Hence training progressively expands from easy subsets to the full difficulty range.

**Hint-Decreasing Curriculum v1 (Countdown implementation).** For teacher CoT tokens $c^\star = (c_1^\star, \ldots, c_m^\star)$, reveal a prefix length

$$m_t^{(1)} = \lfloor \rho_t m \rfloor, \qquad \rho_t = \rho_{\min} + \frac{\rho_{\max} - \rho_{\min}}{2}(1 + \cos(\pi p_t)).$$

The policy conditions on $(x, c_{1:m_t^{(1)}}^\star)$, and we optimize

$$\mathcal{L}_{\text{UFT-v1}}(t) = \mathcal{L}_{\text{pg}}(t) - \alpha \mathcal{L}_{\text{ent}}(t) + \lambda_t \, \mathcal{L}_{\text{sft}}^{\text{prefix}}(t),$$

where $\lambda_t$ is cosine-decayed in this practical prefix-guided variant. This realizes easy-to-hard transition in *hint dependence*, but without explicit trajectory-level decomposition.

**Hint-Decreasing Curriculum v2 (Math implementation, UFT-aligned).** Following Liu et al. (2025b), let $l_t$ be the trajectory switch index (teacher-guided prefix up to $l_t - 1$, RL rollout from $l_t$ onward). The objective is

$$\mathcal{J}_{\text{UFT-v2}} = \mathbb{E}\left[\mathcal{J}_{\text{value}}\big((s_h, a_h)_{h=l_t}^{H-1}\big) - \beta \sum_{h=l_t}^{H-1} \text{KL}\big(\pi(\cdot|s_h) \,\|\, \pi^{\text{ref}}(\cdot|s_h)\big) + \beta \sum_{h=0}^{l_t-1} \log \pi(a_h^\star|s_h^\star)\right].$$

In implementation form:

$$\mathcal{L}_{\text{UFT-v2}}(t) = \mathcal{L}_{\text{pg}}^{\text{suffix}}(t) - \alpha \mathcal{L}_{\text{ent}}^{\text{suffix}}(t) + \beta \, \mathcal{L}_{\text{sft}}^{\text{prefix}}(t),$$

where the same cosine schedule controls $l_t$ through prefix-length reduction. Thus, unlike v1, the effective RL suffix itself changes with curriculum stage, matching the UFT decomposition more closely.

**No-Curriculum Baseline.** The baseline samples from a fixed training distribution and optimizes $\mathcal{L}_{\text{RL}}(t)$ without difficulty expansion or hint scheduling. For the Countdown and Math results reported in this section, the No-Curriculum baseline uses fixed balanced sampling over the available in-distribution training levels.

### C.1. Implementation on Countdown

All Countdown experiments are implemented in the same `Curriculum_EoH` GRPO pipeline, ensuring a fixed backbone, reward interface, rollout backend, and optimization recipe across all methods. All reported Countdown runs are initialized from the same local Hugging Face snapshot of

$$\texttt{Qwen/Qwen2.5-1.5B-Instruct},$$

and optimized with LoRA adaptation rather than full-parameter finetuning. All training and evaluation jobs are executed on **NVIDIA A100 80GB PCIe** GPUs. In the final reported comparison, the E2H run used a seven-GPU training setup, the Hint-Decreasing and No-Curriculum runs used four-GPU training setups, and the final pass@K comparison plots were rendered with fixed-seed vLLM decoding on two A100 GPUs.

Countdown instances are partitioned by required operation count into

$$\mathcal{D}_2, \mathcal{D}_3, \mathcal{D}_4, \mathcal{D}_5, \mathcal{D}_6,$$

corresponding to Trivial (2 ops), Easy (3 ops), Medium (4 ops), Hard (5 ops), and OOD (6 ops), respectively. The reward is defined as

$$r(x, y) = \begin{cases} 1, & \text{valid equation using all numbers exactly once and equals target,} \\ 0.1, & \text{valid format but incorrect equation,} \\ 0, & \text{otherwise.} \end{cases}$$

For E2H, cosine progress maps to maximal operation depth

$$d_t = 2 + \lfloor 4s_t \rfloor, \qquad \mathcal{D}_{\le d_t} = \bigcup_{k=2}^{d_t} \mathcal{D}_k.$$

For the Countdown hint-decreasing implementation, we use

$$(\rho_{\max}, \rho_{\min}) = (0.9, 0.1), \qquad (\lambda_{\max}, \lambda_{\min}) = (1.0, 0.0), \qquad \alpha = 0.$$

Hence the model first receives a long correct CoT prefix and progressively solves a larger suffix on its own.

*Table 2.* Countdown backbone, hardware, and hyperparameters.

| | |
|---|---|
| Backbone initialization | `Qwen/Qwen2.5-1.5B-Instruct` (local Hugging Face snapshot, LoRA fine-tuning) |
| Hardware | NVIDIA A100 80GB PCIe. Depth-Increasing (E2H (Parashar et al., 2025)) training: 7 GPUs. Hint-Decreasing (UFT (Liu et al., 2025b)) training: 4 GPUs. No-Curriculum GRPO training: 4 GPUs. Final pass@K plotting: 2 GPUs. |
| Dataset | https://huggingface.co/datasets/divelab/countdown/tree/main |
| Train / test levels | Train levels: $0, 1, 2, 3$; test levels: $0, 1, 2, 3, 4$ |
| Train / test size | Train size: 327,680; test size: 1,024 |
| Base objective | GRPO with $\beta = 10^{-3}$ and entropy coefficient $\alpha = 0$ |
| Learning rate | $1 \times 10^{-6}$ |
| Per-device train batch size | 16 |
| Gradient accumulation | 4 |
| Number of generations | 8 |
| Steps per generation | 1 |
| Maximum training steps | 1600 |
| Maximum prompt length | 1000 tokens |
| Maximum completion length | 512 tokens |
| Precision / memory settings | bfloat16, TF32, gradient checkpointing, vLLM server rollout |
| LoRA configuration | rank $r = 32$, $\alpha = 64$, dropout $= 0.1$, target modules $= \{$`q_proj, v_proj`$\}$ |
| Logging / evaluation cadence | logging every 10 steps; intermediate evaluation every 400 steps |
| Curriculum-specific settings | E2H: cosine difficulty schedule. Hint-Decreasing: prefix ratio $0.9 \rightarrow 0.1$, SFT coefficient $1.0 \rightarrow 0.0$, maximum supervised target length 256. No-Curriculum: fixed balanced level sampling. |
| Pass@K evaluation | $K \in \{1, 2, 4, 8, 16, 32, 64, 128, 256\}$ with a fixed random seed and identical decoding configuration across methods |

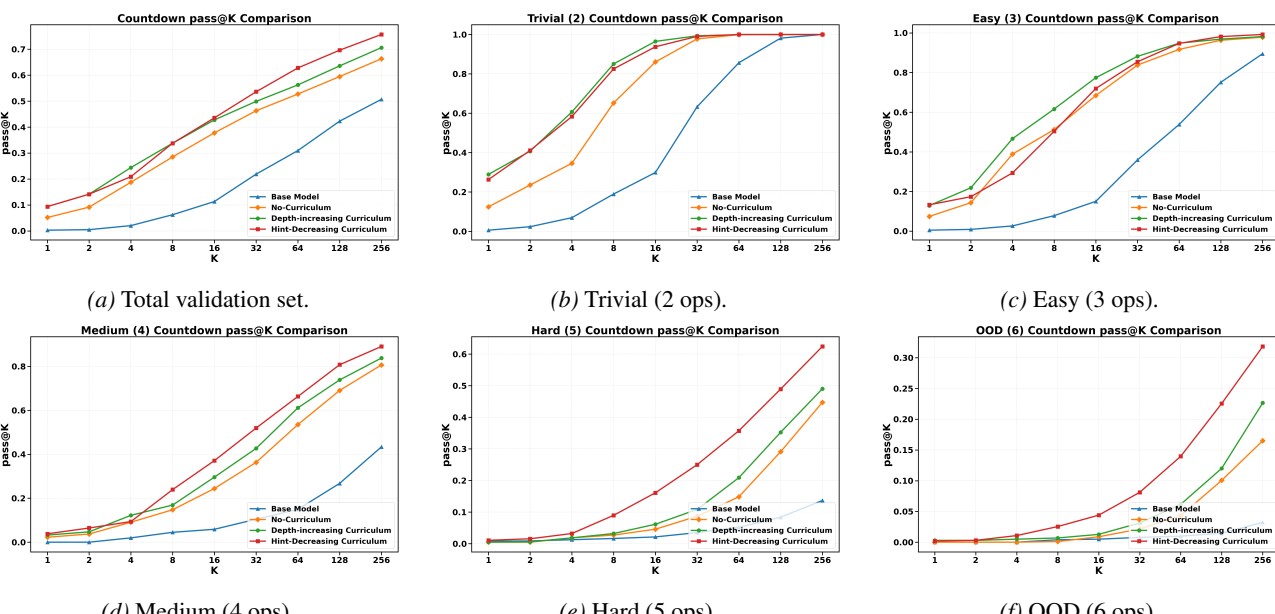

*(a)* Total validation set.      *(b)* Trivial (2 ops).      *(c)* Easy (3 ops).

*(d)* Medium (4 ops).      *(e)* Hard (5 ops).      *(f)* OOD (6 ops).

*Figure 12.* **Test performance on Countdown across difficulty strata.** Both Depth-Increasing (E2H (Parashar et al., 2025), code available at https://github.com/divelab/E2H-Reasoning) and Hint-Decreasing (UFT (Liu et al., 2025b), code available at https://github.com/liumy2010/UFT) curricula outperform the No-Curriculum GRPO baseline as well as the base model Qwen2.5-1.5B Instruct as reasoning depth increases. The largest relative gains appear on the Hard and OOD subsets, indicating improved generalization to longer operation chains.

## C.2. Implementation on Math

All Math experiments are implemented in the same `Curriculum_EoH` GRPO pipeline as Countdown, again with a fixed backbone and reward API. All Math runs are initialized from the same local Hugging Face snapshot of

$$\texttt{Qwen/Qwen2.5-1.5B-Instruct},$$

with LoRA adaptation on top of the frozen base model. All reported Math experiments use **NVIDIA A100 80GB PCIe** GPUs. In our main pipeline, E2H and No-Curriculum training are executed in a four-GPU training setup, while the final pass@K plotting is performed on two A100 GPUs. The final Hint-Decreasing curve reported in Fig. 13 uses the rerun adapter trained from the same Qwen backbone under the UFT-aligned implementation, again on a four-GPU A100 setup.

We use the `divelab/math` dataset with level annotations

$$\mathcal{D}_1, \mathcal{D}_2, \mathcal{D}_3, \mathcal{D}_4, \mathcal{D}_5,$$

corresponding to Trivial (1), Easy (2), Medium (3), Hard (4), and OOD (5), respectively. Training uses in-distribution levels $\mathcal{D}_1 \sim \mathcal{D}_4$, while $\mathcal{D}_5$ is held out for OOD evaluation.

The reward is

$$r(x,y) = \begin{cases} 1, & \text{correct final answer with valid format,} \\ 0.1, & \text{valid format but incorrect answer,} \\ 0, & \text{otherwise,} \end{cases}$$

where correctness is determined by normalized symbolic matching of boxed final answers.

For E2H, cosine progress maps to the maximal in-distribution level

$$d_t = 1 + \lfloor 3s_t \rfloor, \qquad \mathcal{D}_{\leq d_t} = \bigcup_{k=1}^{d_t} \mathcal{D}_k.$$

For Math Hint-Decreasing, we use the UFT-aligned variant in Sec. C with

$$(\rho_{\max}, \rho_{\min}) = (0.95, 0.05), \qquad \text{num\_slices} = 5, \qquad \beta = 10^{-3}, \qquad \alpha = 0.$$

Thus teacher CoT prefixes are initially long and gradually shortened, while the RL-controlled suffix expands correspondingly.

*Table 3.* Math backbone, hardware, and hyperparameters.

| | |
|---|---|
| Backbone initialization | `Qwen/Qwen2.5-1.5B-Instruct` (local Hugging Face snapshot, LoRA fine-tuning) |
| Hardware | NVIDIA A100 80GB PCIe. Depth-Increasing (E2H (Parashar et al., 2025)), Hint-Decreasing (UFT (Liu et al., 2025b)) and No-Curriculum GRPO training: 4 GPUs: 4 GPUs. Final pass@K plotting: 2 GPUs. |
| Dataset | https://huggingface.co/datasets/divelab/math |
| Train / test levels | Train levels: 0, 1, 2, 3; test levels: 0, 1, 2, 3, 4 |
| Train / test size | Train split after level filtering: 5194; test split: 5000 |
| Base objective | GRPO with $\beta = 10^{-3}$ and entropy coefficient $\alpha = 0$ |
| Learning rate | $1 \times 10^{-6}$ |
| Per-device train batch size | 16 |
| Gradient accumulation | 4 |
| Number of generations | 8 |
| Steps per generation | 1 |
| Maximum training steps | 1200 |
| Maximum prompt length | 1600 tokens |
| Maximum completion length | 1600 tokens |
| Precision / memory settings | bfloat16, TF32, gradient checkpointing, vLLM server rollout |
| LoRA configuration | rank $r = 32$, $\alpha = 64$, dropout $= 0.1$, target modules $= \{$`q_proj`, `v_proj`$\}$ |
| Logging / evaluation cadence | logging every 10 steps; intermediate evaluation every 400 steps |
| Curriculum-specific settings | E2H: cosine difficulty schedule over levels $1 \rightarrow 4$. Hint-Decreasing: lower/upper prefix probabilities $(0.05, 0.95)$, `num_slices` $= 5$, prefix SFT coefficient $10^{-3}$, entropy coefficient 0. No-Curriculum: fixed balanced level sampling. |
| Pass@K evaluation | $K \in \{1, 2, 4, 8, 16, 32, 64, 128, 256\}$ with a fixed random seed and identical decoding configuration across methods |

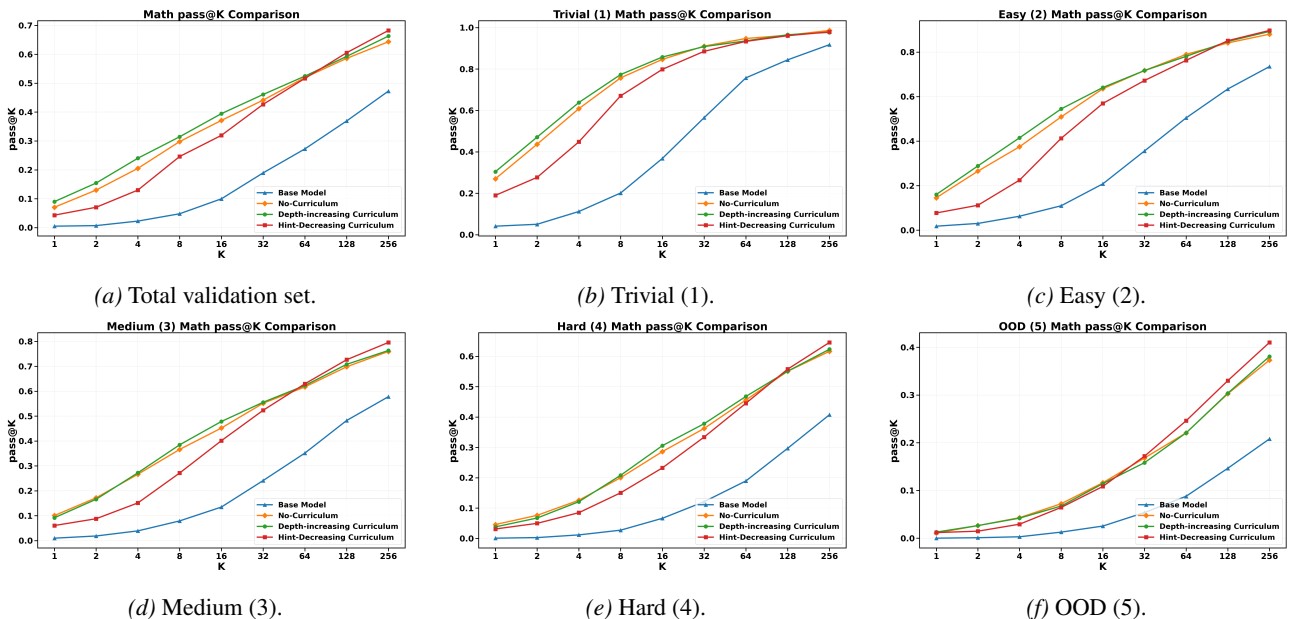

*(a)* Total validation set.      *(b)* Trivial (1).      *(c)* Easy (2).

*(d)* Medium (3).      *(e)* Hard (4).      *(f)* OOD (5).

*Figure 13.* **Test performance on Math across difficulty strata.** We again compare three training strategies implemented in the same GRPO framework: Depth-Increasing Curriculum (adapted from E2H), Hint-Decreasing Curriculum (adapted from UFT), and a No-Curriculum baseline. Both curriculum strategies consistently improve over the Non-Curriculum GRPO as well as the base model, and the gains become larger as problem difficulty increases. The OOD results further suggest that curriculum-guided training improves transfer to harder reasoning regimes.

## C.3. Implementation on Blocksworld

All Blocksworld experiments are implemented in the same GRPO pipeline as Countdown and Math, using the same local snapshot of

$$\texttt{Qwen/Qwen2.5-1.5B-Instruct}$$

with LoRA adaptation on top of the frozen base model. All reported Blocksworld runs use **NVIDIA A100 80GB PCIe** GPUs. For the results reported in Fig. 14, the Depth-Increasing run uses a three-GPU setup (one vLLM rollout GPU and two training GPUs), while the Hint-Decreasing and No-Curriculum runs use five-GPU setups (one vLLM rollout GPU and four training GPUs). Final pass@K plots are rendered with fixed-seed vLLM decoding on two A100 GPUs.

We use the `divelab/blocksworld` dataset with five difficulty strata

$$\mathcal{D}_0, \mathcal{D}_1, \mathcal{D}_2, \mathcal{D}_3, \mathcal{D}_4.$$

Training uses in-distribution levels $\mathcal{D}_0 \sim \mathcal{D}_3$ (8000 examples in total), while $\mathcal{D}_4$ is held out as the hardest OOD evaluation subset. The full evaluation split contains 482 instances.

Unlike Math, the teacher signal in Blocksworld is not a free-form derivation but a gold action plan stored in the `solution` field. The model is prompted to produce a response with a `<think>` block followed by an `<answer>` block, where the `<answer>` block contains the executable action sequence. The reward combines a format term and a plan-execution term:

$$r(x, y) = r_{\mathrm{fmt}}(y) + 2\, r_{\mathrm{plan}}(x, y).$$

Here, $r_{\mathrm{fmt}}(y) = 1$ if the output contains exactly one valid `<think>` block and one valid `<answer>` block in the required order, and 0 otherwise. The term $r_{\mathrm{plan}}(x, y)$ is computed by executing the predicted action sequence in a symbolic Blocksworld simulator starting from the initial state. Invalid action sequences receive zero reward; valid plans that reach the goal receive positive reward, with a mild normalization favoring shorter successful plans relative to the expert plan.

For Depth-Increasing, cosine progress maps to the maximal in-distribution level

$$d_t = \lfloor 3 s_t \rfloor, \qquad \mathcal{D}_{\leq d_t} = \bigcup_{k=0}^{d_t} \mathcal{D}_k.$$

Thus training expands from the easiest planning instances to the full in-distribution set.

For the reported Blocksworld Hint-Decreasing implementation, the gold action sequence in `solution` serves as the teacher hint, and the curriculum decreases the visible expert-plan prefix over training. Concretely, we use the same cosine UFT schedule as in the main pipeline with

$$(\rho_{\max}, \rho_{\min}) = (0.95, 0.05), \qquad \text{num\_slices} = 5, \qquad \beta = 10^{-3}, \qquad \alpha = 0.$$

Hence early training exposes a long prefix of the expert plan, while later training exposes only a short prefix and leaves more of the remaining planning trajectory to be completed by the policy under GRPO. The No-Curriculum baseline uses fixed balanced sampling over $\mathcal{D}_0, \ldots, \mathcal{D}_3$ without difficulty expansion or hint scheduling.

For pass@K, we evaluate

$$K \in \{1, 2, 4, 8, 16, 32, 64, 128, 256\}$$

with the same decoding configuration across all methods (temperature $0.7$, top-$p = 0.9$, top-$k = 50$, and maximum completion length 512).

*Table 4.* Blocksworld backbone, hardware, and hyperparameters.

| | |
|---|---|
| Backbone initialization | `Qwen/Qwen2.5-1.5B-Instruct` (local Hugging Face snapshot, LoRA fine-tuning) |
| Hardware | NVIDIA A100 80GB PCIe. Depth-Increasing (E2H (Parashar et al., 2025)) training: 3 GPUs. Hint-Decreasing (UFT (Liu et al., 2025b)) training: 5 GPUs. No-Curriculum GRPO training: 5 GPUs. Final pass@K plotting uses fixed-seed vLLM decoding. |
| Dataset | https://huggingface.co/datasets/divelab/blocksworld |
| Train / test levels | Train levels: $0, 1, 2, 3$; test levels: $0, 1, 2, 3, 4$ |
| Train / test size | Train split after level filtering: 5000; test split: 482 |
| Base objective | GRPO with $\beta = 10^{-3}$ and entropy coefficient $\alpha = 0$ |
| Learning rate | $1 \times 10^{-6}$ |
| Per-device train batch size | 2 |
| Gradient accumulation | 4 |
| Number of generations | Depth-Increasing / No-Curriculum: 2; Hint-Decreasing: 4 |
| Steps per generation | 2 |
| Maximum training steps | 1600 |
| Maximum prompt length | 1600 tokens |
| Maximum completion length | 512 tokens |
| Precision / memory settings | bfloat16, TF32, gradient checkpointing, vLLM server rollout |
| LoRA configuration | rank $r = 32$, $\alpha = 64$, dropout $= 0.1$, target modules $= \{$`q_proj, v_proj`$\}$ |
| Logging / evaluation cadence | logging every 10 steps; intermediate evaluation every 400 steps |
| Curriculum-specific settings | E2H: cosine difficulty schedule. Hint-Decreasing: lower/upper prefix probabilities $(0.05, 0.95)$, `num_slices` $= 5$, prefix SFT coefficient $10^{-3}$, entropy coefficient 0. No-Curriculum: fixed balanced level sampling. |
| Pass@K evaluation | $K \in \{1, 2, 4, 8, 16, 32, 64, 128, 256\}$ with a fixed random seed and identical decoding configuration across methods |

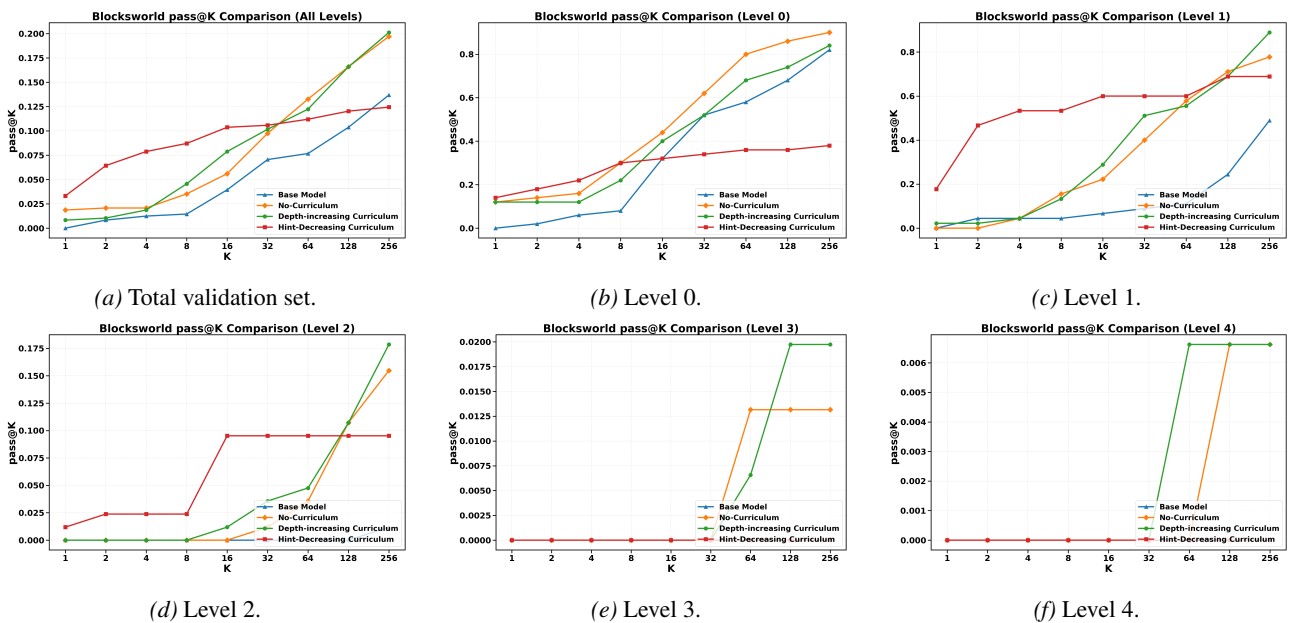

*(a)* Total validation set.  *(b)* Level 0.  *(c)* Level 1.

*(d)* Level 2.  *(e)* Level 3.  *(f)* Level 4.

*Figure 14.* **Test performance on Blocksworld across difficulty strata.** We compare again three training strategies implemented in the same GRPO framework: Depth-Increasing Curriculum, Hint-Decreasing Curriculum, and a No-Curriculum baseline. Both curriculum strategies improve over the base model. On the aggregate Blocksworld test set, Hint-Decreasing is the strongest method in the low-to-medium sampling regime (up to approximately $K = 32$), No-Curriculum is strongest at $K = 64$, and Depth-Increasing catches up at larger sampling budgets, matching the best baseline at $K = 128$ and becoming the strongest method at $K = 256$. Thus, except for $K = 64$, one of the curriculum methods attains the best pass@K performance, indicating complementary benefits of hint-guided and difficulty-guided training in planning-based reasoning.

# D. Cases beyond transformer gradient dynamics over Tree

**Proof.**

*Proof of Cor. 1. The proof is direct leveraging the assumption in Eq. 1.*

$\square$

**Theorem 4** (Curriculum Learning with Spanner Sampling). *Consider the theoretical settings in Foster et al. (2025) (realizable linear softmax parameterization), consider a curriculum of $K$ tasks with increasing difficulty, starting from a base model $\pi_{ref} := \pi_0^\star$. Suppose the sequence of optimal policies $\{\pi_k^\star\}_{k=1}^K$ satisfies*

$$\mathcal{C}_{cov}(\pi_k^\star | \pi_{k-1}^\star) = \Theta(C^\star), \quad \forall k \in [K],$$
$$\mathcal{C}_{cov}(\pi_{k_2}^\star | \pi_{k_1}^\star) = \Theta((C^\star)^{k_2 - k_1}).$$

*For any $\epsilon > 0$ and $\delta \in (0, 1)$, by applying SpannerSampling in Foster et al. (2025) **sequentially** through the curriculum with appropriate choices of $T_{prompt}$, $T_{span}^k$, and $T_{exp}^k$ for each task, the curriculum learning algorithm learns a policy $\hat{\pi}$ with:*

$$\mathbb{E}_{\hat{\pi}_k \sim unif(\hat{\pi}^{(T_{exp}^{k-1})}, \ldots, \hat{\pi}^{(T_{exp}^k)})} \left[ J_\beta(\pi_k^\star) - J_\beta(\hat{\pi}_k) \right] \leq \epsilon, \ k \in [K]$$

*with probability at least $1 - \delta$, and achieves the following computational efficiency bound:*

$$T_{comp}^{curriculum}(\epsilon, \delta) = \tilde{O}\left( K \cdot C^\star \cdot \frac{R_{\max}^2}{\beta^2} \left[ \frac{R_{\max}^2}{\beta} \cdot \frac{d^2 \log^2(\delta^{-1})}{\min\{\epsilon, \beta\}} \right] \right).$$

*Moreover, compared to direct estimation from $\pi_{ref}$ to $\pi_K^\star$, which is*

$$T_{comp}^{direct}(\epsilon, \delta) = \tilde{O}\left( (C^\star)^K \cdot \frac{R_{\max}^2}{\beta^2} \left[ \frac{R_{\max}^2}{\beta} \cdot \frac{d^2 \log^2(\delta^{-1})}{\min\{\epsilon, \beta\}} \right] \right)$$

*Therefore, calling Spanner Sampling in a curriculum manner achieves exponential improvement in computational complexity:*

$$\frac{T_{comp}^{direct}}{T_{comp}^{curriculum}} = \Omega\left(\frac{(C^\star)^{K-1}}{K}\right)$$

*where the direct estimation requires $T_{comp}^{direct} = \tilde{O}\left(\mathcal{C}_{cov}(\pi^\star) \cdot \frac{R_{\max}^2}{\beta^2}\right) \cdot (C^\star)^K$.*

**Proof.** *We work under the realizable linear–softmax parameterization and oracle model of Foster et al. (2025). Let the KL–regularized objective be $J_\beta(\cdot)$ and let $R_{\max}$, $d$, and $\beta$ denote the reward bound, feature dimension, and temperature, respectively. For any pair of policies $(\pi, \pi')$, write the coverage coefficient as $C_{cov}(\pi \mid \pi') := \left|\frac{\pi}{\pi'}\right|_\infty$.*

**Step 1 (Per–task guarantee from Foster et al. (2025)).** *By Theorem 3.1 of Foster et al. (2025), SpannerSampling, when run to target $\pi_k^\star$ starting from a reference $\pi_{k-1}^\star$ in the linear–softmax model, returns a policy $\hat{\pi}$ such that*

$$\mathbb{E}_{\hat{\pi} \sim unif(\hat{\pi}^{(1)}, \ldots, \hat{\pi}^{(T_{\exp})})}[J_\beta(\pi_k^\star) - J_\beta(\hat{\pi})] \leq \epsilon$$

*with probability at least $1 - \delta'$, using a number of oracle computations*

$$T_{comp}^{(k)}(\epsilon, \delta') = \tilde{O}\Big(C_{cov}(\pi_k^\star \mid \pi_{k-1}^\star) \cdot \underbrace{\frac{R_{\max}^2}{\beta^2}\left[\frac{R_{\max}^2}{\beta} \cdot \frac{d^2 \log^2((\delta')^{-1})}{\min\{\epsilon, \beta\}}\right]^2}_{=: \Gamma(\epsilon, \delta')}\Big),$$

*where $\tilde{O}(\cdot)$ hides polylogarithmic factors not depending on the coverage coefficient.[3] We emphasize that data efficiency $T_{data}$ is independent of coverage, while computational efficiency scales with coverage; SpannerSampling matches the linear-in-coverage lower bound.*

**Step 2 (Scheduling across a curriculum).** *Apply Step 1 sequentially for tasks $k = 1, 2, \ldots, K$, using fresh budgets $(T_{prompt}, T_{span}^k, T_{exp}^k)$ at each stage, and set the per–stage failure probability to $\delta' = \delta/K$. The uniform–over–iterates evaluation in the statement mirrors the "return a uniformly random iterate" prescription in Foster et al. (2025), so the stagewise accuracy guarantee transfers verbatim. A union bound then yields that, simultaneously for all $k \in [K]$,*

$$\mathbb{E}_{\hat{\pi}_k \sim unif(\hat{\pi}^{(T_{\exp}^{k-1})}, \ldots, \hat{\pi}^{(T_{\exp}^k)})}[J_\beta(\pi_k^\star) - J_\beta(\hat{\pi}_k)] \leq \epsilon$$

*with probability at least $1 - \delta$. The total compute is the sum of stagewise costs:*

$$T_{comp}^{curriculum}(\epsilon, \delta) = \sum_{k=1}^K T_{comp}^{(k)}(\epsilon, \delta/K) = \tilde{O}\Big(\Gamma(\epsilon, \delta/K) \cdot \sum_{k=1}^K C_{cov}(\pi_k^\star \mid \pi_{k-1}^\star)\Big).$$

**Step 3 (Using the coverage structure of the curriculum).** *By the curriculum assumptions, we have $C_{cov}(\pi_k^\star \mid \pi_{k-1}^\star) = \Theta(C^\star)$ for each $k$, hence*

$$T_{comp}^{curriculum}(\epsilon, \delta) = \tilde{O}\big(K C^\star \cdot \Gamma(\epsilon, \delta/K)\big).$$

*On the other hand, direct estimation of $\pi_K^\star$ from $\pi_{ref} = \pi_0^\star$ is controlled by the single coverage factor $C_{cov}(\pi_K^\star \mid \pi_0^\star) = \Theta((C^\star)^K)$, and Theorem 3.1 gives*

$$T_{comp}^{direct}(\epsilon, \delta) = \tilde{O}\big((C^\star)^K \cdot \Gamma(\epsilon, \delta)\big).$$

**Step 4 (Comparing the two strategies).** *Combining the above displays yields*

---

[3] We only need that $T_{comp}$ is (i) polynomial in $(d, \beta^{-1}, \epsilon^{-1}, \log(1/\delta'))$ and (ii) *linear* (up to logs) in $C_{cov}$, as established by Theorem 3.1 together with the matching lower bound in Foster et al. (2025).

$$\frac{T_{comp}^{direct}(\epsilon,\delta)}{T_{comp}^{curriculum}(\epsilon,\delta)} \;=\; \Omega\!\left(\frac{(C^\star)^{K-1}}{K}\right),$$

*i.e., an exponential-to-polynomial improvement as a function of $K$, as claimed. This recovers the abstract form of Cor. 1 with the concrete instantiation of the per–stage computational complexity provided by Foster et al. (2025).* □

**Remark 5** (Exponential–Polynomial Gaps Under Relaxed Conditions). *The exponential-to-polynomial separation in Thm. 4 remains valid under relaxed coverage conditions, analogous to the discussion after Cor. 1. Specifically, suppose that: (i) for some $k \in [K]$, the per-step coverage is only polynomial rather than constant, i.e.,*

$$C_{cov}(\pi_k^\star \mid \pi_{k-1}^\star) = (C^\star)^{p_k}, \quad 1 \le p_k \ll K,$$

*so that the per-task computational complexity grows as $\tilde{O}((C^\star)^{p_k} \cdot \Gamma(\epsilon,\delta))$; (ii) the direct coverage factor accumulates less than $(C^\star)^K$, relaxing to*

$$C_{cov}(\pi_K^\star \mid \pi_0^\star) = \Theta\big((C^\star)^{\,bK-c}\big), \quad 1 \le b \ll K,\ c \ll K.$$

*Then, under Spanner Sampling, the curriculum complexity satisfies*

$$T_{comp}^{curriculum} = \tilde{O}\Big(K \cdot \max_{k \in [K]} (C^\star)^{p_k} \cdot \Gamma(\epsilon,\delta)\Big) \ll \tilde{\Theta}\big((C^\star)^{bK-c} \cdot \Gamma(\epsilon,\delta)\big) = T_{comp}^{direct}.$$

*Thus, even under relaxed conditions, the curriculum strategy with Spanner Sampling retains an exponential advantage in computational complexity compared to direct estimation.*

**Remark 6** (Relation to E2H (CRL) theory and limits for post-training). *Parashar et al. (2025) analyze curriculum RL for LLMs under an Approximate Policy Iteration (API) lens and derive a stagewise sample complexity bound (their Thm. 3.2). In particular, letting $\epsilon_k$ be per–stage accuracy targets and $L_k$ distribution–sensitivity constants, they obtain $M_{CRL} = \sum_{k=1}^{K} \tilde{O}\big(\log^3(1/\epsilon_k)\,\epsilon_k^{-2}\,L_k^2 \cdots\big)$ and compare it to direct learning via a "curriculum efficiency factor" (CEF). Under geometric schedules $\epsilon_k = \epsilon_K\,e^{K-k}$ and $L_k = L_K/l^{K-k}$, they show*

$$M_{CRL} < M_{Direct} \iff \frac{(e\,l)^{2(1-K)} - 1}{1 - (e\,l)^2} < m - 1,$$

*where $m > 1$ encodes the relative hardness of direct learning (their Eq. (2)). Conceptually, this captures that a well–designed curriculum can turn a multiplicative blow-up across stages into a controlled (geometric) sum. While Parashar et al. (2025) do motivate curricula by a distribution gap between the pretraining distribution $d_0$ and task distribution $d_K$, their API analysis does not instantiate a density-/likelihood-ratio–type coverage term nor a dependence on the base model beyond the abstract factor $m$. Hence their theory, though consonant with ours at a high level (curricula beat direct learning), does not yet provide a coverage–based explanation of post-training that connects "good foundations" (a strong base policy) to concrete computational savings; Theorem 4 fills precisely this gap by making the dependence on coverage explicit.*

**Remark 7** (Relation to UFT complexity measure). *Liu et al. (2025b) propose a complexity notion tailored to unified finetuning: to achieve a 50% pass@1 success rate, the algorithm must explore at least a certain number of nodes in the search space $S_H$. Under this definition, they show that without a curriculum, the exploration complexity can grow exponentially with task depth, whereas curriculum strategies help avoid such exponential bottlenecks. Conceptually, this aligns with our message in Theorem 2, namely that curricula convert exponential costs into polynomial ones.*

*However, their measure does not explicitly account for (i) the number and structure of subtasks, (ii) the relative difficulty of these subtasks, or (iii) the role of a coverage coefficient that quantifies how well a base model supports subsequent targets. Moreover, the UFT framework does not yield a sample–complexity characterization directly, but instead works with node–exploration counts in $S_H$. In contrast, Theorem 2 provides a coverage–based explanation of post-training, connecting subtask decomposition and curriculum schedules to concrete sample–complexity bounds. This makes our result more directly interpretable in terms of the efficiency of real post-training pipelines.*

**Remark 8** (Relation to curriculum via sparse→mixed training in ReLU SGD). *Abbe et al. (2023b) study a model-specific curriculum for learning $k$-parities with a two-layer ReLU network trained by (noisy) gradient descent. Their curriculum is defined by an ordering of input distributions: the network is first trained on sparse inputs, then on the full mixed distribution $D_{\mathrm{mix}} = \rho D_\mu + (1-\rho)D_u$, rather than by changing the task with different correct CoTs or exploiting a foundation model. Under bounded learning rates and a small fraction of sparse inputs, they prove a separation in the number of training steps: curriculum noisy-GD/SGD (sparse-first) learns $k$-parities in $\tilde{O}(d)$ (or $\tilde{O}(d)/\epsilon^2$ under accuracy $\epsilon$), whereas training on randomly ordered (mixed) samples requires at least polynomially more steps (e.g., $\tilde{\Omega}(d^{1+\delta})$ or $\tilde{\Omega}(d^2)$ in stated regimes).*

# E. Auxiliary Lemmas

**Classical probability tools for fixed-confidence identification.** We collect standard concentration and information-theoretic lemmas used in Step 6 of Prop. 1.

**Lemma 1** (Hoeffding/Chernoff inequality for Bernoulli means (Wainwright, 2019)). *Let $X_1, \ldots, X_m$ be i.i.d. Bernoulli(p), and $\hat{p}_m := \frac{1}{m} \sum_{r=1}^m X_r$. Then for any $\epsilon \in (0,1)$,*

$$\mathbb{P}\big(|\hat{p}_m - p| \geq \epsilon\big) \; \leq \; 2 \exp\big(-2m\epsilon^2\big).$$

*Equivalently, to ensure $\mathbb{P}(|\hat{p}_m - p| \geq \epsilon) \leq \delta$ it suffices that $m \geq \frac{1}{2\epsilon^2} \log \frac{2}{\delta}$.*

**Proof.** *See Hoeffding (1963), Chernoff (1952), or Exercise 2.9.(a) in Wainwright (2019).* □

**Lemma 2** (Two-arm fixed-confidence identification lower bound). *Consider two Bernoulli arms with means $p_\star$ and $p$, and gap $\Delta := p_\star - p > 0$. Any (possibly adaptive) $\delta$-correct procedure that outputs the better arm with probability at least $1 - \delta$ must satisfy*

$$\mathbb{E}[T] \; \geq \; c \, \frac{\log(1/\delta)}{\mathrm{KL}(\mathrm{Bern}(p_\star) \,\|\, \mathrm{Bern}(p))} \; \geq \; c' \, \Delta^{-2} \, \log(1/\delta),$$

*for universal constants $c, c' > 0$, where the second inequality uses Pinsker's bound $\mathrm{KL}(\mathrm{Bern}(p_\star) \,\|\, \mathrm{Bern}(p)) \leq 2 \, (p_\star - p)^2$.*

**Proof.** *See Thm. 5 in Mannor & Tsitsiklis (2004) and Thm. 1 in Kaufmann et al. (2016).* □

**Corollary 3** (Per-depth best-arm identification complexity). *Fix depth $\ell$ with $|\mathcal{I}_\ell| = \Theta(d)$ Bernoulli arms and assume a unique best arm with acceptance-gap $\Delta > 0$ to all others. Any $\delta$-correct identification at depth $\ell$ requires at least*

$$\Omega\big(\Delta^{-2} \, \log(d/\delta)\big)$$

*oracle observations in expectation.*

**Proof.** *This follows by applying Lemma 2 to each suboptimal-vs-best pair and distributing the confidence via union bound (or, sufficiency with uniform sampling via Lemma 1).* □

# F. Notation Summary

We have notations summary for main text in Tab. 5 and appendix locally in Tab. 6.

# G. Details and Proofs of Autoregressive Reasoning Tree

**Definition 3** (Full Version of 2-States Conditioned Autoregressive Reasoning Tree (2S-ART)). *Let $[K] := \{1, \ldots, K\}$ be the dictionary and let $\mathrm{EOS} := K+1$. Fix a depth $L \in \mathbb{N}$. A 2S-ART is specified by maps*

$$\{\Phi_l\}_{l=1}^L, \qquad \{\mathcal{I}_l\}_{l=1}^{L+1},$$

*and induces a function class $\mathcal{F}_{\text{2S-ART}}$ of autoregressive tasks $f : [K]^d \to [K+1]^{\leq L+1}$. Given an input $\mathbf{x} = (x_1, \ldots, x_d) \sim \mathrm{Unif}([K]^d)$, define the output chain-of-thought (CoT) by the following iterative rule.*

***State and selectors.*** *For step $l = 1, 2, \ldots$, let*

$$\mathbf{CoT}_{l-1} := (x_1, \ldots, x_d, \mathrm{EOS}, z_1, \ldots, z_{l-1}) \in [K+1]^{d+l}, \quad z_0 := \mathrm{EOS}.$$

*The legal index selector $\mathcal{I}_l : [K+1]^{d+l-1} \to 2^{[d+l]}$ returns a set $\mathcal{I}_l(\mathbf{CoT}_{l-1}) \subseteq [d+l]$ with $|\mathcal{I}_l(\mathbf{CoT}_{l-1})| = \Theta(d)$ for all $l \leq L$, and the terminal selector satisfies $\mathcal{I}_{L+1}(\cdot) = \{d+1\}$. At step $l$, choose an index $i_l \in \mathcal{I}_l(\mathbf{CoT}_{l-1})$ and let $v_l := (\mathbf{CoT}_{l-1})_{i_l} \in [K+1]$.*

***Two-state update.*** *Each update is computed from the previous reasoning state and the chosen clue:*

$$\Phi_l : [K+1] \times [K+1] \to [K+1], \qquad z_l := \Phi_l\big(z_{l-1}, v_l\big), \quad l = 1, 2, \ldots$$

| Symbol | Meaning |
|--------|---------|
| $V$ | Input vocabulary / alphabet size in the 2S-ART definition; accordingly, $\text{EOS} = V + 1$. |
| $K$ | The pass@K parameter used in evaluation. |
| $d$ and $L$ | Input length and maximum reasoning depth / number of curriculum stages in Cor. 1. |
| $\mathbf{x} = (x_1, \ldots, x_d)$ and EOS | Input sequence and the end-of-sequence token. |
| $\mathcal{F}_{\text{2S-ART}}$ and $f_{S_\star}$ | The 2S-ART task class and a target autoregressive reasoning task. |
| $S_\star = (i_1, \ldots, i_{k^\star}, d+1)$, $S_\star^m$, and $k^\star$ | Target index path, its length-$m$ prefix, and the target reasoning depth before EOS. |
| $\mathbf{CoT}_{\ell-1}$ | The partial chain-of-thought available at step $\ell$: $(x_1, \ldots, x_d, \text{EOS}, z_1, \ldots, z_{\ell-1})$. |
| $\mathcal{I}_l(\mathbf{CoT}_{\ell-1})$, $i_l$, and $v_l$ | The legal selector, the chosen index, and the chosen token/clue at step $\ell$, where $v_l := (\mathbf{CoT}_{\ell-1})_{i_l}$. |
| $\Phi_l$ and $z_l$ | The depth-$\ell$ two-state update map and the reasoning state, with $z_l = \Phi_l(z_{l-1}, v_l)$. |
| $\mathcal{F}_{S_\star} = \{f_{S_\star^1}, \ldots, f_{S_\star^{k^\star}}\}$ | Curriculum subtask family induced by the prefixes of the target path. |
| PART and $\pi_{\text{PART}}$ | The probabilistic 2S-ART base sampler and its induced policy, namely the uniform sampler over legal children at each depth. |
| $\pi_{\text{ref}}, \pi^\star, \pi_\ell^\star, \pi_{S^\ell_\star}$, and $N_\varepsilon(\pi_2 \mid \pi_1)$ | Base policy, target policy, stage-wise optimal policies in the abstract theorem, prefix-task optimal policies in the 2S-ART instantiation, and the corresponding sample-complexity notion. |
| $\left\| \frac{\pi'}{\pi} \right\|_\infty$ and $C^\star$ | Coverage coefficient (density ratio) and the geometric curriculum scale $C^\star := \sqrt[L]{N_\varepsilon(\pi^\star \mid \pi_{\text{ref}})}$. |
| $\mathbf{U}, \mathbf{P}, \mathbf{E}[\cdot], d_{\text{X}}$, and $d_{\text{E}}$ | Token-embedding matrix, positional embeddings, combined embeddings, token-embedding dimension, and total embedding dimension. |
| $\mathbf{K}, \mathbf{Q}, \mathbf{V}, \mathbf{W}, \beta$, and $\text{FFN}_l$ | Transformer key/query/value parameters, trainable block, temperature, and depth-$\ell$ feedforward module. |
| $\mathbf{R}_{\mathbf{x}}^{f_{S_\star}}$ and $\mathbf{R}_{\mathbf{x}}^{\mathcal{F}_{S_\star}}$ | Terminal oracle for the full task and family oracle for the curriculum subtasks. |
| $T_{\text{data}}, T_{\text{comp}}, \rho_{\text{spur}}^{\text{sup},>\ell}, \bar{\rho}_{\text{spur}}, \rho_{\text{spur}}^{\text{sup},\ell}$, and $\rho_{\text{spur}}^{\max}$ | Reward-oracle query complexity, model-sampling complexity, and the appendix-level spurious-acceptance parameters used in the outcome-only oracle analysis. |

*Table 5.* Notation summary used in the main paper and appendix.

*with the EOS-absorbing property*

$$\Phi_l(z, \text{EOS}) = \text{EOS} \quad \text{for all } z \in [K+1], \ \ l = 1, \ldots, L.$$

*Hence, once $v_l = \text{EOS}$, all subsequent states remain EOS, denoting the end of the output.*

**Output.** *Let $\ell := \min\{l \leq L : z_l = \text{EOS}\}$ (with the convention $\ell = L$ if no such $l$ exists). The task $f$ outputs the EOS-terminated CoT*

$$f(\mathbf{x}) = (z_1, \ldots, z_\ell, \text{EOS}).$$

*Because $|\mathcal{I}_l(\mathbf{CoT}_{l-1})| = \Theta(d)$ for $l \leq L$, each step selects from $\Theta(d)$ possible clues, giving a tree with branching $\Theta(d)$.*

**Subtask family.** *For a target task $f_{S_\star} \in \mathcal{F}_{\text{2S-ART}}$ with realized index path $S_\star = (i_1, \ldots, i_{k^\star}, d+1)$, define prefix paths $S_\star^m := (i_1, \ldots, i_m, d+1)$ for $m = 1, \ldots, k^\star$. The associated subtask family is*

$$\mathcal{F}_{S_\star} := \{f_{S_\star^1}, \ldots, f_{S_\star^{k^\star}}\} \subset \mathcal{F}_{\text{2S-ART}},$$

*where each $f_{S_\star^m}$ is induced by the same $\{\Phi_l\}, \{\mathcal{I}_l\}$ but terminates immediately after choosing the $m$-th index (the next choice is forced to $d+1$).*

### G.1. Proof of Representation Theorem

**Proof.** *Proof of Corollary 2. At each depth $t = 1, \ldots, l$, the probability of selecting the unique legal child consistent with $S_\star^t$ is $\Theta(d^{-1})$ by the 2S-ART definition (near-uniform over $|\mathcal{I}_t| = \Theta(d)$ legal indices). After producing $z_l$, the next step must legally select EOS to terminate, which occurs with probability $\Theta(d^{-1})$. By the chain rule of conditional probabilities along the unique legal path, the total probability is the product of $(l + 1)$ factors $\Theta(d^{-1})$, i.e., $\Theta(d^{-(l+1)})$.* $\square$

**Proof.** *Proof of Theorem 1. Under the above assumption on $\text{FFN}_l$, it remains to realize the PART index-sampling policy via attention. It suffices to choose parameters satisfying*

| Symbol | Meaning in the appendix |
|---|---|
| FXRS, BAI, and LTAR | Forced X-token Rejection Sampling, Best-Arm Identification over candidate children, and Layer-wise Truncated Accept-Reject. |
| $\Pr_{\text{TF}_{\text{base}}}(\cdot \mid \mathbf{x})$ and $\Pr[\cdot]$ | Conditional probability over the internal sampling of $\text{TF}_{\text{base}}$ given fixed input $\mathbf{x}$, and the corresponding marginal probability after averaging over $\mathbf{x}$. |
| $\mathbf{CoT}_{\ell-1}$, $\mathcal{H}$, and $\mathcal{H}_{\ell-1}$ | Concrete partial rollout/context at depth $\ell$, the committed partial CoT maintained by the algorithms, and the event that the history up to depth $\ell-1$ is correct. |
| $j$, $j_\star$, $j_t^\star$, and $J_t$ | A candidate child index, the unique correct child at the current depth, the unique correct child at future depth $t$, and the random child index sampled at depth $t$. |
| $C_\ell(j)$ | The forcing event that, at depth $\ell$, the visible token is forced to be $x_j$. |
| $E_{\text{true}}$ and $E_{\text{spur}}$ | The true-suffix event and the spurious-but-accepted suffix event in the terminal-oracle analysis. |
| $P_\ell^{\text{term}}(j)$ and $\Delta$ | Expected terminal-oracle acceptance after forcing candidate $j$ at depth $\ell$, and the corresponding one-vs-best acceptance gap. |
| $\rho_{\text{spur}}^{\text{sup},>\ell}$ and $\bar{\rho}_{\text{spur}}$ | Suffix-level spurious-acceptance parameter for the terminal oracle, and its uniform bound across depths. |
| $A_\ell(j)$, $\alpha_j^{(\ell)}$, and $\Delta_\ell$ | Acceptance event under the family oracle with truncation at depth $\ell$, its marginal acceptance probability, and the depth-$\ell$ one-vs-best gap. |
| $\rho_{\text{spur}}^{\text{sup},\ell}$, $\rho_{\text{spur}}^{\max}$, and $\eta_\ell$ | Per-depth spurious-acceptance parameter for the family oracle, its maximum across depths, and a proof-local slack / non-ideality term for the correct child. |
| $T_{\max}$, $m$, and $m_\ell$ | Maximum retry budget in forcing, per-arm repetition count in the terminal-oracle BAI routine, and per-depth per-arm repetition count in LTAR. |
| $Y_{j,r}$, $c_j$, $\hat{p}_j$, and $\hat{j}_\ell$ | Bernoulli observation, cumulative counter, empirical acceptance estimate, and selected best child at depth $\ell$. |
| $\hat{\boldsymbol{\mu}}^{x_{i_\ell}}$ and $\hat{\boldsymbol{\mu}}^{z_\ell}$ | The emitted visible-token embedding and the computed reasoning-state embedding during forcing / commit steps. |
| $c_\ell$ and $s(k,\ell)$ | The depth-specific equalized legal-logit constant and the attention score from position $k$ to the depth-$\ell$ query in the representation-theorem proof. |

*Table 6.* Appendix-local notation used only in the proofs and algorithms.

$$\mathbf{K}^\top\mathbf{Q} = \begin{pmatrix} \mathbf{0}_{d_{\text{X}}\times d_{\text{X}}} & \mathbf{0}_{d_{\text{X}}\times(d_{\text{E}}-d_{\text{X}})} \\ \mathbf{0}_{(d_{\text{E}}-d_{\text{X}})\times d_{\text{X}}} & \mathbf{W} \end{pmatrix}, \qquad \mathbf{V} = \begin{pmatrix} \mathbf{0} & \mathbf{I}_{d_{\text{E}}-d_{\text{X}}} \end{pmatrix}, \tag{11}$$

*so that* $\mathbf{V}\,\mathbf{E}[x_m] = p_m$ *(position only) for any token* $x_m$, *and attention logits depend only on positional encodings:* $\mathbf{E}[u]^\top\mathbf{K}^\top\mathbf{Q}\,\mathbf{E}[z_{l-1}] = \mathbf{p}_k^\top\mathbf{W}\,\mathbf{p}_{d+1+l}$, $\mathbf{p}_{d+1+l} := \mathbf{p}_{i_l}$ *if* $u$ *is at position* $k$. *Moreover, for each* $l$ *choose constants* $c_l > 0$ *and impose a legality mask that sets logits of all* $k \notin \mathcal{I}_l(z_{<l})$ *to* $-\infty$, *while ensuring*

$$\mathbf{p}_k^\top\mathbf{W}\,\mathbf{p}_{d+1+l} = c_l \quad \text{for all } k \in \mathcal{I}_l(z_{<l}). \tag{12}$$

*Because* $\mathbf{p}_i \perp \mathbf{U}$ *and the block structure of* $\mathbf{K}^\top\mathbf{Q}$ *removes content–content and content–position interactions, the attention score from any memory token at position* $k$ *to the query* $\mathbf{E}[z_{l-1}]$ *depends only on positions:* $s(k,l) = \mathbf{p}_k^\top\mathbf{W}\,\mathbf{p}_{d+1+l}$. *By construction and the legality mask,* $s(k,l) = c_l$ *for* $k \in \mathcal{I}_l(z_{<l})$ *and* $s(k,l) = -\infty$ *otherwise (causal masking already suppresses all positions beyond* $d+l$). *Hence the softmax distribution over legal keys is exactly uniform:* $\Pr(i_l \mid z_{<l}) = |\mathcal{I}_l(z_{<l})|^{-1}$. *By assumption on the FFN layers, for each depth* $l$ *there exists* $\text{FFN}_l$ *such that* $\text{FFN}_l\left(\boldsymbol{\mu}^{v_l(i_l)}, \mathbf{E}[z_{l-1}]_{:d_{\text{X}}}\right)$ *implements* $\Phi_l(z_{l-1}, v_l(i_l))$ *on embeddings, yielding* $\mathbf{E}[z_l] = [\boldsymbol{\mu}^{z_l}, \mathbf{p}_{d+1+l}]^\top$. *At* $l=L$, *set the legality mask to permit only* $k = d+1$ *(EOS), which forces termination as in Definition 1. The construction exactly replicates PART.* □

## G.2. Proof of Finetuning Algorithms

**Lemma 3.** *Fix a step* $l$. *Define masked attention logits and weights:*

$$s_l(j) := \mathbf{p}_j^\top\mathbf{W}\,\mathbf{p}_{c_l} + m_l(j), \quad \alpha_l(j) := \frac{\exp(s_l(j))}{\sum_{q\in\mathcal{I}_l}\exp(s_l(q))}, \quad j \in \mathcal{I}_l,$$

*where $m_l$ encodes causal/legal masking and is independent of $\mathbf{W}$. Define the attention-weighted token vector and vocabulary logits:*

$$\hat{\mathbf{p}}_{l+1}^{\mathrm{att}} := \sum_{j \in \mathcal{I}_l} \alpha_l(j)\, \mathbf{p}_j, \qquad \ell_r := \frac{\langle \hat{\mathbf{p}}_{l+1}^{\mathrm{att}}, \mathbf{p}_r \rangle}{\beta}, \;\; r \in \{1, ..., d+1\}.$$

*Let $p_{\mathrm{vocab}}(r) := \exp(\ell_r)/\sum_q \exp(\ell_q)$, and define the index policy as the probability of sampling the token that matches the $i_l$-th input value:*

$$\pi_{\mathbf{W}}(i_l \mid \mathbf{x}, \hat{\mathbf{p}}^{z_{1:l}}) := p_{\mathrm{vocab}}\big(u = \boldsymbol{\mu}_{x_{i_l}} \mid \hat{\mathbf{p}}_{l+1}^{\mathrm{att}}\big).$$

*Then the score admits the positional outer-product decomposition*

$$\nabla_{\mathbf{W}} \log \pi_{\mathbf{W}}(i_l \mid \cdot) = \sum_{k \in \mathcal{I}_l} \alpha_l(k)\, \eta_l(k)\, \mathbf{p}_k \mathbf{p}_{c_l}^{\top}, \quad \eta_l(k) := \frac{\big\langle \mathbf{p}_k - \hat{\mathbf{p}}_{l+1}^{\mathrm{att}},\ \mathbf{p}_{i_l} - \sum_r p_{\mathrm{vocab}}(r)\mathbf{p}_r \big\rangle}{\beta}.$$

*Moreover, under Eq. ([12](navigation)) (positional orthogonality; $\mathbf{W}$ acts only on positional blocks; the mask $m_l$ is $\mathbf{W}$-independent) and a finetuning regime that only reweights legal transitions within these blocks, the family $\{\mathbf{p}_k \mathbf{p}_{c_l}^{\top}\}$ forms an orthogonal basis for the reachable score directions. Consequently, $\nabla_{\mathbf{W}} \log \pi_{\mathbf{W}}$ decomposes uniquely onto orthogonal tensor blocks $\mathrm{span}\{\mathbf{p}_k\} \otimes \mathrm{span}\{\mathbf{p}_{c_l}\}$ (Orthogonal Block Isolation).*

**Proof.** *We expand every ingredient with step-by-step derivations.*

*(1) Vocabulary softmax – definition and gradient. Fix $l$, and define the normalized vocabulary probabilities:*

$$p_{\mathrm{vocab}}(r) = \frac{\exp(\ell_r)}{\sum_q \exp(\ell_q)}, \qquad \ell_r = \frac{\langle \hat{\mathbf{p}}_{l+1}^{\mathrm{att}}, \mathbf{p}_r \rangle}{\beta},$$

*with classes $r \in \{0, 1, \mathrm{EOS}\}$. The policy is*

$$\pi_{\mathbf{W}}(i_l \mid \cdot) = p_{\mathrm{vocab}}(x_{i_l}) := p_{\mathrm{vocab}}\big(u = \boldsymbol{\mu}_{x_{i_l}} \mid \hat{\mathbf{p}}_{l+1}^{\mathrm{att}}\big).$$

*We differentiate $\log p_{\mathrm{vocab}}(x_{i_l})$ w.r.t. $\hat{\mathbf{p}}_{l+1}^{\mathrm{att}}$:*

$$\nabla_{\hat{\mathbf{p}}_{l+1}^{\mathrm{att}}} \log p_{\mathrm{vocab}}(x_{i_l}) \stackrel{(1.1)}{=} \nabla_{\hat{\mathbf{p}}_{l+1}^{\mathrm{att}}} \Big( \ell_{x_{i_l}} - \log \sum_r e^{\ell_r} \Big) \tag{13}$$

$$\stackrel{(1.2)}{=} \frac{\mathbf{p}_{i_l}}{\beta} - \sum_r \frac{e^{\ell_r}}{\sum_q e^{\ell_q}} \frac{\mathbf{p}_r}{\beta} \tag{14}$$

$$\stackrel{(1.3)}{=} \frac{\mathbf{p}_{i_l} - \sum_r p_{\mathrm{vocab}}(r)\, \mathbf{p}_r}{\beta} \;=:\; \mathbf{g}_l^{\mathrm{vocab}}. \tag{15}$$

*Step (1.1) expands the log-softmax; (1.2) uses $\partial \ell_r / \partial \hat{\mathbf{p}}_{l+1}^{\mathrm{att}} = \mathbf{p}_r / \beta$; (1.3) recognizes $p_{\mathrm{vocab}}$.*

*(2) Attention softmax – Jacobian and gradient w.r.t. $\mathbf{W}$. The masked logits and weights are*

$$s_l(k) = \mathbf{p}_k^{\top} \mathbf{W}\, \mathbf{p}_{c_l} + m_l(k), \qquad \alpha_l(k) = \frac{e^{s_l(k)}}{\sum_{q \in \mathcal{I}_l} e^{s_l(q)}}.$$

*The softmax Jacobian is $\partial \alpha_l(j)/\partial s_l(k) = \alpha_l(j)(\delta_{jk} - \alpha_l(k))$. Since $m_l$ is $\mathbf{W}$-independent,*

$$\nabla_{\mathbf{W}} s_l(k) = \nabla_{\mathbf{W}} \big(\mathbf{p}_k^{\top} \mathbf{W}\, \mathbf{p}_{c_l}\big) = \mathbf{p}_k \mathbf{p}_{c_l}^{\top}.$$

*Thus the gradient of the attention-weighted token vector is:*

$$\nabla_{\mathbf{W}} \hat{\mathbf{p}}_{l+1}^{\mathrm{att}} = \nabla_{\mathbf{W}} \Big( \sum_{j \in \mathcal{I}_l} \alpha_l(j) \mathbf{p}_j \Big) \tag{16}$$

$$\stackrel{(2.1)}{=} \sum_j \mathbf{p}_j \sum_k \frac{\partial \alpha_l(j)}{\partial s_l(k)} \nabla_{\mathbf{W}} s_l(k) \tag{17}$$

$$\stackrel{(2.2)}{=} \sum_j \mathbf{p}_j \sum_k \alpha_l(j) \big( \delta_{jk} - \alpha_l(k) \big) \mathbf{p}_k \mathbf{p}_{c_l}^\top \tag{18}$$

$$\stackrel{(2.3)}{=} \sum_k \alpha_l(k) \Big( \mathbf{p}_{x_k} - \sum_j \alpha_l(j) \mathbf{p}_j \Big) \mathbf{p}_k \mathbf{p}_{c_l}^\top \tag{19}$$

$$\stackrel{(2.4)}{=} \sum_k \alpha_l(k) \big( \mathbf{p}_k - \hat{\mathbf{p}}_{l+1}^{\mathrm{att}} \big) \mathbf{p}_k \mathbf{p}_{c_l}^\top. \tag{20}$$

*Step (2.1) is the chain rule; (2.2) uses the softmax Jacobian and $\nabla_{\mathbf{W}} s_l$; (2.3) collects terms; (2.4) recognizes $\hat{\mathbf{p}}_{l+1}^{\mathrm{att}}$.*

*(3) Chain rule for the policy score. Combining (1)–(2),*

$$\nabla_{\mathbf{W}} \log \pi_{\mathbf{W}}(i_l \mid \cdot) \stackrel{(3.1)}{=} \Big( \nabla_{\hat{\mathbf{p}}_{l+1}^{\mathrm{att}}} \log p_{\mathrm{vocab}}(x_{i_l}) \Big)^\top \cdot \nabla_{\mathbf{W}} \hat{\mathbf{p}}_{l+1}^{\mathrm{att}} \tag{21}$$

$$\stackrel{(3.2)}{=} \sum_{k \in \mathcal{I}_l} \alpha_l(k) \underbrace{\Big\langle \mathbf{p}_k - \hat{\mathbf{p}}_{l+1}^{\mathrm{att}}, \, \mathbf{g}_l^{\mathrm{vocab}} \Big\rangle}_{:= \, \eta_l(k)} \mathbf{p}_k \mathbf{p}_{c_l}^\top. \tag{22}$$

*This yields the claimed positional outer-product decomposition with weights $\alpha_l(k)\eta_l(k)$, where $\mathbf{g}_l^{\mathrm{vocab}} = (\mathbf{p}_{i_l} - \sum_r p_{\mathrm{vocab}}(r)\mathbf{p}_r)/\beta$.*

*(4) Interchange and smoothness conditions. The above steps use: (i) $m_l$ is $\mathbf{W}$-independent, so $\nabla_{\mathbf{W}} s_l$ exists and is continuous; (ii) softmax is $C^\infty$, thus $\alpha_l$ and $p_{\mathrm{vocab}}$ are smooth in $\mathbf{W}$; (iii) boundedness of token embeddings $\mathbf{p}_\cdot$, vocabulary $\mathbf{U}$, and temperature $\beta > 0$ ensures an $L^1$ dominator for Leibniz interchange when taking expectations over trajectories (used later in policy-gradient proofs).*

*(5) Orthogonal Block Isolation (OBI) and the role of Eq. (12). Under Eq. (12): positional embeddings $\{\mathbf{p}_j\}$ are mutually orthogonal; $\mathbf{K}^\top \mathbf{Q}$ (hence $\mathbf{W}$) acts only on positional blocks; $\mathbf{V}$ projects out positional-channel–orthogonal components when forming logits with $\mathbf{U}$. Finetuning reweights only the legal transitions while preserving the block structure (the mask $m_l$ is fixed). Therefore, the reachable score directions lie in the span of $\{\mathbf{p}_k \mathbf{p}_{c_l}^\top\}$, and for any $(k, l) \neq (k', l')$,*

$$\big\langle \mathbf{p}_k \mathbf{p}_{c_l}^\top, \, \mathbf{p}_{k'} \mathbf{p}_{c_{l'}}^\top \big\rangle = (\mathbf{p}_k^\top \mathbf{p}_{k'})(\mathbf{p}_{c_l}^\top \mathbf{p}_{c_{l'}}) = 0.$$

*Hence cross-terms vanish and the decomposition in (3.2) is unique on the block-diagonal positional subspace $\mathrm{span}\{\mathbf{p}_k\} \otimes \mathrm{span}\{\mathbf{p}_{c_l}\}$. This establishes OBI with Eq. (12) as sufficient conditions. In practice, these are met at initialization (masking, orthogonality, block action), and finetuning that only reweights legal transitions preserves the required block-isolation.* □

**Lemma 4** (From per-step logit margins to overall 0-1 loss). *Fix a target subtask family $\mathcal{F}_{S_\star} = \{f_{S_\star^1}, \ldots, f_{S_\star^{k^\star}}\}$ and consider decoding $k^\star + 1$ steps (the last for EOS). For each step $l \in \{1, \ldots, k^\star + 1\}$, let the legal index set be $\mathcal{I}_l$ with cardinality $d_l := |\mathcal{I}_l| \geq 1$, and denote the correct index by $i_l$ (with $i_{k^\star+1} := d + 1$ for EOS). Define masked attention logits $s_l(j) := \mathbf{p}_j^\top \mathbf{W} \mathbf{p}_{c_l} + m_l(j)$ and attention weights $\alpha_l(j) = \mathrm{softmax}_j(s_l(j))$. Let*

$$\Delta_l := s_l(i_l) - \max_{j \in \mathcal{I}_l \setminus \{i_l\}} s_l(j) \geq 0$$

*be the per-step attention logit gap on the correct child. Assume pairwise-orthogonal token embeddings, bounded norms, and temperature $\beta > 0$ so that*

$$\big| \eta_l(k) \big| \leq \frac{4}{\beta}, \qquad \text{and} \qquad p_{\mathrm{vocab}}(r) = \mathrm{softmax}_r \big( \langle \hat{\mathbf{p}}_{l+1}^{\mathrm{att}}, \mathbf{p}_r \rangle / \beta \big)$$

*as in Lemma 3. Let $M := K + 1$ denote the vocabulary size. For any error budget $(\varepsilon_1, \ldots, \varepsilon_{k^\star+1}) \in (0,1)^{k^\star+1}$, define the per-step required attention probability and corresponding margin threshold by*

$$\alpha_l^{\mathrm{req}} := \tfrac{1}{2} + \tfrac{\beta}{2} \log\Big(\tfrac{M(1-\varepsilon_l)}{\varepsilon_l}\Big), \qquad \Gamma_l := \log\Big(\tfrac{d_l-1}{(\alpha_l^{\mathrm{req}})^{-1}-1}\Big),$$

*whenever $\alpha_l^{\mathrm{req}} < 1$ (otherwise the requirement is infeasible).*

*If the per-step logit gaps satisfy $\Delta_l \geq \Gamma_l$ for all $l \in [k^{star} + 1]$, then the per-step selection probability of the correct token obeys*

$$\pi_l := p_{\mathrm{vocab}}\big(u = \mathbf{p}_{i_l} \mid \hat{\mathbf{p}}_{l+1}^{\mathrm{att}}\big) \geq 1 - \varepsilon_l, \quad l = 1, \ldots, k^\star + 1.$$

*Consequently, letting the 0-1 loss be $L := 1 - R^{k^\star}(\mathrm{TF}(\cdot; \mathbf{W}))$ (so that $\mathbb{E}[L] = 1 - \mathbb{E}[R^{k^\star}]$), we have*

$$\mathbb{E}_{\mathbf{x} \sim \mathrm{Unif}([K]^d)}\big[L\big] \leq \sum_{l=1}^{k^\star+1} \varepsilon_l.$$

*In particular, choosing any $(\varepsilon_l)$ with $\sum_l \varepsilon_l \leq \varepsilon$ implies $\mathbb{E}[L] \leq \varepsilon$.*

**Proof.** *Step 1 (attention probability from logit gap). For $\alpha_l(\cdot) = \mathrm{softmax}(s_l(\cdot))$ and gap $\Delta_l$, the correct-child attention weight obeys the standard softmax bound*

$$\alpha_l(i_l) = \frac{1}{1 + \sum_{j \neq i_l} \exp\big(-(s_l(i_l) - s_l(j))\big)} \geq \frac{1}{1 + (d_l - 1)\, e^{-\Delta_l}}.$$

*Thus if $\Delta_l \geq \Gamma_l$ with $\Gamma_l$ defined in the statement, then*

$$\alpha_l(i_l) \geq \alpha_l^{\mathrm{req}}.$$

*Step 2 (vocabulary probability from attention concentration). With orthogonal embeddings, $\langle \hat{\mathbf{p}}_{l+1}^{\mathrm{att}}, \mathbf{p}_{i_l} \rangle = \alpha_l(i_l)$ and for any competitor token $r \neq x_{i_l}$, $\langle \hat{\mathbf{p}}_{l+1}^{\mathrm{att}}, \mathbf{p}_r \rangle \leq 1 - \alpha_l(i_l)$ (competitors include tokens not present in the mixture, whose inner products are $0 \leq 1 - \alpha_l$). Hence the vocabulary-logit gap between the correct token and any competitor is at least*

$$\gamma_l^{\mathrm{vocab}} \geq \alpha_l(i_l) - (1 - \alpha_l(i_l)) = 2\alpha_l(i_l) - 1.$$

*The softmax lower bound then gives*

$$\pi_l = \frac{e^{\alpha_l(i_l)/\beta}}{\sum_r e^{\langle \hat{\mathbf{p}}_{l+1}^{\mathrm{att}}, \mathbf{p}_r \rangle/\beta}} \geq \frac{1}{1 + \sum_{r \neq x_{i_l}} e^{-\gamma_l^{\mathrm{vocab}}/\beta}} \geq \frac{1}{1 + M\, e^{-(2\alpha_l(i_l)-1)/\beta}}.$$

*Therefore $\alpha_l(i_l) \geq \alpha_l^{\mathrm{req}}$ implies*

$$\pi_l \geq \frac{1}{1 + M\, e^{-(2\alpha_l^{\mathrm{req}}-1)/\beta}} = 1 - \varepsilon_l.$$

*Step 3 (from per-step success to sequence success). The sequence fails only if at least one step fails, hence by the union bound*

$$\mathbb{P}(\textit{sequence fails}) \leq \sum_{l=1}^{k^\star+1} (1 - \pi_l) \leq \sum_{l=1}^{k^\star+1} \varepsilon_l.$$

*Taking expectation over $\mathbf{x} \sim \mathrm{Unif}([K]^d)$ does not increase the bound, yielding the displayed inequality for the 0-1 loss $L = 1 - R^{k^\star}$.* $\square$

**Lemma 5** (Policy Gradients for REINFORCE). *Let the input $\mathbf{x}$ be fixed and the decoding length be $k^\star \geq 2$. Denote the generated token sequence by $\hat{\mathbf{p}}^{z_{1:k^\star}} = (\hat{\mathbf{p}}^{z_1}, \ldots, \hat{\mathbf{p}}^{z_{k^\star}})$. At step $l$ $(1 \leq l \leq k^\star + 1)$, an attention policy first samples a secret index's token $i_l \sim \pi_{\mathbf{W}^{k^\star}}(\cdot \mid \mathbf{x}, \hat{\mathbf{p}}^{z_{1:l}}) := \hat{p}_{\mathbf{W}}\big(u = \boldsymbol{\mu}_{x_{i_l}} \mid \mathbf{x}, \hat{\mathbf{p}}^{z_{1:l}}\big)$. The next token is then deterministically produced by the deterministic Feedforward $\mathrm{FFN}_m(\cdot)$ at $m$-th reasoning procedure, so that the only source of randomness is the sampling of*

tokens $\boldsymbol{\mu}_{x_{i_l}}$ corresponding to secret index sequences $i_{1:k^\star+1}$. We therefore define the (random) trajectory as $\tau := (i_{1:k^\star+1})$ and the induced measure

$$p_{\mathbf{W}^{k^\star}}(\tau \mid \mathbf{x}) = \prod_{l=1}^{k^\star+1} \pi_{\mathbf{W}^{k^\star}}\left(i_l \mid \mathbf{x}, \hat{\mathbf{p}}^{z_{1:l}}\right),$$

where the tokens $\hat{\mathbf{p}}^{z_{1:k^\star}}$ are deterministic functions of $\tau$ via $g_{\mathbf{W}^{k^\star}}$. Let the terminal reward be $R^{k^\star}(\mathrm{TF}(\cdot; \mathbf{W}))$, which does not explicitly depend on $\mathbf{W}^{k^\star}$. Define the population loss

$$\mathcal{J}_{\mathrm{REINFORCE}}^{k^\star}(\mathbf{W}^{k^\star}) := \mathbb{E}_{\mathbf{x}\sim\mathcal{P}_x, \tau\sim p_{\mathbf{W}^{k^\star}}(\cdot|\mathbf{x})}\left[R^{k^\star}\left(\boldsymbol{\mu}^{z_{k^\star}}(\tau_{-2})\right)\right]. \tag{23}$$

Assume the following regularity conditions hold: (i) for all $l$, $\pi_{\mathbf{W}^{k^\star}}(i_l \mid \cdot) > 0$ is Fréchet differentiable in $\mathbf{W}^{k^\star}$ with log-smoothness; (ii) $R^{k^\star}$ is bounded and integrable; and (iii) differentiation can be interchanged with integration (or expectation) under dominated convergence / parameterized measure continuity. Denote $\mathbb{E}_\tau[\cdot] = \mathbb{E}_{\mathbf{x}\sim\mathcal{P}_x, \tau\sim p_{\mathbf{W}^{k^\star}}(\cdot|\mathbf{x})}[\cdot]$, the policy gradients are:

$$\nabla_{\mathbf{W}^{k^\star}} \mathcal{J}_{\mathrm{REINFORCE}}^{k^\star}(\mathbf{W}^{k^\star}) = \mathbb{E}_\tau\left[R^{k^\star}\left(\boldsymbol{\mu}^{z_{k^\star}}(\tau_{-2})\right) \cdot \sum_{l=1}^{k^\star+1} \nabla_{\mathbf{W}^{k^\star}} \log \pi_{\mathbf{W}^{k^\star}}\left(i_l \mid \mathbf{x}, \hat{\mathbf{p}}^{z_{1:l}}\right)\right]. \tag{24}$$

The formula reflects that token generation is deterministic via $\phi$ while the non-Markovian dependency arises from the index policy $\pi$ depending on the full history. Therefore, by Lemma 3, we have

$$\nabla_{\mathbf{W}^{k^\star}} \mathcal{J}_{\mathrm{REINFORCE}}^{k^\star}(\mathbf{W}^{k^\star}) = \mathbb{E}_\tau\left[R^{k^\star}\left(\boldsymbol{\mu}^{z_{k^\star}}(\tau_{-2})\right) \sum_{l=1}^{k^\star+1} \sum_{k\in\mathcal{I}_l} \alpha_l^{k^\star}(k)\, \eta_l^{k^\star}(k)\, \mathbf{p}_k \mathbf{p}_{c_l}^\top\right], \tag{25}$$

where

- $\alpha_l^{k^\star}(k) := \mathrm{softmax}_k(\mathbf{p}_k^\top \mathbf{W}^{k^\star} \mathbf{p}_{c_l} + m_l(k))$,

- $\hat{\mathbf{p}}_{l+1}^{\mathrm{att},k^\star} := \sum_{j\in\mathcal{I}_l} \alpha_l^{k^\star}(j)\mathbf{p}_j$,

- $\eta_l^{k^\star}(k) := \langle \mathbf{p}_k - \hat{\mathbf{p}}_{l+1}^{\mathrm{att},k^\star}, \mathbf{p}_{i_l} - \sum_r p_{\mathrm{vocab}}^{k^\star}(r)\mathbf{p}_r\rangle/\beta$,

- $p_{\mathrm{vocab}}^{k^\star}(\cdot) := \dfrac{\exp(\frac{\langle \hat{\mathbf{p}}_{l+1}^{\mathrm{att}}, \mathbf{p}_r\rangle}{\beta})}{\sum_q \exp(\frac{\langle \hat{\mathbf{p}}_{l+1}^{\mathrm{att}}, \mathbf{p}_q\rangle}{\beta})}$.

**Proof.** Write $\tau = (i_{1:k^\star+1})$ and abbreviate $\mathbb{E}_\tau$ for $\mathbb{E}_{\tau\sim p_{\mathbf{W}^{k^\star}}(\cdot|\mathbf{x})}$. The tokens $\hat{\mathbf{p}}^{z_{1:k^\star}}(\tau)$ are deterministic given $\tau$ via $g_{\mathbf{W}^{k^\star}}$. We derive the gradient in a step-numbered manner:

$$\nabla_{\mathbf{W}^{k^\star}} \mathcal{J}_{\mathrm{REINFORCE}}^{k^\star} \overset{(1)}{=} \nabla_{\mathbf{W}^{k^\star}} \int R^{k^\star}\left(\boldsymbol{\mu}^{z_{k^\star}}(\tau_{-2})\right) p_{\mathbf{W}^{k^\star}}(\tau \mid \mathbf{x})\, d\tau \tag{26}$$

$$\overset{(2)}{=} \int R^{k^\star}\left(\boldsymbol{\mu}^{z_{k^\star}}(\tau_{-2})\right) \nabla_{\mathbf{W}^{k^\star}} p_{\mathbf{W}^{k^\star}}(\tau \mid \mathbf{x})\, d\tau \tag{27}$$

$$\overset{(3)}{=} \int R^{k^\star}\left(\boldsymbol{\mu}^{z_{k^\star}}(\tau_{-2})\right) p_{\mathbf{W}^{k^\star}}(\tau \mid \mathbf{x}) \nabla_{\mathbf{W}^{k^\star}} \log p_{\mathbf{W}^{k^\star}}(\tau \mid \mathbf{x})\, d\tau \tag{28}$$

$$\overset{(4)}{=} \mathbb{E}_\tau\left[R^{k^\star}\left(\boldsymbol{\mu}^{z_{k^\star}}(\tau_{-2})\right) \sum_{l=1}^{k^\star+1} \nabla_{\mathbf{W}^{k^\star}} \log \pi_{\mathbf{W}^{k^\star}}\left(i_l \mid \mathbf{x}, \hat{\mathbf{p}}^{z_{1:l}}\right)\right], \tag{29}$$

which equals equation 24.

Step (1) is the definition of $\mathcal{J}_{\mathrm{REINFORCE}}^{k^\star}$; Step (2) uses the Leibniz interchange $\nabla_\theta \int f(\tau, \theta)d\tau = \int \nabla_\theta f(\tau, \theta)d\tau$ under the following conditions: (i) $R^{k^\star} \circ g_{\mathbf{W}^{k^\star}}$ is bounded; (ii) $p_{\mathbf{W}^{k^\star}}(\tau \mid \mathbf{x})$ is differentiable in $\mathbf{W}^{k^\star}$; (iii) there exists an integrable

*dominator $h(\tau)$ with $\|R^{k^\star}\nabla p\| \le h$ (dominated convergence / parameterized measure continuity). These hold because $R^{k^\star} \in \{0,1\}$ and $\pi$ is softmax-based with log-smoothness. Step (3) is the score-function identity $\nabla p = p\nabla \log p$. Step (4) applies the chain rule to $\log p_{\mathbf{W}^{k^\star}}(\tau \mid \mathbf{x}) = \sum_l \log \pi_{\mathbf{W}^{k^\star}}(i_l \mid \cdot)$, which holds regardless of parameter sharing across time. The resulting decomposition does not require Markovity of the state, since the history enters through the conditioning of $\pi$.*

*Finally, we note that deterministic $\phi$ does not affect the score terms and only enters through $R^{k^\star}\big(\boldsymbol{\mu}^{z_{k^\star}}(\tau_{-2})\big)$, preserving all interchanges above. Fubini's theorem applies whenever $\mathbb{E}_\tau[\,|R^{k^\star}\sum_l \nabla \log \pi_l|\,] < \infty$, which holds by boundedness of $R^{k^\star}$ and square-integrability (log-smoothness) of the score.*

*By Lemma 3, for each $l$ we have the score decomposition*

$$\nabla_{\mathbf{W}^{k^\star}} \log \pi_{\mathbf{W}^{k^\star}}(i_l \mid \cdot) = \sum_{k\in\mathcal{I}_l} \alpha_l^{k^\star}(k)\eta_l^{k^\star}(k)\mathbf{p}_k\mathbf{p}_{c_l}^\top.$$

*Substituting this into the REINFORCE identity*

$$\nabla \mathcal{J}_{\text{REINFORCE}} = \mathbb{E}[R\sum_l \nabla \log \pi]$$

*and using linearity of expectation, yields the final expanded formula. Orthogonal Block Isolation guarantees each term lies in a unique positional tensor block $p_k p_{c_l}^T$, ensuring no cross-block interference in these expansions.* $\square$

**Lemma 6** (One-step REINFORCE update of per-step attention logit gap). *Fix a step $l \in \{1,\ldots,k^\star+1\}$. Define*

$$s_l(j;\mathbf{W}) := \mathbf{p}_j^\top \mathbf{W}\, \mathbf{p}_{c_l} + m_l(j), \qquad \Delta_l(\mathbf{W}) := s_l(i_l;\mathbf{W}) - \max_{j\in\mathcal{I}_l\setminus\{i_l\}} s_l(j;\mathbf{W}).$$

*Consider the one-step REINFORCE update that maximizes the reward (step size $\eta > 0$)*

$$\mathbf{W}^{(t+1)} = \mathbf{W}^{(t)} + \eta\,\nabla_{\mathbf{W}}\mathcal{J}_{\text{REINFORCE}}^{k^\star}(\mathbf{W}^{(t)}),$$

*with all gradients evaluated at $\mathbf{W}^{(t)}$. Let $\mathbb{E}_\tau^{(t)}[\cdot]$ denote the trajectory expectation under the policy at $\mathbf{W}^{(t)}$ in Lemma 5, and reuse the notation of Lemma 3*

$$\alpha_l^{(t)}(j) = \text{softmax}_j\big(\mathbf{p}_j^\top \mathbf{W}^{(t)}\mathbf{p}_{c_l} + m_l(j)\big), \quad \eta_l^{(t)}(j) = \frac{\langle \mathbf{p}_j - \hat{\mathbf{p}}_{l+1}^{\text{att},(t)},\ \mathbf{p}_{i_l} - \sum_r p_{\text{vocab}}^{(t)}(r)\mathbf{p}_r\rangle}{\beta}.$$

*Introduce the path-success and spurious-success decomposition. Let $\mathcal{T}_{\text{true}}$ be the set of trajectories that select the correct indices $i_{1:k^\star}$ (and then EOS); define the true-path success probability*

$$p_{\text{path}} := \mathbb{E}_{\mathbf{x}}\,\mathbb{P}_{\tau|\mathbf{x}}\{\tau \in \mathcal{T}_{\text{true}}\} = \mathbb{E}_{\mathbf{x}}\Big[\prod_{l=1}^{k^\star+1} \pi_{\mathbf{W}^{k^\star}}\big(i_l \mid \mathbf{x}, \hat{\mathbf{p}}^{z_{1:l}}\big)\Big].$$

*Let the spurious (reward-hacking) success rate be*

$$\rho_{\text{spur}} := \mathbb{E}_{\mathbf{x}}\Big[\mathbb{P}_{\tau|\mathbf{x}}\big\{R^{k^\star}(\hat{\mathbf{p}}^{z_{k^\star}}(\tau)) = 1 \mid \tau \notin \mathcal{T}_{\text{true}}\big\}\Big],$$

*which depends on the task structure encoded by $\{\Phi_l\}, \{\mathcal{I}_l\}$. Then, for any policy,*

$$p_{\text{succ}} = p_{\text{path}} + (1 - p_{\text{path}})\,\rho_{\text{spur}}. \tag{30}$$

*In particular, at near-uniform initialization with $d_l := |\mathcal{I}_l|$ and $\pi(i_l \mid \cdot) \approx 1/d_l$ on legal children,*

$$p_{\text{path}} \le \prod_{l=1}^{k^\star+1} \frac{1}{d_l}, \qquad p_{\text{succ}} \le \rho_{\text{spur}} + (1 - \rho_{\text{spur}})\prod_{l=1}^{k^\star+1}\frac{1}{d_l}. \tag{31}$$

*For the parity case study with $\mathbf{x} \sim \text{Unif}\{0,1\}^d$ and XOR kernels $\{\Phi_l\}$, any wrong index set yields the correct parity with probability $1/2$; hence*

$$\rho_{\text{spur}}^{\text{parity}} = \tfrac{1}{2}, \qquad p_{\text{succ}} \le \tfrac{1}{2} + \tfrac{1}{2}\prod_{l=1}^{k^\star+1}\frac{1}{d_l}. \tag{32}$$

*Then:*

*1) (exact logit update for affine logits) For any $j \in \mathcal{I}_l$, we have*

$$s_l\big(j; \mathbf{W}^{(t+1)}\big) = s_l\big(j; \mathbf{W}^{(t)}\big) + \eta\, \mathbb{E}_\tau^{(t)}\big[\, R^{k^\star} \alpha_l^{(t)}(j)\, \eta_l^{(t)}(j)\,\big]. \tag{33}$$

*2) (subgradient inequality for the max) Let the active competitor set be $\mathcal{A}_l^{\max} := \arg\max_{j \neq i_l} s_l\big(j; \mathbf{W}^{(t-1)}\big)$. Then*

$$\Delta_l\big(\mathbf{W}^{(t+1)}\big) \geq \Delta_l\big(\mathbf{W}^{(t)}\big) + \eta\Big(\mathbb{E}_\tau^{(t)}[R^{k^\star} \alpha_l^{(t)}(i_l)\, \eta_l^{(t)}(i_l)] - \sup_{j \neq i_l} \mathbb{E}_\tau^{(t)}[R^{k^\star} \alpha_l^{(t)}(j)\, \eta_l^{(t)}(j)]\Big) + \mathcal{O}(\eta^2). \tag{34}$$

*If, in a neighborhood of $\mathbf{W}^{(t-1)}$, the active maximizer is unique and remains unchanged (there exists $j_l^{\max} \in \mathcal{A}_l^{\max}$ that stays the maximizer in that neighborhood), then the first-order exact form is*

$$\Delta_l\big(\mathbf{W}^{(t+1)}\big) = \Delta_l\big(\mathbf{W}^{(t)}\big) + \eta\Big(\mathbb{E}_\tau^{(t)}[R^{k^\star} \alpha_l^{(t)}(i_l)\, \eta_l^{(t)}(i_l)] - \mathbb{E}_\tau^{(t)}[R^{k^\star} \alpha_l^{(t)}(j_l^{\max})\, \eta_l^{(t)}(j_l^{\max})]\Big) + \mathcal{O}(\eta^2). \tag{35}$$

*3) (update of a smooth lower bound) Define the competitors' log-sum-exp lower bound*

$$\phi_l(\mathbf{W}) := s_l(i_l; \mathbf{W}) - \log \sum_{j \neq i_l} \exp\big(s_l(j; \mathbf{W})\big) \leq \Delta_l(\mathbf{W}).$$

*Let $\tilde{\alpha}_l(j) := \frac{\exp(s_l(j))}{\sum_{q \neq i_l} \exp(s_l(q))}$ be the softmax normalized only over competitors, evaluated at $\mathbf{W}^{(t-1)}$. Then*

$$\phi_l\big(\mathbf{W}^{(t+1)}\big) = \phi_l\big(\mathbf{W}^{(t)}\big) + \eta\Big(\mathbb{E}_\tau^{(t)}[R^{k^\star} \alpha_l^{(t)}(i_l)\, \eta_l^{(t)}(i_l)] - \sum_{j \neq i_l} \tilde{\alpha}_l^{(t)}(j)\, \mathbb{E}_\tau^{(t)}[R^{k^\star} \alpha_l^{(t)}(j)\, \eta_l^{(t)}(j)]\Big) + \mathcal{O}(\eta^2). \tag{36}$$

*Moreover, to recompute intermediate quantities after one update, the following first-order relations (Fréchet differentials at $\mathbf{W}^{(t-1)}$) hold:*

$$\delta s_l^{(t)}(k) = s_l\big(k; \mathbf{W}^{(t+1)}\big) - s_l\big(k; \mathbf{W}^{(t)}\big) = \eta\, \mathbb{E}_\tau^{(t)}[R^{k^\star} \alpha_l^{(t)}(k)\, \eta_l^{(t)}(k)], \tag{37}$$

$$\delta \alpha_l^{(t)}(k) = \alpha_l^{(t)}(k)\Big(\delta s_l(k) - \sum_{q \in \mathcal{I}_l} \alpha_l^{(t)}(q)\, \delta s_l(q)\Big) + \mathcal{O}(\eta^2), \tag{38}$$

$$\delta \hat{\mathbf{p}}_{l+1}^{\mathrm{att},(t)} = \sum_{k \in \mathcal{I}_l} \delta \alpha_l^{(t)}(k)\, \mathbf{p}_k + \mathcal{O}(\eta^2), \tag{39}$$

$$\delta p_{\mathrm{vocab}}^{(t)}(r) = p_{\mathrm{vocab}}^{(t)}(r) \frac{\big\langle \delta \hat{\mathbf{p}}_{l+1}^{\mathrm{att},(t)},\, \mathbf{p}_r - \sum_q p_{\mathrm{vocab}}^{(t)}(q)\mathbf{p}_q \big\rangle}{\beta} + \mathcal{O}(\eta^2), \tag{40}$$

$$\delta \eta_l^{(t)}(k) = \frac{1}{\beta}\Big(-\big\langle \delta \hat{\mathbf{p}}_{l+1}^{\mathrm{att},(t)},\, \mathbf{p}_{i_l} - \sum_r p_{\mathrm{vocab}}^{(t)}(r)\mathbf{p}_r \big\rangle - \big\langle \mathbf{p}_k - \hat{\mathbf{p}}_{l+1}^{\mathrm{att},(t)},\, \sum_r \delta p_{\mathrm{vocab}}^{(t)}(r)\, \mathbf{p}_r \big\rangle\Big) + \mathcal{O}(\eta^2). \tag{41}$$

*In practice, with $n$ trajectories prompted by $\mathbf{x}^s \sim \mathrm{Unif}([K]^d), \forall s \in [n]$, Monte Carlo estimators can replace the expectations in equation 33–equation 36, e.g.,*

$$\widehat{\delta s}_l^{(t)}(k) = \eta\, \frac{1}{n} \sum_{s=1}^n R^{k^\star}(\mathrm{TF}(\mathbf{x}^s; \mathbf{W}^{(t)}))\, \alpha_l^{s,(t)}(k)\, \eta_l^{s,(t)}(k).$$

**Proof.** *(1) By Lemma 5 and Lemma 3, at $\mathbf{W}^{(t-1)}$,*

$$\nabla_{\mathbf{W}} \mathcal{J}_{\mathrm{REINFORCE}}^{k^\star} = \mathbb{E}_\tau\Big[R^{k^\star} \sum_{u=1}^{k^\star+1} \sum_{k \in \mathcal{I}_u} \alpha_u(k)\, \eta_u(k)\, \mathbf{p}_k \mathbf{p}_{c_u}^\top\Big].$$

*Using $\nabla_{\mathbf{W}} s_l(j) = \mathbf{p}_j \mathbf{p}_{c_l}^\top$ and the Frobenius inner product, a Taylor expansion up to second order along the explicit-Euler path $\mathbf{W}^{(t+1)} = \mathbf{W}^{(t)} + \eta\, \mathbf{G}_t$ (with fixed $\mathbf{G}_t := \nabla_{\mathbf{W}} \mathcal{J}_{\mathrm{REINFORCE}}^{k^\star}(\mathbf{W}^{(t)})$) gives*

$$s_l\big(j; \mathbf{W}^{(t+1)}\big) = s_l\big(j; \mathbf{W}^{(t)}\big) + \eta \left\langle \mathbf{p}_j \mathbf{p}_{c_l}^\top,\ \nabla_{\mathbf{W}} \mathcal{J}_{\mathrm{REINFORCE}}^{k^\star} \right\rangle\ +\ \mathcal{O}(\eta^2),$$

*and, more explicitly,*

$$s_l\big(j; \mathbf{W}^{(t)} + \eta\, \mathbf{G}_t\big) = s_l\big(j; \mathbf{W}^{(t)}\big) + \eta \left\langle \nabla_{\mathbf{W}} s_l(j),\ \mathbf{G}_t \right\rangle + \frac{\eta^2}{2} \left\langle \mathbf{G}_t,\ \nabla_{\mathbf{W}}^2 s_l(j)[\mathbf{G}_t] \right\rangle.$$

*Since $s_l(j; \mathbf{W}) = \langle \mathbf{p}_j \mathbf{p}_{c_l}^\top,\ \mathbf{W} \rangle + m_l(j)$ is affine in $\mathbf{W}$, its Hessian vanishes: $\nabla_{\mathbf{W}}^2 s_l(j) \equiv 0$. Therefore the second-order term above is identically zero under the explicit-Euler step (no dependence of the direction on $\eta$), and only the $\mathcal{O}(\eta^2)$ bookkeeping remains for uniformity with subsequent nonlinear propagations. and OBI guarantees $\langle \mathbf{p}_j \mathbf{p}_{c_l}^\top,\ \mathbf{p}_k \mathbf{p}_{c_u}^\top \rangle = 0$ unless $(k, u) = (j, l)$, which yields equation 33.*

*(2) Let $g_j := \eta\, \mathbb{E}_\tau[R^{k^\star}\, \alpha_l(j)\, \eta_l(j)]$ denote the first-order increment of $s_l(j)$. By Danskin's theorem / the subgradient inequality,*

$$\max_{j \neq i_l} \big(s_l(j) + g_j\big)\ \leq\ \max_{j \neq i_l} s_l(j)\ +\ \max_{j \neq i_l} g_j,$$

*and substituting the definitions gives the lower bound equation 34. If the active maximizer is unique and remains unchanged in a neighborhood, then the directional derivative of $\max_{j \neq i_l}$ is given by that $j_l^{\max}$, which yields equation 35.*

*(3) Let $\phi_l = s_l(i_l) - \log \sum_{j \neq i_l} e^{s_l(j)}$, so $\Delta_l \geq \phi_l$. Its directional derivative is*

$$\delta \phi_l = \delta s_l(i_l) - \sum_{j \neq i_l} \tilde{\alpha}_l(j)\, \delta s_l(j).$$

*Substituting equation 33 (with $t \to t + 1$) and collecting $\mathcal{O}(\eta^2)$ terms gives equation 36.*

*Finally, we detail the second-order sources for each intermediate quantity and then summarize them as $\mathcal{O}(\eta^2)$ terms:*

*(i) Softmax attention $\alpha_l$ to second order. Let $\mathbf{s}_l \in \mathbb{R}^{d_l}$ collect $s_l(\cdot)$ and write $\boldsymbol{\alpha}_l = \mathrm{softmax}(\mathbf{s}_l)$. For a perturbation $\delta \mathbf{s}_l = \mathcal{O}(\eta)$, the second-order expansion of component $j$ is*

$$\delta \alpha_l(j) = \sum_k J_{jk}^\alpha\, \delta s_l(k) + \frac{1}{2} \sum_{k,m} H_{j,km}^\alpha\, \delta s_l(k)\, \delta s_l(m) + \mathcal{O}(\eta^3),$$

*with Jacobian $J_{jk}^\alpha = \alpha_l(j)(\delta_{jk} - \alpha_l(k))$ and Hessian*

$$H_{j,km}^\alpha = \frac{\partial^2 \alpha_l(j)}{\partial s_l(k)\, \partial s_l(m)} = \alpha_l(j)\Big((\delta_{jm} - \alpha_l(m))(\delta_{jk} - \alpha_l(k))\ -\ \alpha_l(k)(\delta_{km} - \alpha_l(m))\Big).$$

*Because $\delta \mathbf{s}_l = \mathcal{O}(\eta)$ from equation 33, the quadratic term contributes $\mathcal{O}(\eta^2)$.*

*(ii) Attention-weighted token vector $\hat{\mathbf{p}}_{l+1}^{\mathrm{att}}$. Since $\hat{\mathbf{p}}_{l+1}^{\mathrm{att}} = \sum_j \alpha_l(j)\, \mathbf{p}_j$ is linear in $\boldsymbol{\alpha}_l$,*

$$\delta \hat{\mathbf{p}}_{l+1}^{\mathrm{att}} = \sum_j \delta \alpha_l(j)\, \mathbf{p}_j = \sum_{j,k} J_{jk}^\alpha\, \delta s_l(k)\, \mathbf{p}_j + \frac{1}{2} \sum_{j,k,m} H_{j,km}^\alpha\, \delta s_l(k)\, \delta s_l(m)\, \mathbf{p}_j + \mathcal{O}(\eta^3),$$

*so $\delta \hat{\mathbf{p}}_{l+1}^{\mathrm{att}} = \mathcal{O}(\eta) + \mathcal{O}(\eta^2)$.*

*(iii) Vocabulary softmax $p_{\mathrm{vocab}}$. Let $\ell_r = \langle \hat{\mathbf{p}}_{l+1}^{\mathrm{att}},\ \mathbf{p}_r \rangle / \beta$ and $\mathbf{p} = \mathrm{softmax}(\boldsymbol{\ell})$. Then*

$$\delta \ell_r = \frac{\langle \delta \hat{\mathbf{p}}_{l+1}^{\mathrm{att}},\ \mathbf{p}_r \rangle}{\beta}, \qquad \delta p(r) = \sum_m J_{rm}^p\, \delta \ell_m + \frac{1}{2} \sum_{m,n} H_{r,mn}^p\, \delta \ell_m\, \delta \ell_n + \mathcal{O}(\eta^3),$$

*where $J_{rm}^p = p(r)(\delta_{rm} - p(m))$ and*

$$H_{r,mn}^p = \frac{\partial^2 p(r)}{\partial \ell_m\, \partial \ell_n} = p(r)\Big((\delta_{rn} - p(n))(\delta_{rm} - p(m))\ -\ p(m)(\delta_{mn} - p(n))\Big).$$

Since $\delta\hat{\mathbf{p}}_{l+1}^{\text{att}} = \mathcal{O}(\eta) + \mathcal{O}(\eta^2)$, we have $\delta\ell = \mathcal{O}(\eta) + \mathcal{O}(\eta^2)$ and thus $\delta p = \mathcal{O}(\eta) + \mathcal{O}(\eta^2)$.

(iv) The scalar $\eta_l(k)$. Write

$$\eta_l(k) = \frac{1}{\beta}\Big\langle \underbrace{\mathbf{p}_k - \hat{\mathbf{p}}_{l+1}^{\text{att}}}_{:=\,\mathbf{u}}, \; \underbrace{\mathbf{p}_{i_l} - \sum_r p(r)\,\mathbf{p}_r}_{:=\,\mathbf{v}} \Big\rangle.$$

Perturbing $(\mathbf{u}, \mathbf{v}) \mapsto (\mathbf{u} - \delta\hat{\mathbf{p}}_{l+1}^{\text{att}}, \; \mathbf{v} - \sum_r \delta p(r)\,\mathbf{p}_r)$ and expanding to second order yields

$$\delta\eta_l(k) = \tfrac{1}{\beta}\Big(-\langle\delta\hat{\mathbf{p}}_{l+1}^{\text{att}},\,\mathbf{v}\rangle - \langle\mathbf{u},\,\textstyle\sum_r \delta p(r)\,\mathbf{p}_r\rangle\Big) + \tfrac{1}{\beta}\Big(-\tfrac{1}{2}\langle\delta^2\hat{\mathbf{p}}_{l+1}^{\text{att}},\,\mathbf{v}\rangle - \langle\delta\hat{\mathbf{p}}_{l+1}^{\text{att}},\,\textstyle\sum_r \delta p(r)\,\mathbf{p}_r\rangle - \tfrac{1}{2}\langle\mathbf{u},\,\textstyle\sum_r \delta^2 p(r)\,\mathbf{p}_r\rangle\Big) + \mathcal{O}(\eta^3),$$

where $\delta^2\hat{\mathbf{p}}_{l+1}^{\text{att}}$ and $\delta^2 p(r)$ collect the quadratic terms shown in (ii) and (iii). Since $\delta\hat{\mathbf{p}}_{l+1}^{\text{att}} = \mathcal{O}(\eta)$ and $\delta p = \mathcal{O}(\eta)$ at leading order, all bracketed second-line contributions are $\mathcal{O}(\eta^2)$.

Collecting these, we obtain the one-step relations stated in equation 37–equation 41, with every omitted higher-order contribution explicitly accounted for by the quadratic (Hessian) terms above and summarized as $\mathcal{O}(\eta^2)$ due to the small step size. $\qquad\square$

**Lemma 7** (Stepwise expected gradients at iteration $t$ with spurious success). *Fix iteration $t$ and a step $l \in \{1, \ldots, k^\star + 1\}$. For any index block $j \in \mathcal{I}_l$, write the block-projected expected REINFORCE gradient at $\mathbf{W}^{(t)}$ as*

$$G_{l,j}^{(t)} := \Big\langle \nabla_{\mathbf{W}}\mathcal{J}_{\text{REINFORCE}}^{k^\star}(\mathbf{W}^{(t)}),\, \mathbf{p}_j\mathbf{p}_{c_l}^\top \Big\rangle = \mathbb{E}^{(t)}\big[R^{k^\star}\,\alpha_l^{(t)}(j)\,\eta_l^{(t)}(j)\big],$$

*where $\mathbb{E}^{(t)}[\cdot]$ denotes the expectation over $\mathbf{x} \sim \mathcal{P}_x$ and $\tau \sim p_{\mathbf{W}^{(t)}}(\cdot \mid \mathbf{x})$, and $\alpha_l^{(t)}(\cdot), \eta_l^{(t)}(\cdot)$ are computed at $\mathbf{W}^{(t)}$ (Lemma 3).*

*Let $C_{<l}$ be the event that the index prefix $(i_1, \ldots, i_{l-1})$ matches the target $(i_1^\star, \ldots, i_{l-1}^\star)$, and denote*

$$\begin{aligned}
p_{<l}^{(t)} &:= \mathbb{P}^{(t)}(C_{<l} = 1),\\
S_{l,j,\text{corr}}^{(t)} &:= \mathbb{E}^{(t)}\big[\alpha_l^{(t)}(j)\eta_l^{(t)}(j)\,\mathbf{1}\{i_l = i_l^\star\} \,\big|\, C_{<l} = 1\big],\\
S_{l,j,\text{wrong}}^{(t)} &:= \mathbb{E}^{(t)}\big[\alpha_l^{(t)}(j)\eta_l^{(t)}(j)\,\mathbf{1}\{i_l \neq i_l^\star\} \,\big|\, C_{<l} = 1\big],\\
\bar{S}_{l,j}^{(t)} &:= \mathbb{E}^{(t)}\big[\alpha_l^{(t)}(j)\eta_l^{(t)}(j) \,\big|\, C_{<l} = 0\big].
\end{aligned} \tag{42}$$

*Then*

$$G_{l,j}^{(t)} = p_{<l}^{(t)}\Big(q_{l|\text{corr}}^{(t)} S_{l,j,\text{corr}}^{(t)} + q_{l|\text{wrong}}^{(t)} S_{l,j,\text{wrong}}^{(t)}\Big) + (1 - p_{<l}^{(t)})\, q_{<l|\text{wrong}}^{(t)}\, \bar{S}_{l,j}^{(t)}, \tag{43}$$

*where for the given reward $R^{k^\star}$ we define the conditional success probabilities:*

$$\begin{aligned}
q_{l|\text{corr}}^{(t)} &:= \mathbb{P}^{(t)}\big(R^{k^\star} = 1 \,\big|\, C_{<l} = 1,\, i_l = i_l^\star\big),\\
q_{l|\text{wrong}}^{(t)} &:= \mathbb{P}^{(t)}\big(R^{k^\star} = 1 \,\big|\, C_{<l} = 1,\, i_l \neq i_l^\star\big),\\
q_{<l|\text{wrong}}^{(t)} &:= \mathbb{P}^{(t)}\big(R^{k^\star} = 1 \,\big|\, C_{<l} = 0\big).
\end{aligned} \tag{44}$$

*(a) Final-answer reward $R^{k^\star} = R^{f_{S_\star}}$. Let $p_{\text{tail}}^{(t)}(l+1)$ denote the probability of completing the remaining true path from step $l + 1$ onward under $\mathbf{W}^{(t)}$, and let $\rho_{\text{spur},\geq l}^{(t)}$ be the spurious-success rate conditioned on having a wrong index at step $r \geq l$ (first deviation at $\geq l$), and $\rho_{\text{spur},<l}^{(t)}$ for having deviated before step $l$. Then*

$$q_{l|\text{corr}}^{(t)} = p_{\text{tail}}^{(t)}(l+1) + \big(1 - p_{\text{tail}}^{(t)}(l+1)\big)\rho_{\text{spur},\geq l+1}^{(t)}, \tag{45}$$

$$q_{l|\text{wrong}}^{(t)} = \rho_{\text{spur},\geq l}^{(t)}, \qquad q_{<l|\text{wrong}}^{(t)} = \rho_{\text{spur},<l}^{(t)}. \tag{46}$$

*For the parity case with $\mathbf{x} \sim \text{Unif}\{0,1\}^d$ and XOR kernels, $\rho_{\text{spur},\geq l}^{(t)} = \rho_{\text{spur},<l}^{(t)} = 1/2$ for all $l$.*

*(b) Subtask-family reward $R^{k^\star} = R^{\mathcal{F}_{S_\star}}$. Let $T^{(t)} \in \{1, \ldots, k^\star + 1\}$ be the (random) termination step (the first step where EOS is sampled). For each depth $r \in [k^\star]$, define the event*

$$U_r := \{T^{(t)} = r + 1 \text{ and the pre-EOS token equals } \boldsymbol{\mu}^{f_{S_\star^\tau}}(\mathbf{x})\}.$$

*Then $R^{\mathcal{F}_{S_\star}} = \mathbf{1}\{\cup_{r=1}^{k^\star} U_r\}$, and the stepwise expected gradient admits the depth-wise expansion*

$$G_{l,j}^{(t)} = \sum_{r=l}^{k^\star} \mathbb{E}^{(t)}\big[\mathbf{1}\{U_r\} \alpha_l^{(t)}(j) \eta_l^{(t)}(j)\big]. \tag{47}$$

*Consequently, for every $l \leq k^\star$,*

$$G_{l,j}^{(t)}\big|_{R=R^{\mathcal{F}_{S_\star}}} = G_{l,j}^{(t)}\big|_{R=R^{f_{S_\star}}} + \sum_{r=l}^{k^\star-1} \mathbb{E}^{(t)}\big[\mathbf{1}\{U_r\} \alpha_l^{(t)}(j) \eta_l^{(t)}(j)\big] \geq G_{l,j}^{(t)}\big|_{R=R^{f_{S_\star}}},$$

*which shows the subtask-family reward strictly adds nonnegative depth-wise contributions on shared prefixes.*

***Explicit per-depth success probabilities for (b).*** *For $s \in [k^\star]$, define the per-step true-path continuation probabilities and their products*

$$\pi_s^{(t)} := \mathbb{P}^{(t)}\big(i_s = i_s^\star \mid C_{<s} = 1\big), \qquad \Pi_{a:b}^{(t)} := \prod_{s=a}^{b} \pi_s^{(t)} \ \ (\Pi_{a:b}^{(t)} \equiv 1 \text{ if } a > b).$$

*Define the path events*

$$A_r := \{C_{<r} = 1, \ i_r = i_r^\star\}, \qquad B_r := \{C_{<r} = 1, \ i_r \neq i_r^\star\}, \qquad D_r := \{C_{<r} = 0\}.$$

*For each $r \geq l$, define branch-specific one-step termination probabilities at depth $r+1$:*

$$\theta_{r+1,A}^{(t)} := \mathbb{P}^{(t)}(i_{r+1} = \text{EOS} \mid A_r), \quad \theta_{r+1,B}^{(t)} := \mathbb{P}^{(t)}(i_{r+1} = \text{EOS} \mid B_r), \quad \theta_{r+1,D}^{(t)} := \mathbb{P}^{(t)}(i_{r+1} = \text{EOS} \mid D_r),$$

*and subtask-match probabilities at depth $r$ under the three branches (task dependent through $\{\Phi_l\}$):*

$$\rho_{r|A}^{(t)} := \mathbb{P}^{(t)}(\text{subtask match at } r \mid A_r), \ \ \rho_{r|B}^{(t)} := \mathbb{P}^{(t)}(\text{subtask match at } r \mid B_r), \ \ \rho_{r|D}^{(t)} := \mathbb{P}^{(t)}(\text{subtask match at } r \mid D_r).$$

*Then the event-level probabilities that enter equation 47 decompose explicitly as*

$$\mathbb{P}^{(t)}\big(U_r \mid C_{<l} = 1, \ i_l = i_l^\star\big)$$
$$= \underbrace{\rho_{r|A}^{(t)} \Pi_{l+1:r}^{(t)} \theta_{r+1,A}^{(t)}}_{\text{true-path to } r} + \underbrace{\rho_{r|B}^{(t)} \Pi_{l+1:r-1}^{(t)}(1 - \pi_r^{(t)}) \theta_{r+1,B}^{(t)}}_{\text{first deviation at } r} + \underbrace{\rho_{r|D}^{(t)} (1 - \Pi_{l+1:r-1}^{(t)}) \theta_{r+1,D}^{(t)}}_{\text{deviation before } r},$$

$$\mathbb{P}^{(t)}\big(U_r \mid C_{<l} = 1, \ i_l \neq i_l^\star\big) \tag{48}$$
$$= \mathbf{1}\{r = l\} \rho_{r|B}^{(t)} \theta_{l+1,B}^{(t)} + \mathbf{1}\{r > l\} \rho_{r|D}^{(t)} \theta_{r+1,D}^{(t)},$$

$$\mathbb{P}^{(t)}\big(U_r \mid C_{<l} = 0\big)$$
$$= \rho_{r|D}^{(t)} \theta_{r+1,D}^{(t)}.$$

*Consequently, identifying with the conditional success probabilities in Eq. equation 43 for $R = R^{\mathcal{F}_{S_\star}}$, the per-step success factors equal*

$$q_{l|\text{corr}}^{(t)} = \sum_{r=l}^{k^\star} \mathbb{P}^{(t)}\big(U_r \mid C_{<l} = 1, \ i_l = i_l^\star\big),$$

$$q_{l|\text{wrong}}^{(t)} = \sum_{r=l}^{k^\star} \mathbb{P}^{(t)}\big(U_r \mid C_{<l} = 1, \ i_l \neq i_l^\star\big) = \rho_{l|B}^{(t)} \theta_{l+1,B}^{(t)} + \sum_{r=l+1}^{k^\star} \rho_{r|D}^{(t)} \theta_{r+1,D}^{(t)}, \tag{49}$$

$$q_{<l|\text{wrong}}^{(t)} = \sum_{r=l}^{k^\star} \mathbb{P}^{(t)}\big(U_r \mid C_{<l} = 0\big) = \sum_{r=l}^{k^\star} \rho_{r|D}^{(t)} \theta_{r+1,D}^{(t)}.$$

In the parity case, $\rho_{r|A}^{(t)} \equiv 1$ and $\rho_{r|B}^{(t)} = \rho_{r|D}^{(t)} \equiv 1/2$, yielding the explicit decomposition Eq. equation 50 and, under uniform 2S-ART, the closed form Eq. equation 53.

**Parity specialization** Under the XOR kernel for subtasks with $\mathbf{x} \sim \text{Unif}\{0,1\}^d$, the per-branch subtask-match probabilities satisfy $\rho_{r|A}^{(t)} \equiv 1$ and $\rho_{r|B}^{(t)} = \rho_{r|D}^{(t)} \equiv \frac{1}{2}$. Writing the one-step termination probability at $r+1$ under a condition $E \in \{A_r, B_r, D_r\}$ as

$$\theta_{r+1}^{(t)}(E) := \mathbb{P}^{(t)}(i_{r+1} = \text{EOS} \mid E),$$

the probability of the event $U_r$ decomposes as

$$\mathbb{P}^{(t)}(U_r) = \underbrace{\mathbb{P}^{(t)}(A_r)\,\theta_{r+1}^{(t)}(A_r)}_{\text{true-path contribution}} + \underbrace{\tfrac{1}{2}\,\mathbb{P}^{(t)}(B_r)\,\theta_{r+1}^{(t)}(B_r) + \tfrac{1}{2}\,\mathbb{P}^{(t)}(D_r)\,\theta_{r+1}^{(t)}(D_r)}_{\text{spurious (reward-hacking) contribution}}. \tag{50}$$

Moreover, writing $p_{\text{path}}^{(t)}(r) := \mathbb{P}^{(t)}(A_r)$ and $p_{\text{dev},\geq r}^{(t)} := \mathbb{P}^{(t)}(B_r)$, $p_{\text{dev},<r}^{(t)} := \mathbb{P}^{(t)}(D_r)$, we have bounds

$$\mathbb{P}^{(t)}(U_r) \geq p_{\text{path}}^{(t)}(r)\,\inf\theta_{r+1}^{(t)}(A_r), \quad \mathbb{P}^{(t)}(U_r) \leq p_{\text{path}}^{(t)}(r)\,\sup\theta_{r+1}^{(t)}(A_r) + \tfrac{1}{2}\big(p_{\text{dev},\geq r}^{(t)}\,\sup\theta_{r+1}^{(t)}(B_r) + p_{\text{dev},<r}^{(t)}\,\sup\theta_{r+1}^{(t)}(D_r)\big). \tag{51}$$

If, in addition, the EOS policy at step $r+1$ is conditionally independent of the path branch (same marginal $\bar{\theta}_{r+1}^{(t)}$),

$$\mathbb{P}^{(t)}(U_r) = \bar{\theta}_{r+1}^{(t)}\Big(p_{\text{path}}^{(t)}(r) + \tfrac{1}{2}\big(p_{\text{dev},\geq r}^{(t)} + p_{\text{dev},<r}^{(t)}\big)\Big). \tag{52}$$

Under the base uniform 2S-ART for parity, namely for $s \in [k^\star]$ and $r \in [k^\star]$,

$$\pi_s^{(t)} = \frac{1}{d-s+1}, \qquad \theta_{r+1,A}^{(t)} = \theta_{r+1,B}^{(t)} = \theta_{r+1,D}^{(t)} = \frac{1}{d-r+1},$$

define

$$P_{r-1} := \prod_{s=1}^{r-1}\frac{1}{d-s+1} = \frac{1}{d(d-1)\cdots(d-r+2)} \quad (\text{and } P_0 := 1).$$

Then for every $r \in [k^\star]$,

$$\mathbb{P}^{(t)}(U_r) = \frac{P_{r-1}}{(d-r+1)^2} + \frac{1}{2}\frac{P_{r-1}(d-r)}{(d-r+1)^2} + \frac{1}{2}\frac{1-P_{r-1}}{d-r+1} = \frac{P_{r-1}(d-r+2)}{2(d-r+1)^2} + \frac{1-P_{r-1}}{2(d-r+1)}. \tag{53}$$

**Proof.** Starting from Lemma 5 and Lemma 3, for any $l, j$ we have

$$G_{l,j}^{(t)} = \mathbb{E}^{(t)}\big[R^{k^\star}\alpha_l^{(t)}(j)\,\eta_l^{(t)}(j)\big].$$

Condition on $C_{<l}$ and $i_l$ (tower property):

$$\begin{aligned}
G_{l,j}^{(t)} &= \mathbb{E}^{(t)}\Big[\mathbb{E}^{(t)}\big[R^{k^\star}\alpha_l^{(t)}(j)\,\eta_l^{(t)}(j)\mid C_{<l},\,i_l\big]\Big]\\
&= p_{<l}^{(t)}\,\mathbb{E}^{(t)}\Big[\alpha_l^{(t)}(j)\,\eta_l^{(t)}(j)\,\mathbb{E}^{(t)}\big[R^{k^\star}\mid C_{<l}=1,\,i_l\big]\mid C_{<l}=1\Big]\\
&\quad + (1-p_{<l}^{(t)})\,\mathbb{E}^{(t)}\Big[\alpha_l^{(t)}(j)\,\eta_l^{(t)}(j)\,\mathbb{E}^{(t)}\big[R^{k^\star}\mid C_{<l}=0\big]\mid C_{<l}=0\Big].
\end{aligned}$$

This gives equation 43 once we identify $q_{l|\cdot}^{(t)}$ by whether $i_l = i_l^\star$.

For (a), $R^{k^\star} = R^{f_{S\star}}$: on the true path through step $l$ with $i_l = i_l^\star$, success thereafter requires completing the remainder true path; otherwise, success can occur spuriously, which yields the formulas for $q_{l|\text{corr}}^{(t)}$, $q_{l|\text{wrong}}^{(t)}$ and $q_{<l|\text{wrong}}^{(t)}$.

For (b), $R^{k^\star} = R^{\mathcal{F}_{S\star}}$: success occurs at the unique termination depth $r$ where the pre-EOS token equals the subtask token; hence $R^{\mathcal{F}_{S\star}} = \sum_{r=1}^{k^\star}\mathbf{1}\{U_r\}$. Expanding $\mathbf{1}\{\cup_r U_r\}$ as $\sum_r \mathbf{1}\{U_r\}$, exchanging sum and expectation, and applying the same conditioning as above yields

$$G_{l,j}^{(t)} = \sum_{r=l}^{k^\star}\mathbb{E}^{(t)}\big[\mathbf{1}\{U_r\}\,\alpha_l^{(t)}(j)\,\eta_l^{(t)}(j)\big],$$

which implies $G_{l,j}^{(t)}|_{R=R^{\mathcal{F}_{S_\star}}} \geq G_{l,j}^{(t)}|_{R=R^{f_{S_\star}}}$ *via the nonnegative extra depths* $r \in \{l, \ldots, k^\star - 1\}$. *In the parity case, producing the correct subtask token at depth* $r$ *deterministically fixes* $z_r$; *any immediate-EOS strategy yields the subtask reward at* $r$, *while spurious subtask matches contribute via the corresponding conditional probabilities.* $\square$

**Lemma 8** (Finite-sample variance of REINFORCE objective and gradient (general post-training)). *Let* $(\mathbf{x}^{(s)}, \tau^{(s)})_{s=1}^n$ *be i.i.d., where* $\mathbf{x}^{(s)} \sim \mathcal{P}_x$ *and* $\tau^{(s)} \sim p_{\mathbf{W}^{k^\star}}(\cdot \mid \mathbf{x}^{(s)})$. *Define the Monte Carlo estimators*

$$\widehat{\mathcal{J}}_n^{k^\star}(\mathbf{W}^{k^\star}) := \frac{1}{n} \sum_{s=1}^n R^{k^\star}\big(\boldsymbol{\mu}^{z_{k^\star}}(\tau_{-2}^{(s)})\big),$$

$$\widehat{\mathbf{g}}_n^{k^\star}(\mathbf{W}^{k^\star}) := \frac{1}{n} \sum_{s=1}^n R^{k^\star}(\hat{\mathbf{p}}^{z_{k^\star}}(\tau_{-2}^{(s)})) \sum_{l=1}^{k^\star+1} \nabla_{\mathbf{W}^{k^\star}} \log \pi_{\mathbf{W}^{k^\star}}\big(i_l^{(s)} \mid \mathbf{x}^{(s)}, \hat{\mathbf{p}}^{z_{1:l}}\big).$$

*Assume* $R^{k^\star} \in \{0, 1\}$, *the regularity conditions of Lemma 5, and Orthogonal Block Isolation (Lemma 3). Let*

$$p_{\mathrm{succ}}(\mathbf{W}^{k^\star}) := \mathbb{E}_{\mathbf{x}\sim\mathcal{P}_x}\Big[\mathbb{P}_{\tau\sim p_{\mathbf{W}^{k^\star}}(\cdot|\mathbf{x})}\big\{R^{k^\star}(\hat{\mathbf{p}}^{z_{k^\star}}(\tau_{-2})) = 1\big\}\Big] = \mathbb{E}_{\mathbf{x},\tau}\big[R^{k^\star}(\hat{\mathbf{p}}^{z_{k^\star}}(\tau_{-2}))\big].$$

*Then:*

*1) (Objective) Unbiasedness and variance:*

$$\mathbb{E}\big[\widehat{\mathcal{J}}_n^{k^\star}\big] = \mathcal{J}_{\mathrm{REINFORCE}}^{k^\star}, \qquad \mathrm{Var}\big[\widehat{\mathcal{J}}_n^{k^\star}\big] = \frac{1}{n} p_{\mathrm{succ}}(1 - p_{\mathrm{succ}}) \leq \frac{1}{4n}. \tag{54}$$

*In particular, under near-uniform initialization and using equation 31, we have*

$$p_{\mathrm{succ}} \leq \rho_{\mathrm{spur}} + (1 - \rho_{\mathrm{spur}}) \prod_{l=1}^{k^\star+1} \frac{1}{d_l} \qquad (if\ d_l = \Theta(d),\ p_{\mathrm{succ}} \leq \rho_{\mathrm{spur}} + (1 - \rho_{\mathrm{spur}}) \Theta(d^{-(k^\star+1)})), \tag{55}$$

*so that*

$$\mathrm{Var}\big[\widehat{\mathcal{J}}_n^{k^\star}\big] \leq \frac{1}{n} \Big(\rho_{\mathrm{spur}} + (1 - \rho_{\mathrm{spur}}) d^{-(k^\star+1)}\Big)\Big(1 - \rho_{\mathrm{spur}} - (1 - \rho_{\mathrm{spur}}) d^{-(k^\star+1)}\Big) \leq \frac{1}{4n}. \tag{56}$$

*2) (Gradient) Unbiasedness and covariance. Let* $\boldsymbol{\mu} := \nabla_{\mathbf{W}^{k^\star}} \mathcal{J}_{\mathrm{REINFORCE}}^{k^\star}$ *be the population gradient. Then*

$$\mathbb{E}\big[\widehat{\mathbf{g}}_n^{k^\star}\big] = \boldsymbol{\mu}, \qquad \mathrm{Cov}\big[\widehat{\mathbf{g}}_n^{k^\star}\big] = \frac{1}{n} \Big( \underbrace{\mathbb{E}_{\mathbf{x},\tau}\Big[R^{k^\star} \sum_{l,t=1}^{k^\star+1} \nabla \log \pi_l \otimes \nabla \log \pi_t\Big] - \boldsymbol{\mu} \otimes \boldsymbol{\mu}}_{=:\,\boldsymbol{\Sigma}_{\mathrm{pop}}} \Big), \tag{57}$$

*where* $\pi_l$ *abbreviates* $\pi_{\mathbf{W}^{k^\star}}(i_l \mid \mathbf{x}, \hat{\mathbf{p}}^{z_{1:l}})$ *and* $\otimes$ *is the outer product in parameter space.*

*Furthermore, using Lemma 3, write a block-orthogonal expansion*

$$\nabla \log \pi_l = \sum_{k \in \mathcal{I}_l} \alpha_l^{k^\star}(k)\, \eta_l^{k^\star}(k)\, \mathbf{B}_{l,k}, \qquad \mathbf{B}_{l,k} := \mathbf{p}_k \mathbf{p}_{c_l}^\top.$$

*Denote the block coefficient* $C_{l,k} := R^{k^\star} \alpha_l^{k^\star}(k) \eta_l^{k^\star}(k)$. *Then the scalar variance in each orthogonal block is*

$$\mathrm{Var}\Big[\big\langle \widehat{\mathbf{g}}_n^{k^\star}, \frac{\mathbf{B}_{l,k}}{\|\mathbf{B}_{l,k}\|_F}\big\rangle\Big] = \frac{1}{n}\Big(\mathbb{E}[C_{l,k}^2] - \mathbb{E}[C_{l,k}]^2\Big), \tag{58}$$

*and the Frobenius-mean-square error (variance) satisfies*

$$\mathbb{E}\big\|\widehat{\mathbf{g}}_n^{k^\star} - \boldsymbol{\mu}\big\|_F^2 = \frac{1}{n}\Big( \underbrace{\mathbb{E}\big[R^{k^\star} \sum_{l=1}^{k^\star+1} \sum_{k \in \mathcal{I}_l} (\alpha_l^{k^\star}(k))^2 (\eta_l^{k^\star}(k))^2 \|\mathbf{B}_{l,k}\|_F^2\big] - \|\boldsymbol{\mu}\|_F^2}_{=:\,\Xi} \Big). \tag{59}$$

*Assume bounded embeddings and temperature $\beta > 0$ so that $|\eta_l^{k^\star}(k)| \leq 4/\beta$, and define the concentration factor*

$$S_l := \mathbb{E}_{\mathbf{x},\tau}\Big[\sum_{k \in \mathcal{I}_l^{\text{legal}}}(\alpha_l^{k^\star}(k))^2\Big] \in \Big[\frac{1}{d_l}, 1\Big].$$

*Let $C_{\text{pos}} := \max_{l,k}\|\mathbf{p}_k\|_2^2\,\|\mathbf{p}_{c_l}\|_2^2 = 1$. Then*

$$\Xi \leq \mathbb{E}[R^{k^\star}]\sum_{l=1}^{k^\star+1}\Big(\frac{16}{\beta^2}C_{\text{pos}}\Big)\mathbb{E}\Big[\sum_{k \in \mathcal{I}_l^{\text{legal}}}(\alpha_l^{k^\star}(k))^2\Big] \tag{60}$$

$$= \frac{16}{\beta^2}\,p_{\text{succ}}\sum_{l=1}^{k^\star+1}S_l. \tag{61}$$

*Consequently,*

$$\mathbb{E}\big\|\widehat{\mathbf{g}}_n^{k^\star} - \boldsymbol{\mu}\big\|_F^2 \leq \frac{1}{n}\frac{16}{\beta^2}\,p_{\text{succ}}\sum_{l=1}^{k^\star+1}S_l. \tag{62}$$

*Two useful specializations:*

- *(Near-uniform init) $S_l = \Theta(d^{-1})$,it holds that $\mathbb{E}\big\|\widehat{\mathbf{g}}_n^{k^\star} - \boldsymbol{\mu}\big\|_F^2 \leq \frac{1}{n}\frac{16k^\star+1}{\beta^2 d}\,p_{\text{succ}}$.*

- *(Worst-case spiky attention) $S_l \leq 1$, yielding $\mathbb{E}\|\widehat{\mathbf{g}}_n^{k^\star} - \boldsymbol{\mu}\|_F^2 \leq \frac{1}{n}\frac{16}{\beta^2}\,p_{\text{succ}}\,(k^\star + 1)$, which holds for the entire post-training.*

*Accordingly, the signal-to-noise ratio satisfies the general lower bound*

$$\text{SNR} := \frac{\|\boldsymbol{\mu}\|_F^2}{\mathbb{E}\|\widehat{\mathbf{g}}_n^{k^\star} - \boldsymbol{\mu}\|_F^2} \geq \frac{n\,\|\boldsymbol{\mu}\|_F^2}{(16/\beta^2)\,p_{\text{succ}}\sum_l S_l},$$

*and reduces to the near-uniform initialization scaling when $S_l \approx 1/d_l$.*

**Proof.** *We proceed in parts.*

*(A) Objective: unbiasedness and variance. Define the single-sample random variable*

$$Y := R^{k^\star}\big(\boldsymbol{\mu}^{z_{k^\star}}(\tau_{-2})\big) \in \{0, 1\}.$$

*By definition of $\mathcal{J}_{\text{REINFORCE}}^{k^\star}$ (Eq. (24)'s expectation target without the score factor),*

$$\mathbb{E}[Y] = p_{\text{succ}}(\mathbf{W}^{k^\star}) = \mathcal{J}_{\text{REINFORCE}}^{k^\star}.$$

*Hence $\widehat{\mathcal{J}}_n^{k^\star} = \frac{1}{n}\sum_{s=1}^n Y^{(s)}$ is unbiased. Since $(Y^{(s)})$ are i.i.d. Bernoulli($p_{\text{succ}}$),*

$$\text{Var}(\widehat{\mathcal{J}}_n^{k^\star}) = \frac{1}{n}\text{Var}(Y) = \frac{1}{n}\,p_{\text{succ}}(1 - p_{\text{succ}}) \leq \frac{1}{4n}.$$

*(B) Upper bound on $p_{\text{succ}}$ under 2S-ART near-uniformity. Suppose at each step $l$ there are $d_l \geq 1$ legal children and success requires selecting a unique correct legal child. Under 2S-ART, conditionally almost surely $\pi_l(\text{correct} \mid \cdot) \leq 1/d_l$. Therefore, for any fixed $(\mathbf{x}, \tau)$-measurable history,*

$$\mathbb{P}(Y = 1 \mid \text{history}) = \prod_{l=1}^{k^\star+1}\pi_l(\text{correct} \mid \cdot) \leq \prod_{l=1}^{k^\star+1}\frac{1}{d_l},$$

*and taking expectation over the history yields $p_{\text{succ}} \le \prod_l d_l^{-1}$. If $d_l \equiv d$, then $p_{\text{succ}} \le d^{-(k^\star+1)}$. Substituting into part (A) gives the displayed variance bound for $\widehat{\mathcal{J}}_n^{k^\star}$.*

*(C) Gradient: unbiasedness and covariance. Let the single-sample gradient random tensor be*

$$\mathbf{G} := R^{k^\star} \sum_{l=1}^{k^\star+1} \nabla_{\mathbf{W}^{k^\star}} \log \pi_{\mathbf{W}^{k^\star}}\big(i_l \mid \mathbf{x}, \hat{\mathbf{p}}^{z_{1:l}}\big).$$

*By Lemma 5, $\mathbb{E}[\mathbf{G}] = \nabla \mathcal{J}_{\text{REINFORCE}}^{k^\star} =: \boldsymbol{\mu}$, hence $\widehat{\mathbf{g}}_n^{k^\star} = \frac{1}{n}\sum_{s=1}^n \mathbf{G}^{(s)}$ is unbiased. With i.i.d. samples,*

$$\text{Cov}(\widehat{\mathbf{g}}_n^{k^\star}) = \frac{1}{n}\text{Cov}(\mathbf{G}) = \frac{1}{n}\Big(\mathbb{E}[\mathbf{G} \otimes \mathbf{G}] - \boldsymbol{\mu} \otimes \boldsymbol{\mu}\Big).$$

*Since $R^{k^\star} \in \{0,1\}$, $R^{k^\star 2} = R^{k^\star}$, giving*

$$\mathbb{E}[\mathbf{G} \otimes \mathbf{G}] = \mathbb{E}\Big[R^{k^\star} \sum_{l,t} \nabla \log \pi_l \otimes \nabla \log \pi_t\Big] =: \boldsymbol{\Sigma}_{\text{pop}}.$$

*This proves the displayed covariance formula.*

*(D) Block-wise variance via OBI. By Lemma 3, for each step $l$,*

$$\nabla \log \pi_l = \sum_{k \in \mathcal{I}_l} \alpha_l^{k^\star}(k)\, \eta_l^{k^\star}(k)\, \mathbf{B}_{l,k}, \quad \mathbf{B}_{l,k} := \mathbf{p}_k \mathbf{p}_{c_l}^\top, \quad \langle \mathbf{B}_{l,k}, \mathbf{B}_{l',k'}\rangle_F = 0\ ((l,k) \ne (l',k')).$$

*Hence*

$$\mathbf{G} = \sum_{l,k} C_{l,k}\, \mathbf{B}_{l,k}, \qquad C_{l,k} := R^{k^\star} \alpha_l^{k^\star}(k)\eta_l^{k^\star}(k).$$

*Projecting onto a unit-Frobenius block direction gives a scalar average of i.i.d. terms:*

$$\Big\langle \widehat{\mathbf{g}}_n^{k^\star},\, \frac{\mathbf{B}_{l,k}}{\|\mathbf{B}_{l,k}\|_F}\Big\rangle = \frac{1}{n}\sum_{s=1}^n C_{l,k}^{(s)}.$$

*Thus $\text{Var}(\langle \widehat{\mathbf{g}}_n^{k^\star}, \mathbf{B}_{l,k}/\|\mathbf{B}_{l,k}\|_F\rangle) = \frac{1}{n}(\mathbb{E}[C_{l,k}^2] - \mathbb{E}[C_{l,k}]^2)$.*

*(E) Frobenius MSE identity. Using orthogonality of blocks,*

$$\|\mathbf{G}\|_F^2 = R^{k^\star} \sum_{l,k} (\alpha_l^{k^\star}(k))^2 (\eta_l^{k^\star}(k))^2 \|\mathbf{B}_{l,k}\|_F^2.$$

*Since $\mathbb{E}\|\widehat{\mathbf{g}}_n^{k^\star} - \boldsymbol{\mu}\|_F^2 = \frac{1}{n}\big(\mathbb{E}\|\mathbf{G}\|_F^2 - \|\boldsymbol{\mu}\|_F^2\big)$ for i.i.d. averages, we obtain the displayed identity with*

$$\Xi := \mathbb{E}\big[R^{k^\star} \sum_{l=1}^{k^\star+1} \sum_{k \in \mathcal{I}_l} (\alpha_l^{k^\star}(k))^2 (\eta_l^{k^\star}(k))^2 \|\mathbf{B}_{l,k}\|_F^2\big].$$

*(F) Concrete upper bound for general post-training. Assume bounded embeddings and temperature $\beta > 0$ so that $|\eta_l^{k^\star}(k)| \le 4/\beta$. Define $S_l := \mathbb{E}[\sum_{k \in \mathcal{I}_l^{\text{legal}}} (\alpha_l^{k^\star}(k))^2] \in [1/d_l, 1]$. With $C_{\text{pos}} := \max_{l,k} \|\mathbf{B}_{l,k}\|_F^2$,*

$$\Big|\eta_l^{k^\star}(k)\Big| = \frac{\big|\langle \mathbf{p}_k - \hat{\mathbf{p}}^{\text{att}}, \mathbf{p}_{i_l} - \sum_r p_{\text{vocab}}(r)\mathbf{p}_r\rangle\big|}{\beta} \le \frac{\|\mathbf{p}_k - \hat{\mathbf{p}}^{\text{att}}\|_2 \|\mathbf{p}_{i_l} - \sum_r p_{\text{vocab}}(r)\mathbf{p}_r\|_2}{\beta} \le \frac{4}{\beta},$$

*where we used triangle inequality and that each difference of two convex combinations of bounded, pairwise-orthogonal embeddings has norm at most 2. Denoting $C_{\text{pos}} := \max_{l,k} \|\mathbf{B}_{l,k}\|_F^2 = \max_{l,k} \|\mathbf{p}_k\|_2^2 \|\mathbf{p}_{c_l}\|_2^2$, we bound*

$$\Xi \le \mathbb{E}[R^{k^\star}] \sum_{l=1}^{k^\star+1} \sum_{k \in \mathcal{I}_l^{\text{legal}}} \Big(\frac{4}{\beta}\Big)^2 C_{\text{pos}}(\alpha_l^{k^\star}(k))^2 = \frac{16}{\beta^2} C_{\text{pos}}\, p_{\text{succ}} \sum_{l=1}^{k^\star+1} S_l. \tag{63}$$

*Absorbing $c_\alpha^2$ into the constant (redefining the front coefficient) matches the displayed bound on $\Xi$. Finally, recall the exact identity*

$$\mathbb{E}\big\|\widehat{\mathbf{g}}_n^{k^\star} - \boldsymbol{\mu}\big\|_F^2 = \frac{1}{n}\Big(\Xi - \|\boldsymbol{\mu}\|_F^2\Big).$$

*By Jensen's inequality $\|\mathbb{E}[\mathbf{G}]\|_F^2 \le \mathbb{E}\|\mathbf{G}\|_F^2$, we have $\|\boldsymbol{\mu}\|_F^2 \le \Xi$, so $\Xi - \|\boldsymbol{\mu}\|_F^2 \le \Xi$. Therefore, for upper bounds it is valid to drop the negative term and use*

$$\mathbb{E}\big\|\widehat{\mathbf{g}}_n^{k^\star} - \boldsymbol{\mu}\big\|_F^2 \le \frac{1}{n}\Xi \le \frac{1}{n}\frac{16\,C_{\mathrm{pos}}}{\beta^2}\,p_{\mathrm{succ}}\sum_{l=1}^{k^\star+1}\frac{1}{d_l}.$$

*Remark on the SNR line. From the variance bound just proved,*

$$\mathbb{E}\big\|\widehat{\mathbf{g}}_n^{k^\star} - \boldsymbol{\mu}\big\|_F^2 \le \frac{C}{n}\,p_{\mathrm{succ}}\sum_{l=1}^{k^\star+1}\frac{1}{d_l}$$

*for a constant $C$ depending only on $(\beta, C_{\mathrm{pos}})$ and the near-uniformity constant. Therefore*

$$\mathrm{SNR} = \frac{\|\boldsymbol{\mu}\|_F^2}{\mathbb{E}\|\widehat{\mathbf{g}}_n^{k^\star} - \boldsymbol{\mu}\|_F^2} \ge \frac{n\,\|\boldsymbol{\mu}\|_F^2}{C\,p_{\mathrm{succ}}\sum_l d_l^{-1}}.$$

*If, in addition, one has a non-degeneracy bound $\|\boldsymbol{\mu}\|_F^2 \le C'\,p_{\mathrm{succ}}$ with $C'$ independent of depth (e.g., when the coefficient mass $\sum_{l,k}\mathbb{E}[|\alpha_l \eta_l|]$ remains $\mathcal{O}(\sum_l d_l^{-1})$), this yields the crude scaling*

$$\mathrm{SNR} = \mathcal{O}\Big(\frac{n\,p_{\mathrm{succ}}}{\sum_l d_l^{-1}}\Big),$$

*which reduces to $\mathcal{O}(n\,p_{\mathrm{succ}}\,d/(k^\star+1))$ for $d_l \equiv d$. The main message is that under 2S-ART, $p_{\mathrm{succ}}$ decays exponentially with depth, making SNR scale poorly with $k^\star$ unless $n$ grows accordingly.* $\qquad\square$

**Theorem 9** (Per-step REINFORCE sample size for block dominance). *Fix step $\ell$ and confidence $\delta \in (0,1)$. Let $|\mathcal{I}_\ell| = \Theta(d)$ and let $i_l^\star \in \mathcal{I}_\ell$ denote the unique correct child. For any $j \in \mathcal{I}_\ell$, given $n$ i.i.d. pairs $(\mathbf{x}^{(s)}, \tau^{(s)})$, define the empirical block mean*

$$\hat{\mu}_{\ell,j} := \frac{1}{n}\sum_{s=1}^n R^{k^\star}\alpha_\ell^{(s)}(j)\,\eta_\ell^{(s)}(j), \qquad R^{k^\star} \in \big\{\mathbf{R}_{\mathbf{x}}^{f_{S_\star}}(\cdot),\ \mathbf{R}_{\mathbf{x}}^{\mathcal{F}_{S_\star}}(\cdot)\big\}.$$

*Consider three scenario:*

(A) *No-curriculum ($R_{\mathbf{x}}^{f_{S_\star}}(\cdot)$ as oracle; Algorithm. 1).*
(B) *Depth-increasing Curriculum ($\mathbf{R}_{\mathbf{x}}^{\mathcal{F}_{S_\star}}(\cdot)$ as oracle; Algorithm. 2).*
(C) *Hint-decreasing Curriculum ($\mathbf{R}_{\mathbf{x}}^{\mathcal{F}_{S_\star}}(\cdot)$ as oracle; Algorithm. 3).*

*Then, for the target $f_{S_\star} \in \mathcal{F}_{\text{2S-ART}}$ with $|S_\star| = k^\star+1$, $S_\star = \{i_1^\star, i_2^\star, ..., i_{k^\star}^\star, d+1\}$, the sample complexity $n_\ell(\delta)$ to ensure with probability at least $1-\delta$, $\max_{j\in\mathcal{I}_\ell}|\hat{\mu}_{\ell,j} - \mathbb{E}[\hat{\mu}_{\ell,j}]| \le \Theta(\mu_{\ell,i_l^\star} - \max_{j\neq i_l^\star}\mu_{\ell,j})$ in case A − C is*

$$\text{(A)}\ \ n_\ell(\delta) \ge \tilde{\Omega}\Big(d^{2(k^\star+2-\ell)-2}\,(1 - \rho_{\mathrm{spur}}^{\mathrm{sup},>\ell})^{-2}\Big),$$

$$\text{(B)}\ \ n_\ell(\delta) \le \tilde{O}\Big(d^2\,(1 - \rho_{\mathrm{spur}}^{\mathrm{sup},\ell})^{-2}\Big), \tag{64}$$

$$\text{(C)}\ \ n_\ell(\delta) \le \tilde{O}\Big(d^2\,(1 - \rho_{\mathrm{spur}}^{\mathrm{sup},\,k^\star+1-\ell})^{-2}\Big),$$

*where $\tilde{\Omega}, \tilde{O}$ hide polylogarithmic factors in $d$ and $1/\delta$ and absolute constants depending on $\beta$.*

**Proof.** *We quantify, at a fixed step $\ell$, how many i.i.d. inputs $\mathbf{x}^{(s)} \sim \mathrm{Unif}([K]^d)$ suffice so that, with probability at least $1-\delta$, a one-step REINFORCE update increases the correct block more than any competitor by at least a fixed margin $C > 0$.*

***Distributional conventions and the REINFORCE gradient.*** *For each i.i.d. draw* $\mathbf{x}^{(s)} \sim \mathrm{Unif}([K]^d)$, *the base model* $\mathrm{TF}_{\mathrm{base}}$ *samples a trajectory* $\tau^{(s)} \sim p_{\mathbf{W}}(\cdot \mid \mathbf{x}^{(s)})$ *(independently across s), where* $\mathbf{W}$ *is the current parameter. We write*

$$\mathbb{E}[\cdot] := \mathbb{E}_{\mathbf{x} \sim \mathrm{Unif}([K]^d)} \, \mathbb{E}_{\tau \sim p_{\mathbf{W}}(\cdot \mid \mathbf{x})}[\cdot]$$

*for the population expectation. The outcome reward is*

$$R^{k^{\star}} \in \big\{ \mathbf{R}_{\mathbf{x}}^{f_{S_{\star}}}(\cdot), \, \mathbf{R}_{\mathbf{x}}^{\mathcal{F}_{S_{\star}}}(\cdot) \big\}, \qquad R^{k^{\star}} \in \{0, 1\}.$$

*By Lemma 5 and Lemma 3, the REINFORCE gradient admits the block decomposition*

$$\nabla_{\mathbf{W}} \mathcal{J}_{\mathrm{REINFORCE}} = \mathbb{E}\Big[ R^{k^{\star}} \sum_{t=1}^{k^{\star}+1} \sum_{k \in \mathcal{I}_t} \alpha_t(k) \, \eta_t(k) \, \mathbf{p}_k \mathbf{p}_{c_t}^{\top} \Big].$$

*Projecting onto the step-$\ell$ block* $\mathbf{B}_{\ell,j} := \mathbf{p}_j \mathbf{p}_{c_\ell}^{\top}$ *gives*

$$\underbrace{\big\langle \nabla_{\mathbf{W}} \mathcal{J}_{\mathrm{REINFORCE}}, \, \mathbf{B}_{\ell,j} \big\rangle}_{=: \; \mu_{\ell,j}} = \mathbb{E}\big[ R^{k^{\star}} \alpha_\ell(j) \, \eta_\ell(j) \big]. \tag{65}$$

***Setup and block-wise variables.*** *At step $\ell$, fix $j \in \mathcal{I}_\ell$ and define the single-sample block variable for $(\mathbf{x}, \tau)$:*

$$X_{\ell,j} := R^{k^{\star}} \alpha_\ell(j) \, \eta_\ell(j), \qquad R^{k^{\star}} \in \{0, 1\},$$

*where* $\alpha_\ell(j), \eta_\ell(j)$ *are computed from* $(\mathbf{x}, \tau)$ *and the current* $\mathbf{W}$. *By equation 65,*

$$\mu_{\ell,j} := \mathbb{E}[X_{\ell,j}] = \big\langle \nabla_{\mathbf{W}} \mathcal{J}_{\mathrm{REINFORCE}}, \, \mathbf{B}_{\ell,j} \big\rangle.$$

*Given $n$ i.i.d. pairs* $(\mathbf{x}^{(s)}, \tau^{(s)})$, *define the empirical estimator and its range bound*

$$\hat{\mu}_{\ell,j} := \tfrac{1}{n} \sum_{s=1}^{n} X_{\ell,j}^{(s)}, \qquad |X_{\ell,j}| \leq B = \Theta(\tfrac{4}{d\beta}),$$

*where* $|\eta_\ell(j)| \leq 4/\beta$ *(Lemma 3) and* $\alpha_\ell(j) = \Theta(d^{-1})$ *. Note $B$ is the Hoeffding range parameter; it is* independent *of the margin defined below.*

*Let $i_l^{\star}$ be the unique correct child at depth $\ell$. We say the empirical gradient exhibits* block dominance with margin $C$ *if*

$$\hat{\mu}_{\ell, i_l^{\star}} - \max_{j \in \mathcal{I}_\ell \setminus \{i_l^{\star}\}} \hat{\mu}_{\ell,j} \geq C.$$

*By Lemma 6, this implies the increase of $s_\ell(i_l^{\star})$ exceeds all competitors by at least $\eta \, C$ (up to $\mathcal{O}(\eta^2)$), hence strictly enlarging the step-$\ell$ attention gap.*

***Concentration.*** *Hoeffding's inequality for bounded variables (Lemma 1) yields for any fixed $j$ (expectation over the joint randomness of $(\mathbf{x}^{(s)}, \tau^{(s)})$):*

$$\Pr\big( |\hat{\mu}_{\ell,j} - \mu_{\ell,j}| \geq t \big) \leq 2 \exp\Big( -\tfrac{2nt^2}{B^2} \Big).$$

*A union bound over $|\mathcal{I}_\ell| = \Theta(d)$ blocks gives that, with probability $\geq 1 - \delta$,*

$$\max_{j \in \mathcal{I}_\ell} |\hat{\mu}_{\ell,j} - \mu_{\ell,j}| \leq t_n := \frac{B}{\sqrt{2n}} \sqrt{\log \tfrac{2d}{\delta}}. \tag{66}$$

*If the* population *margin*

$$\gamma_\ell := \mu_{\ell, i_l^{\star}} - \max_{j \neq i_l^{\star}} \mu_{\ell,j} > 0,$$

*then setting $t_n \leq \gamma_\ell/4$ guarantees dominance with margin $C = \gamma_\ell/2$.*

**Master relation and the choice of** $C$**.** *From equation 66, requiring empirical dominance with margin* $C \leq \gamma_\ell/2$ *is ensured by taking* $t_n \leq \gamma_\ell/2$*. Solving*

$$\frac{B}{\sqrt{2n}}\sqrt{\log \frac{2d}{\delta}} \leq \frac{\gamma_\ell}{2}$$

*for* $n$ *yields the explicit sample-size condition*

$$n \geq \frac{2B^2}{\gamma_\ell^2}\log\frac{2d}{\delta}.$$

*Throughout this proof we fix* $C := \gamma_\ell/2$ *so the target constant margin is explicit.*

**Population margin under three settings.** *Using Lemma 7 and copied-PART near-uniformity (*$\alpha_\ell(i_l^\star) = \Theta(1/d)$*), with expectations taken over* $\mathbf{x} \sim \mathrm{Unif}([K]^d)$ *and* $\tau \sim p_{\mathbf{W}}(\cdot \mid \mathbf{x})$*:*

(A) **No curriculum; terminal oracle** $\mathbf{R}_{\mathbf{x}}^{f_{S_\star}}(\cdot)$*. Conditioning on a correct prefix to* $\ell$*, completing the true suffix (including EOS) has probability* $\Theta\big(d^{-(k^\star+1-\ell)}\big)$*, while any deviation may be accepted with probability at most* $\rho_{\mathrm{spur}}^{\sup,>\ell}$*. Under copied-PART near-uniformity* $\alpha_\ell(i_l^\star) = \Theta(1/d)$ *and* $4/\beta \geq \eta_l(i_l^\star) \geq O(1/\beta) \geq \eta_l(j), j \neq i_l^\star$*, Lemma 7 implies*

$$q_{\ell|\mathrm{corr}} - q_{\ell|\mathrm{wrong}} = \Theta(p_{\mathrm{tail}}(\ell+1)\left(1 - \rho_{\mathrm{spur}}^{\sup,>\ell}\right)), \qquad p_{\mathrm{tail}}(\ell+1) = \Theta\big(d^{-(k^\star+1-\ell)}\big),$$

*Consequently, in case (A) we have the matching scaling over the expected margin*

$$\gamma_\ell^{\mathsf{A}} = \Theta\Big(\frac{d^{-(k^\star+2-\ell)}}{\beta}\left(1 - \rho_{\mathrm{spur}}^{\sup,>\ell}\right)\Big),$$

*up to absolute constants, which will be used to derive the* lower bound *on* $n_\ell(\delta)$*.*

(B) **Depth-increasing curriculum; family oracle** $\mathbf{R}_{\mathbf{x}}^{\mathcal{F}_{S_\star}}(\cdot)$ **with external truncation at depth** $\ell$*. Similar to Thm. 11,* $\alpha_{i_l^\star}^{(\ell)} \geq 1 - \eta_\ell = \Theta(d^{-1})$ *and any wrong child has acceptance at most* $\rho_{\mathrm{spur}}^{\sup,\ell}$*. Also, we see that* $4/\beta \geq \eta_l(i_l^\star) \geq O(1/\beta) \geq \eta_l(j), j \neq i_l^\star$*. Thus*

$$\gamma_\ell^{\mathsf{B}} = \Theta\Big(\frac{1}{\beta d^2}\left(1 - \rho_{\mathrm{spur}}^{\sup,\ell}\right)\Big).$$

(C) **Hint-decreasing curriculum (reverse indexing);** *identical to (B) with* $\ell \mapsto k^\star+1-\ell$*, such that* $\{\gamma_\ell^{\mathsf{C}}\}_{\ell=1}^{k^\star+1} = \{\gamma_{k^\star+1-\ell}^{\mathsf{B}}\}_{\ell=1}^{k^\star+1}$*.*

*From equation 66, taking* $t_n \leq \gamma_\ell/2$ *ensures block dominance with margin* $\gamma_\ell/2$*. Solving for* $n$ *with range bound* $B = 4/(\beta d)$ *gives the master condition* $n \geq (2B^2/\gamma_\ell^2)\log\frac{2d}{\delta}$*. For case (A), substituting the* upper bound *on* $\gamma_\ell$ *above yields the displayed* lower bound *on* $n_\ell$*. For cases (B) and (C), substituting the* lower bounds *on* $\gamma_\ell$ *yields the displayed* upper bounds *on* $n_\ell$*. Here* $B$ *is purely the bounded-range constant of* $X_{\ell,j}$ *(Hoeffding parameter);* $\gamma_\ell$ *is the population margin and does not influence the range. Under the near-uniformity and spurious-success cap assumptions used above, summarizing (A) as a* $\Theta(\cdot)$ *scaling for* $\gamma_\ell$ *is appropriate; we keep one-sided bounds when only sufficiency is needed for (B)/(C).* $\qquad\square$

**Theorem 10** (One-pass and online per-step schedules achieve per-step margin thresholds)**.** *For* $\forall \varepsilon > 0$*, consider the target per-step margin thresholds* $\{\Gamma_\ell\}_{\ell=1}^{k^\star+1}$ *for Lemma 4, with* $\varepsilon_l = \varepsilon/(k^\star + 1)$*, namely*

$$\Gamma_l = \log\Big(\frac{d-1}{(\frac{1}{2} + \frac{\beta}{2}\log(\frac{(K+1)((k^\star+1)-\varepsilon)}{\varepsilon}))^{-1} - 1}\Big) \leq \Theta(\log(d))$$

*For each step* $\ell$*, let* $n_\ell(\delta)$ *be the per-step sample size from Theorem 9 that guarantees, with probability at least* $1 - \delta$*,* $\max_{j \in \mathcal{I}_\ell}|\hat{\mu}_{\ell,j} - \mathbb{E}[\hat{\mu}_{\ell,j}]| \leq (\mu_{\ell,i_l^\star} - \max_{j \neq i_l^\star}\mu_{\ell,j})/4$*, guaranteeing that the empirical block-dominance margin to be no less than the thresholds* $\{\gamma_\ell^{\mathsf{A}}/2\}_{\ell=1}^{k^\star+1}, \{\gamma_\ell^{\mathsf{B}}/2\}_{\ell=1}^{k^\star+1}, \{\gamma_\ell^{\mathsf{C}}/2\}_{\ell=1}^{k^\star+1}$*.*

*There exist learning-rate schedules under which, with probability at least* $1 - \delta$*, the post-update per-step logit gaps satisfy* $\Delta_\ell \geq \Gamma_\ell$ *for all* $\ell \in \{1, \ldots, k^\star+1\}$*, so that Lemma 4 applies. We state them for three settings:*

*(A) No curriculum (Algorithm. 1); terminal oracle* $\mathbf{R}_{\mathbf{x}}^{f_{S_\star}}(\cdot)$*. Draw a single batch of size*

$$n_0 \geq \max_{\ell \in [k^\star+1]} n_\ell\big(\delta/(k^\star+1)\big) = \tilde{\Omega}\big(d^{2k^\star+2}(1 - \rho_{\mathrm{spur}}^{\sup,>1})^{-2}\big),$$

*where the last equality takes the worst step (the first decision), and the extra union over steps would not influence the lower bound. Consider perform a single gradient step (one-shot) with learning rate $\eta^{\mathsf{A}} = \Theta(\beta \log(d) d^{k^\star+1}(1 - \rho_{\mathrm{spur}}^{\sup,>1})^{-1}) \geq \Theta(\Gamma_\ell/(\gamma_1^{\mathsf{A}}/2))$. Guarantee: with probability at least $1 - \delta$, after this one update we have $\Delta_\ell \geq \Gamma_\ell$ for all $\ell$, including the final EOS step.*

*(B) Depth-increasing curriculum (Algorithm. 2); family oracle $\mathbf{R}_{\mathbf{x}}^{\mathcal{F}_{S_\star}}(\cdot)$ (external truncation). Sampling and updates are online per step: for $\ell = 1, 2, \ldots, k^\star+1$ do (i) draw $n_\ell = \tilde{\Theta}(d^2(1 - \rho_{\mathrm{spur}}^{\sup,\ell})^{-2})$ fresh samples; (ii) take one gradient step with learning rate $\eta_\ell^{\mathsf{B}} = \Omega(\beta \log(d) d^2 (1 - \rho_{\mathrm{spur}}^{\sup,\ell})^{-1}) \geq \Theta(\Gamma_l/(\gamma_\ell^{\mathsf{B}}/2))$. Guarantee: with probability at least $1 - \delta$, after $k^\star+1$ updates, $\Delta_\ell \geq \Gamma_\ell$ holds for all steps. The total sample size is $\sum_{\ell=1}^{k^\star+1} n_\ell = \tilde{\Theta}((k^\star + 1)d^2(1 - \rho_{\mathrm{spur}}^{\max})^{-2})$.*

*(C) Hint-decreasing curriculum (Algorithm. 3); family oracle $\mathbf{R}_{\mathbf{x}}^{\mathcal{F}_{S_\star}}(\cdot)$. Sampling and updates are online per step in reverse: for $\ell = k^\star+1, \ldots, 1$ do (i) draw $n_\ell = \tilde{\Theta}(d^2(1 - \rho_{\mathrm{spur}}^{\sup, k^\star+1-\ell})^{-2})$ fresh samples; (ii) take one gradient step with learning rate $\eta_\ell^{\mathsf{C}} = \Omega(\beta \log(d) d^2 (1 - \rho_{\mathrm{spur}}^{\sup, k^\star+1-\ell})^{-1}) \geq \Theta(\Gamma_l/(\gamma_\ell^{\mathsf{C}}/2))$. Guarantee: with probability at least $1 - \delta$, after $k^\star+1$ updates, $\Delta_\ell \geq \Gamma_\ell$ holds for all steps. The total sample size is $\sum_{\ell=1}^{k^\star+1} n_\ell = \tilde{O}((k^\star + 1)d^2(1 - \rho_{\mathrm{spur}}^{\max})^{-2})$.*

**Proof.** *The proof is direct based on OBI (Lemma 3), as well as the convergence conditions in Lemma 6 for a target $\varepsilon > 0$.* $\square$

---

**Algorithm 1** No-Curriculum REINFORCE Finetuning (A)

---

1: Inputs: input distribution $\mathbf{x} \sim \mathrm{Unif}([K]^d)$, error budget $\varepsilon > 0$, target length $k^\star$, terminal oracle $\mathbf{R}_{\mathbf{x}}^{f_{S_\star}}(\cdot)$, sample complexity $n_0 = \tilde{\Omega}(d^{2k^\star+4}(1 - \rho_{\mathrm{spur}}^{\sup,>1})^{-2})$, learning rate $\eta^{\mathsf{A}} = \Theta(\beta \log(d) d^{k^\star+1}(1 - \rho_{\mathrm{spur}}^{\sup,>1})^{-1})$.
2: Initialize transformer parameters $\mathbf{W}^{(0)}$.
3: Draw a single batch of $n_0$ i.i.d. samples $(\mathbf{x}^{(s)}, \tau^{(s)})_{s=1}^{n_0}$ where $\mathbf{x}^{(s)} \sim \mathrm{Unif}([K]^d)$ and $\tau^{(s)} \sim p_{\mathbf{W}^{(0)}}(\cdot \mid \mathbf{x}^{(s)})$.
4: Compute empirical REINFORCE gradient:
5: $\quad \widehat{\mathbf{g}} = \frac{1}{n_0} \sum_{s=1}^{n_0} \mathbf{R}_{\mathbf{x}^{(s)}}^{f_{S_\star}}(\hat{\mathbf{p}}^{z_{k^\star}}(\tau_{-2}^{(s)})) \sum_{\ell=1}^{k^\star+1} \nabla_{\mathbf{W}} \log \pi_{\mathbf{W}^{(0)}}(i_\ell^{(s)} \mid \mathbf{x}^{(s)}, \hat{\mathbf{p}}^{z_{1:\ell}})$.
6: Update parameters: $\mathbf{W}^{(1)} = \mathbf{W}^{(0)} + \eta^{\mathsf{A}} \cdot \widehat{\mathbf{g}}$.
7: Return $\mathbf{W}^{(1)}$.

---

**Algorithm 2** Depth-Increasing Curriculum REINFORCE Finetuning (B)

---

1: Inputs: input distribution $\mathbf{x} \sim \mathrm{Unif}([K]^d)$, error budget $\varepsilon > 0$, target length $k^\star$, family oracle $\mathbf{R}_{\mathbf{x}}^{\mathcal{F}_{S_\star}}(\cdot)$, per-step sample complexity $n_\ell = \tilde{\Theta}(d^2(1 - \rho_{\mathrm{spur}}^{\sup,\ell})^{-2})$, learning rates $\eta_\ell^{\mathsf{B}} = \Omega(\beta \log(d) d^2 (1 - \rho_{\mathrm{spur}}^{\sup,\ell})^{-1})$.
2: Initialize transformer parameters $\mathbf{W}^{(0)}$.
3: **for** $\ell = 1$ to $k^\star + 1$ **do**
4: $\quad$ Draw fresh samples $(\mathbf{x}^{(s)}, \tau^{(s)})_{s=1}^{n_\ell}$ where $\mathbf{x}^{(s)} \sim \mathrm{Unif}([K]^d)$ and $\tau^{(s)} \sim p_{\mathbf{W}^{(\ell-1)}}(\cdot \mid \mathbf{x}^{(s)})$.
5: $\quad$ If $\ell < k^\star + 1$, for each sample $s$, truncate trajectory at length $\ell + 1$:
6: $\quad\quad$ If $\tau^{(s)}$ contains EOS at position $\leq \ell$, discard sample.
7: $\quad\quad$ Otherwise, append EOS to create truncated sequence $\tau_{\mathrm{trunc}}^{(s)}$.
8: $\quad$ If $\ell = k^\star + 1$, $\tau_{\mathrm{trunc}}^{(s)} = \tau^{(s)}$.
9: $\quad$ Compute empirical REINFORCE gradient for step $\ell$:
10: $\quad\quad \widehat{\mathbf{g}}_\ell = \frac{1}{n_\ell} \sum_{s=1}^{n_\ell} \mathbf{R}_{\mathbf{x}^{(s)}}^{\mathcal{F}_{S_\star}}(\tau_{\mathrm{trunc}}^{(s)}) \nabla_{\mathbf{W}} \log \pi_{\mathbf{W}^{(\ell-1)}}(i_\ell^{(s)} \mid \mathbf{x}^{(s)}, \hat{\mathbf{p}}^{z_{1:\ell}})$.
11: $\quad$ Update parameters: $\mathbf{W}^{(\ell)} = \mathbf{W}^{(\ell-1)} + \eta_\ell^{\mathsf{B}} \cdot \widehat{\mathbf{g}}_\ell$.
12: **end for**
13: Return $\mathbf{W}^{(k^\star+1)}$.

---

**Proof.** *Proof of Theorem. 2. The proof follows by collaborating Theorem. 10 and Lemma 6.* $\square$

### G.3. Proof of Test-time Scaling

**Proposition 1** (Formal Version of the First Item in Thm. 3). *Assume* $\mathrm{TF}_{\mathrm{base}}$ *copies PART probability behavior with a unique correct child per depth. At depth $\ell$, consider FXRS (Alg. 4) to force a chosen visible $x$-token and BAI (Alg. 5) that, for each $j \in \mathcal{I}_\ell$, repeats FXRS+oracle $m$ times and selects the empirical best arm.*

*Then any $\delta$-correct identification of the ground-truth path using only the terminal oracle $\mathbf{R}_{\mathbf{x}}^{f_{S_\star}}$ under token-only observability*

---

**Algorithm 3** Hint-Decreasing Curriculum REINFORCE Finetuning (C)

---

1: Inputs: input distribution $\mathbf{x} \sim \text{Unif}([K]^d)$, error budget $\varepsilon > 0$, target length $k^\star$, family oracle $\mathbf{R}_{\mathbf{x}}^{\mathcal{F}_{S^\star}}(\cdot)$, per-step sample complexity $n_\ell = \tilde{\Theta}\big(d^2(1 - \rho_{\text{spur}}^{\sup, k^\star+1-\ell})^{-2}\big)$, learning rates $\eta_\ell^{\mathsf{C}} = \Omega(\beta \log(d)d^2(1 - \rho_{\text{spur}}^{\sup, k^\star+1-\ell})^{-1})$.
2: Initialize transformer parameters $\mathbf{W}^{(0)}$.
3: **for** $\ell = 1$ down to $k^\star + 1$ **do**
4:     Draw fresh samples $(\mathbf{x}^{(s)}, \tau^{(s)})_{s=1}^{n_\ell}$ where $\mathbf{x}^{(s)} \sim \text{Unif}([K]^d)$ and $\tau^{(s)} \sim p_{\mathbf{W}^{(k^\star+1-\ell)}}(\cdot \mid \mathbf{x}^{(s)})$.
5:     For each sample $s$, provide hint prefix of length $k^\star + 1 - \ell$:
6:         Let $\mathbf{h}^{(s)} = (i_1^\star, i_2^\star, \ldots, i_{k^\star+1-\ell}^\star)$ be the correct prefix.
7:         Generate remaining tokens $\tau_{\text{hint}}^{(s)}$ from $\mathbf{W}^{(k^\star+1-\ell)}$ conditioned on $\mathbf{h}^{(s)}$.
8:     Compute empirical REINFORCE gradient for step $\ell$:
9:     $\widehat{\mathbf{g}}_\ell = \frac{1}{n_\ell} \sum_{s=1}^{n_\ell} \mathbf{R}_{\mathbf{x}^{(s)}}^{\mathcal{F}_{S^\star}}(\tau_{\text{hint}}^{(s)}) \nabla_{\mathbf{W}} \log \pi_{\mathbf{W}^{(k^\star+1-\ell)}}(i_\ell^{(s)} \mid \mathbf{x}^{(s)}, \hat{\mathbf{p}}^{z_{1:\ell}})$.
10:    Update parameters: $\mathbf{W}^{(k^\star+2-\ell)} = \mathbf{W}^{(k^\star+1-\ell)} + \eta_\ell^{\mathsf{C}} \cdot \widehat{\mathbf{g}}_\ell$.
11: **end for**
12: Return $\mathbf{W}^{(k^\star+1)}$.

---

*requires*

$$T_{data} \geq \tilde{\Omega}\big(d^{2k^\star+1}(1 - \bar{\rho}_{spur})^{-2}\big),$$

$$T_{comp} \geq \tilde{\Omega}\big(d^{2k^\star+1}(1 - \bar{\rho}_{spur})^{-2}\big),$$

*where* $\bar{\rho}_{spur} := \sup_\ell \rho_{spur}^{sup,>\ell}$. *Here* $\rho_{spur}^{sup,>\ell} \in [0,1)$ *denotes the* terminal-oracle, suffix-level spurious-acceptance *parameter at depth* $\ell$; *formally, with* $\mathcal{H}_{\ell-1}$ *to denote the event that the history up to depth* $\ell-1$ *is correct,* $C_\ell(j) := \{$*at depth* $\ell$, *the visible token is forced to* $x_{i_\ell} = x_j\}$ *for the FXRS forcing event, define the* true-suffix *event*

$$E_{true} := \bigcap_{t=\ell+1}^{k^\star+1} \{J_t = j_t^\star\},$$

*where* $j_t^\star$ *is the unique correct child at depth* $t$ *given the correct history (with* $j_{k^\star+1}^\star = d+1$*). Let* $E_{spur}$ *be the complement of* $E_{true}$ *while still yielding acceptance by* $\mathbf{R}_{\mathbf{x}}^{f_{S^\star}}$ *on the final pre-EOS token. We then defines*

$$\rho_{spur}^{sup,>\ell} := \sup_{j \in \mathcal{I}_\ell} \mathbb{E}_{\mathbf{x}}\Big[ \Pr_{\text{TF}_{\text{base}}} \big(E_{spur} \mid \mathbf{x}, \mathcal{H}_{\ell-1}, C_\ell(j)\big) \Big],$$

*which upper-bounds, in expectation over* $\mathbf{x}$, *the probability that a* wrong suffix *(after depth* $\ell$*) is nonetheless accepted by* $\mathbf{R}_{\mathbf{x}}^{f_{S^\star}}$ *when the depth-*$\ell$ *visible token is forced to* $x_{i_\ell} = x_j$.

---

**Algorithm 4** Forced X-token Rejection Sampling (FXRS) under token-only observability

---

1: Inputs: depth $\ell$, committed history $\mathbf{CoT}_{\ell-1}$, legal set $\mathcal{I}_\ell$, candidate $j \in \mathcal{I}_\ell$, model $\text{TF}_{\text{base}}$, oracle $\mathbf{R}_{\mathbf{x}}^{f_{S^\star}}$, max trials $T_{\max} = \Theta(d \log(\frac{d}{\delta}))$.
2: **for** $t = 1$ to $T_{\max}$ **do**
3:     From context $\mathbf{CoT}_{\ell-1}$, sample one step with $\text{TF}_{\text{base}}$ to emit $\hat{\boldsymbol{\mu}}^{x_{i_\ell}}$.
4:     **if** $\hat{\boldsymbol{\mu}}^{x_{i_\ell}} = \boldsymbol{\mu}^{x_j}$ **then**
5:         Roll out the remaining suffix using $\text{TF}_{\text{base}}$ until termination (EOS occurs at some depth by the copied PART behavior).
6:         Return the full rollout sequence and **stop**.
7:     **end if**
8: **end for**
9: If no match after $T_{\max}$, return **fail** (increase $T_{\max}$ if needed).

---

---

**Algorithm 5** BAI across depths with FXRS and $\mathbf{R}_{\mathbf{x}}^{f_{S\star}}$

---

1: Inputs: legal sets $\{\mathcal{I}_\ell\}_{\ell=1}^{k^\star}$, model $\text{TF}_{\text{base}}$, oracle $\mathbf{R}_{\mathbf{x}}^{f_{S\star}}$, confidence $\delta$.
2: Initialize committed partial CoT $\mathcal{H} \leftarrow (\ )$.
3: **for** $\ell = 1$ to $k^\star + 1$ **do**
4:     Set per-arm repetitions $m \leftarrow \Theta\big(d^{2(k^\star+1-\ell)} \log(d/\delta)\big)$.
5:     **for** each $j \in \mathcal{I}_\ell$ **do**
6:         Initialize counter $c_j \leftarrow 0$.
7:         **for** $r = 1$ to $m$ **do**
8:             Run FXRS (Alg. 4) with candidate $j$ to obtain a full rollout (or retry until success). If FXRS fails, repeat until success.
9:             Query $\mathbf{R}_{\mathbf{x}}^{f_{S\star}}$ on the final pre-EOS token of the rollout; record one Bernoulli outcome $Y_{j,r} \in \{0,1\}$.
10:             Update $c_j \leftarrow c_j + Y_{j,r}$.
11:         **end for**
12:         Set empirical acceptance $\hat{p}_j \leftarrow c_j/m$.
13:     **end for**
14:     Let $\hat{j}_\ell \leftarrow \arg\max_{j \in \mathcal{I}_\ell} \hat{p}_j$.
15:     Commit depth-$\ell$: resample until $\hat{\boldsymbol{\mu}}^{x_{i_\ell}} = \boldsymbol{\mu}^{x_{\hat{j}_\ell}}$, then compute $\hat{\boldsymbol{\mu}}^{z_\ell} = \text{FFN}_\ell(\hat{\boldsymbol{\mu}}^{x_{i_\ell}}, \mathbf{E}[z_{\ell-1}]_{:d_{\mathbf{x}}})$; update $\mathcal{H} \leftarrow (\mathcal{H}, \hat{\boldsymbol{\mu}}^{x_{\hat{j}_\ell}}, \hat{\boldsymbol{\mu}}^{z_\ell})$.
16: **end for**
17: Return the committed path encoded by $\mathcal{H}$.

---

**Proof.** *Proof of Prop. 1.*

***Step 0 (probability spaces and notation).*** *For any event A, we write*

$$\Pr[A] := \mathbb{E}_{\mathbf{x} \sim \text{Unif}([K]^d)}\big[\Pr_{\text{TF}_{\text{base}}} (A \mid \mathbf{x})\big].$$

*That is, all probabilities $\Pr_{\text{TF}_{\text{base}}}(\cdot \mid \mathbf{x})$ are w.r.t. the internal sampling of $\text{TF}_{\text{base}}$, conditional on the fixed input $\mathbf{x}$. Unconditional probabilities/expectations $\Pr[\cdot]$ and $\mathbb{E}_{\mathbf{x}}[\cdot]$ are taken over $\mathbf{x} \sim \text{Unif}([K]^d)$. We use $\mathcal{H}_{\ell-1}$ to denote the event that the history up to depth $\ell-1$ is correct, and we write*

$$C_\ell(j) := \{\text{at depth } \ell, \text{ the visible token is forced to } x_{i_\ell} = x_j\}$$

*for the FXRS forcing event.*

***Step 1 (conditioning and notation).*** *Fix $\mathbf{x}$ and condition on $\mathcal{H}_{\ell-1}$. Let $j_\star$ be the unique correct child at depth $\ell$. For $t \in \{\ell+1, \ldots, k^\star+1\}$, let $J_t$ denote the random child index selected by $\text{TF}_{\text{base}}$ at depth $t$ given the preceding context; conditionally on $\mathcal{H}_{\ell-1}$ and on $C_\ell(j)$, we have $J_t \sim \text{Unif}(\mathcal{I}_t)$ with $|\mathcal{I}_t| = \Theta(d)$, independent across $t$; at $t=k^\star+1$, the correct choice is $d+1$ (EOS).*

***Step 2 (events and acceptance).*** *Define the* true-suffix *event*

$$E_{true} := \bigcap_{t=\ell+1}^{k^\star+1} \{J_t = j_t^\star\},$$

*where $j_t^\star$ is the unique correct child at depth $t$ given the correct history (with $j_{k^\star+1}^\star = d+1$). Let $E_{spur}$ be the complement of $E_{true}$ while still yielding acceptance by $\mathbf{R}_{\mathbf{x}}^{f_{S\star}}$ on the final pre-EOS token. Then, for fixed $j$ at depth $\ell$,*

$$P_\ell^{term}(j) := \mathbb{E}_{\mathbf{x}}\Big[\Pr_{\text{TF}_{\text{base}}} (E_{true} \mid \mathbf{x}, \mathcal{H}_{\ell-1}, C_\ell(j)) + \Pr_{\text{TF}_{\text{base}}} (E_{spur} \mid \mathbf{x}, \mathcal{H}_{\ell-1}, C_\ell(j))\Big].$$

***Step 3 (true-suffix probability including EOS).*** *Under the copied PART uniform branching and uniqueness, for $j=j_\star$ we must select the unique correct child at each of the remaining depths, including the terminating EOS choice. Hence, for every $\mathbf{x}$,*

$$\Pr_{\text{TF}_{\text{base}}} (E_{true} \mid \mathbf{x}, \mathcal{H}_{\ell-1}, C_\ell(j_\star)) := \prod_{t=\ell+1}^{k^\star+1} \frac{1}{|\mathcal{I}_t|} := \Theta\big(d^{-(k^\star+1-\ell)}\big).$$

*Taking $\mathbb{E}_{\mathbf{x}}[\cdot]$ preserves the same order.*

**Step 4 (uniform spurious bound).** *By definition,*

$$\mathbb{E}_{\mathbf{x}}\Big[\Pr_{\mathrm{TF}_{\mathrm{base}}}\big(E_{spur} \mid \mathbf{x}, \mathcal{H}_{\ell-1}, C_\ell(j)\big)\Big] \leq \rho_{spur}^{sup,>\ell} < 1$$

*for all $j \in \mathcal{I}_\ell$.*

**Step 5 (gap lower bound).** *Therefore,*

$$\begin{aligned}
P_\ell^{term}(j_\star) - P_\ell^{term}(j) &= \mathbb{E}_{\mathbf{x}}\Big[\Pr_{\mathrm{TF}_{\mathrm{base}}}\big(E_{true} \mid \mathbf{x}, \mathcal{H}_{\ell-1}, C_\ell(j_\star)\big)\Big] \\
&\quad + \mathbb{E}_{\mathbf{x}}\Big[\Pr_{\mathrm{TF}_{\mathrm{base}}}\big(E_{spur} \mid \mathbf{x}, \mathcal{H}_{\ell-1}, C_\ell(j_\star)\big) - \Pr_{\mathrm{TF}_{\mathrm{base}}}\big(E_{spur} \mid \mathbf{x}, \mathcal{H}_{\ell-1}, C_\ell(j)\big)\Big] \\
&\geq c\, d^{-(k^\star+1-\ell)} - \rho_{spur}^{sup,>\ell},
\end{aligned}$$

*for some absolute constant $c > 0$. Rearranging constants yields the stated $\Theta(\cdot) \cdot (1 - \rho_{spur}^{sup,>\ell})$ form.*

**Step 6 (oracle query complexity per depth).** *Let the one-vs-best acceptance-gap at depth $\ell$ be $\Delta := P_\ell^{term}(j_\star) - \max_{j \neq j_\star} P_\ell^{term}(j) = \Theta\big(d^{-(k^\star+1-\ell)}\big)\big(1 - \rho_{spur}^{sup,>\ell}\big)$. Viewing each candidate $j \in \mathcal{I}_\ell$ as a Bernoulli arm with mean $P_\ell^{term}(j)$, any $\delta$-correct identification among $|\mathcal{I}_\ell| = \Theta(d)$ arms requires (in expectation)*

$$\Omega\big(\Delta^{-2}\log(d/\delta)\big)$$

*oracle observations by Lemma 2 and the confidence allocation via union bound (sufficiency for uniform sampling also follows from Lemma 1). Substituting the explicit $\Delta$ gives $\Omega\big(d^{2(k^\star+1-\ell)}\log(d/\delta)\big)$.*

**Step 7 (overall $T_{data}$).** *Summing over depths $\ell = 1, \ldots, k^\star + 1$ gives $T_{data} \geq \tilde{\Omega}\big(d^{2k^\star+1}(1 - \bar{\rho}_{spur})^{-2}\big)$.*

**Step 8 (overall $T_{comp}$).** *In the best case (minimal token emissions) for FXRS, once the forced visible token matches (success on first try), one can immediately append EOS. This uses a constant number of model emissions (exactly 2). Therefore the model-sampling cost is at minimal a constant multiple of the oracle-query cost at that depth (for parity task, the chance is $1/2$, therefore the number to reach is $\Theta(\log(1/\delta))$ for allowable probability $1 - o(1/\delta)$), and aggregating over depths yields a $T_{comp}$ lower bound matching the order of $T_{data}$ (up to polylog and spurious factors). Also, the best case for the resampling at Line 15 of Alg. 5 is $k^\star + 1$ token emissions, which is smaller than $\tilde{\Omega}\big(d^{2k^\star+1}(1 - \bar{\rho}_{spur})^{-2}\big)$, yielding the final $T_{comp} \geq \big(d^{2k^\star+1}(1 - \bar{\rho}_{spur})^{-2}\big)$.* $\qquad\square$

**Theorem 11** (Formal Version of the Second Item in Thm. 3 (complexities in $T_{\mathrm{data}}$ and $T_{\mathrm{comp}}$)). *Assume $\mathrm{TF}_{\mathrm{base}}$ copies PART probability behavior with a unique correct child per depth. Consider the layer-wise procedure in Alg. 6 that, at each depth $\ell$, uses FXRS (Alg. 4) to force a candidate visible x-token for every $j \in \mathcal{I}_\ell$, queries the family oracle $\mathbf{R}_{\mathbf{x}}^{\mathcal{F}_{S_\star}}$ under external truncation, and runs BAI (Alg. 5) to pick the best arm; then commits the chosen $(x_{i_\ell}, z_\ell)$ and proceeds to the next depth.*

*Then there exists a choice of per-depth repetitions $m_\ell = \Theta\big(d^2(1 - \rho_{spur}^{sup,\ell})^{-2}\log(d/\delta)\big)$ such that identifying $S_\star$ with probability at least $1 - \delta$ is achievable with*

$$T_{data} \leq \tilde{O}\big((k^\star+1)d^2(1 - \rho_{spur}^{\max})^{-2}\big), \qquad T_{comp} \leq \tilde{O}\big((k^\star+1)d^3(1 - \rho_{spur}^{\max})^{-2}\big),$$

*where $\rho_{spur}^{\max} := \max_{\ell \in [k^\star+1]} \rho_{spur}^{sup,\ell}$. Here $\rho_{spur}^{sup,\ell} \in [0,1)$ denotes the family-oracle, per-depth spurious-acceptance parameter under external truncation at depth $\ell$; formally,*

$$\rho_{\mathrm{spur}}^{\mathrm{sup},\ell} := \sup_{\substack{j \in \mathcal{I}_\ell \\ j \neq j_\star}} \mathbb{E}_{\mathbf{x}}\Big[\Pr_{\mathrm{TF}_{\mathrm{base}}}\big(\mathbf{R}_{\mathbf{x}}^{\mathcal{F}_{S_\star}}(\textit{accept at depth } \ell) = 1 \mid \mathbf{x}, \mathcal{H}_{\ell-1}, C_\ell(j)\big)\Big].$$

*Detailed derivations are given in the proof.*

**Proof.** **Step 0 (notation and conditioning).** *We reuse $\mathcal{H}_{\ell-1}$ for the event that the history up to depth $\ell-1$ is correct, and $C_\ell(j)$ for the inline-forcing event at depth $\ell$ that sets the visible token to $x_{i_\ell} = x_j$. Given $\mathcal{H}_{\ell-1}$ and $C_\ell(j)$, for $t \geq \ell$ let $J_t$ denote the random child index sampled by $\mathrm{TF}_{\mathrm{base}}$ at depth $t$; under PART-like uniform branching, $J_t \sim \mathrm{Unif}(\mathcal{I}_t)$ with $|\mathcal{I}_t| = \Theta(d)$, independent across $t$, and the correct EOS index is $d+1$ at $t = k^\star + 1$.*

---

**Algorithm 6** Layer-wise Truncated Accept-Reject (LTAR) with $\mathbf{R}_{\mathbf{x}}^{\mathcal{F}_{S_\star}}$ (BAI with inline forcing and truncation)

---

1: Inputs: legal sets $\{\mathcal{I}_\ell\}_{\ell=1}^{k^\star}$, confidence level $\delta$, per-depth budgets $m_\ell = \Theta\big(d^2(1 - \rho_{\mathrm{spur}}^{\mathrm{sup},\ell})^{-2} \log(\frac{d}{\delta})\big)$, max trials
   $T_{\max} = \Theta(d\log(\frac{d}{\delta}))$.
2: Initialize committed partial CoT $\mathcal{H} \leftarrow (\,)$.
3: **for** $\ell = 1$ to $k^\star + 1$ **do**
4:    For each $j \in \mathcal{I}_\ell$, estimate acceptance by $m_\ell$ repetitions with inline forcing:
5:       Initialize $c_j \leftarrow 0$.
6:    **for** $r = 1$ to $m_\ell$ **do**
7:       **for** $t = 1$ to $T_{\max}$ **do**
8:          From context $\mathcal{H}$, sample one step with $\mathrm{TF}_{\mathrm{base}}$ to emit $\hat{\boldsymbol{\mu}}^{x_{i_\ell}}$.
9:          **if** $\hat{\boldsymbol{\mu}}^{x_{i_\ell}} = \boldsymbol{\mu}^{x_j}$ **then**
10:             Compute $\hat{\boldsymbol{\mu}}^{z_\ell} \leftarrow \mathrm{FFN}_\ell\big(\hat{\boldsymbol{\mu}}^{x_{i_\ell}}, \mathbf{E}[z_{\ell-1}]_{:d_{\mathrm{x}}}\big)$.
11:             Externally append EOS if $l < k^\star + 1$; query $\mathbf{R}_{\mathbf{x}}^{\mathcal{F}_{S_\star}}$ at depth $\ell$; record Bernoulli $Y_{j,r} \in \{0,1\}$.
12:             Update $c_j \leftarrow c_j + Y_{j,r}$ and **break** the retry loop.
13:          **end if**
14:       **end for**
15:       If no match within $T_{\max}$, optionally increase $T_{\max}$ and repeat this repetition.
16:    **end for**
17:    Set empirical acceptance $\hat{p}_j \leftarrow c_j/m_\ell$ for all $j \in \mathcal{I}_\ell$ and pick $\hat{j}_\ell = \arg\max_j \hat{p}_j$.
18:    Commit depth-$\ell$: resample until $\hat{\boldsymbol{\mu}}^{x_{i_\ell}} = \boldsymbol{\mu}^{x_{\hat{j}_\ell}}$, then compute $\hat{\boldsymbol{\mu}}^{z_\ell} = \mathrm{FFN}_\ell(\hat{\boldsymbol{\mu}}^{x_{i_\ell}}, \mathbf{E}[z_{\ell-1}]_{:d_{\mathrm{x}}})$; update $\mathcal{H} \leftarrow (\mathcal{H}, \hat{\boldsymbol{\mu}}^{x_{\hat{j}_\ell}}, \hat{\boldsymbol{\mu}}^{z_\ell})$.
19: **end for**
20: Return the committed path encoded by $\mathcal{H}$.

---

*Define the depth-$\ell$ acceptance event under the family oracle with external truncation as*

$$A_\ell(j) := \{\mathbf{R}_{\mathbf{x}}^{\mathcal{F}_{S_\star}}(\textit{FXRS-forced } j \text{ at depth } \ell, \text{ truncated at } z_\ell) = 1\}.$$

*The (marginal) acceptance probability for candidate $j$ is*

$$\alpha_j^{(\ell)} := \mathbb{E}_{\mathbf{x}}\Big[\Pr_{\mathrm{TF}_{\mathrm{base}}} \big(A_\ell(j) \mid \mathbf{x}, \mathcal{H}_{\ell-1}, C_\ell(j)\big)\Big].$$

*By definition of the per-depth spurious parameter (family-oracle, external truncation)*

$$\rho_{\mathrm{spur}}^{\mathrm{sup},\ell} := \sup_{\substack{j \in \mathcal{I}_\ell \\ j \neq j_\star}} \mathbb{E}_{\mathbf{x}}\Big[\Pr_{\mathrm{TF}_{\mathrm{base}}} \big(A_\ell(j) \mid \mathbf{x}, \mathcal{H}_{\ell-1}, C_\ell(j)\big)\Big] < 1,$$

*we have $\alpha_j^{(\ell)} \leq \rho_{\mathrm{spur}}^{\mathrm{sup},\ell}$ for all $j \neq j_\star$. For the unique correct child $j_\star$, we assume*

$$\alpha_{j_\star}^{(\ell)} \geq 1 - \eta_\ell, \qquad \eta_\ell \in [0,1), \tag{67}$$

*which captures any oracle or model non-idealities.*

***Step 1 (acceptance gap and BAI sample complexity).*** *Under PART-like uniform branching, a single inline-forcing repetition succeeds in emitting the visible token $x_{i_\ell} = x_j$ within expected $\Theta(d)$ retries. Conditioned on success (we then deterministically compute $z_\ell$ via $\mathrm{FFN}_\ell$ and externally append EOS before querying), the depth-$\ell$ acceptance probabilities satisfy*

$$\alpha_{j_\star}^{(\ell)} = \Theta\big(d^{-1}\big), \qquad \alpha_j^{(\ell)} \leq c\,d^{-1} \rho_{\mathrm{spur}}^{\mathrm{sup},\ell} \quad (j \neq j_\star),$$

*for some absolute constant $c > 0$, since the correct child is selected with probability $\Theta(d^{-1})$ and wrong children are upper-bounded by the spurious-acceptance parameter. Hence the one-vs-best gap obeys*

$$\Delta_\ell := \alpha_{j_\star}^{(\ell)} - \max_{j \neq j_\star} \alpha_j^{(\ell)} = \Theta\big(d^{-1}\big) \cdot \big(1 - \rho_{\mathrm{spur}}^{\mathrm{sup},\ell}\big).$$

*Running BAI at depth $\ell$ with*

$$m_\ell = \Theta\big(\Delta_\ell^{-2}\log(d/\delta)\big)$$

*oracle trials* per arm *identifies $j_\star$ with probability at least $1 - \delta$ by Chernoff bounds and a union bound over the $d$ arms.*

**Step 2 (per-depth reward-oracle query complexity).** *Each successful inline-forcing repetition issues one query to $\mathbf{R}_\mathbf{x}^{\mathcal{F}_{S_\star}}$. With $|\mathcal{I}_\ell|=\Theta(d)$ arms and $m_\ell$ repetitions per arm,*

$$\tilde{O}\big(m_\ell\big) = \tilde{O}\big(\Delta_\ell^{-2}\big) = \tilde{O}\big(d^2, (1 - \rho_{\mathrm{spur}}^{\mathrm{sup},\ell})^{-2}\big),$$

*since $\Delta_\ell^{-2} = \Theta\big(d^2\,(1 - \rho_{\mathrm{spur}}^{\mathrm{sup},\ell})^{-2}\big)$.*

**Step 3 (aggregated $T_{data}$).** *Summing over depths $\ell=1,\ldots,k^\star$ and upper-bounding each per-depth spurious term by $\rho_{\mathrm{spur}}^{\mathrm{max}} := \max_{\ell\in[k^\star+1]}\rho_{\mathrm{spur}}^{\mathrm{sup},\ell}$,*

$$T_{data} \leq \tilde{O}\Big(\sum_{\ell=1}^{k^\star+1} d^2(1 - \rho_{\mathrm{spur}}^{\mathrm{sup},\ell})^{-2}\Big) \leq \tilde{O}\big((k^\star + 1)d^2(1 - \rho_{\mathrm{spur}}^{\mathrm{max}})^{-2}\big).$$

**Step 4 (per-depth model-sampling cost).** *Under LTAR external truncation, each repetition computes $\hat{\boldsymbol{\mu}}^{z_\ell}$ and appends EOS, costing $O(1)$ emissions per retry; with worst-case $O(d\log(d/\delta))$ retries, the per-repetition cost is $O(d\log(d/\delta))$. Hence, per-depth model-sampling cost is*

$$\tilde{O}\big(d \cdot m_\ell\big) = \tilde{O}\big(d^3\,(1 - \rho_{\mathrm{spur}}^{\mathrm{sup},\ell})^{-2}\big).$$

**Step 5 (aggregated $T_{comp}$).** *Summing over depths and upper-bounding by $\rho_{\mathrm{spur}}^{\mathrm{max}}$,*

$$T_{comp} \leq \tilde{O}\Big(\sum_{\ell=1}^{k^\star+1} d^3\,(1 - \rho_{\mathrm{spur}}^{\mathrm{sup},\ell})^{-2}\Big) \leq \tilde{O}\big((k^\star + 1)d^3\,(1 - \rho_{\mathrm{spur}}^{\mathrm{max}})^{-2}\big).$$

**Additional commit cost across depths.** *After identifying $\hat{\hat{j}}_\ell$ at each depth, LTAR performs a commit by resampling until $\hat{\boldsymbol{\mu}}^{x_{i_\ell}} = \boldsymbol{\mu}^{x_{\hat{j}_\ell}}$ and computing $\hat{\boldsymbol{\mu}}^{z_\ell}$, then (optionally) appending EOS. This contributes an extra $\tilde{O}(d)$ emissions per depth in the worst case (Bernstein-type retry bound), for a total $\tilde{O}((k^\star+1)d)$ over all depths. This term is strictly dominated by the $\tilde{O}((k^\star + 1)d^3(1 - \rho_{\mathrm{spur}}^{\mathrm{max}})^{-2})$ bound derived above, and hence is absorbed into $T_{comp}$.*

**Step 6 (success probability across depths).** *Allocate confidence across depths (e.g., replace $\delta$ by $\delta/(k^\star + 1)$ per depth) or absorb this into polylog factors; combining the steps above yields the stated bounds for $T_{data}$ and $T_{comp}$.* $\qquad\square$

**Remark 12** (Spurious-acceptance parameters and their roles). *We use two families of spurious-acceptance parameters, tied to the oracle being queried and to how the rollout is conditioned:*

- $\rho_{\mathrm{spur}}^{\mathrm{sup},>\ell} \in [0,1)$ *(terminal-oracle, suffix-level): for queries to $\mathbf{R}_\mathbf{x}^{f_{S_\star}}$, after forcing a visible child $j$ at depth $\ell$ via FXRS, it upper-bounds the expected probability (over $\mathbf{x}$ and the internal sampling of $\mathrm{TF}_{\mathrm{base}}$) that a wrong suffix on depths $\ell+1,\ldots,k^\star+1$ is nonetheless accepted by the oracle. We also define a uniform bound $\bar{\rho}_{\mathrm{spur}} := \sup_\ell \rho_{\mathrm{spur}}^{\mathrm{sup},>\ell}$ for aggregating across depths.*
- $\rho_{\mathrm{spur}}^{\mathrm{sup},\ell} \in [0,1)$ *(family-oracle, per-depth): for queries to $\mathbf{R}_\mathbf{x}^{\mathcal{F}_{S_\star}}$ under external truncation at depth $\ell$ (so the queried token is $z_\ell$), it upper-bounds the expected acceptance probability of any wrong child $j \neq j_\star$ at depth $\ell$. We also use $\rho_{\mathrm{spur}}^{\mathrm{max}} := \max_{\ell\in[k^\star+1]}\rho_{\mathrm{spur}}^{\mathrm{sup},\ell}$.*

*These quantities are not interchangeable: $\rho_{\mathrm{spur}}^{\mathrm{sup},>\ell}$ refers to suffix-level acceptance under the terminal oracle $\mathbf{R}_\mathbf{x}^{f_{S_\star}}$, while $\rho_{\mathrm{spur}}^{\mathrm{sup},\ell}$ refers to per-depth acceptance under the family oracle $\mathbf{R}_\mathbf{x}^{\mathcal{F}_{S_\star}}$ with truncation. Both families depend on task class parameters (e.g., output alphabet size $K$) and encapsulate the severity of reward hacking. Our bounds carry these terms explicitly: lower bounds involve $(1 - \bar{\rho}_{\mathrm{spur}})^{-2}$; upper bounds involve $\sum_\ell d\,(1 - \rho_{\mathrm{spur}}^{\mathrm{sup},\ell})^{-2}$.*

## H. Proofs of Parity Problem

**Theorem 13** (Representation Theorem of Any Parity Function). *For any $k' \in [d/2]$, any index set $S = S^{k'} \subset [d]$, and the associated parity function $f_{S^{k'}} \in \mathcal{P}_{d,k'}$ defined in equation 6, there exists a transformer*

$$\mathrm{TF}^{(k')}(\cdot; \mathbf{W}^\star)$$

*with parameters $\mathbf{W}^\star$ such that for every $\mathbf{x} \in \{0,1\}^d$, the autoregressive next-token prediction defined in equation 5 produces exactly the subtask decomposition sequence in equation 6, up to the concatenation with positional embeddings.*

*Concretely, let $S^{k'} = \{i_1, \ldots, i_{k'}\}$ and define $z_m$ recursively as in equation 6. Then*

$$\mathrm{TF}^{(k')}\big(\mathbf{E}[x_1], \ldots, \mathbf{E}[x_d], \mathbf{E}[\mathrm{EOS}]; \mathbf{W}^\star\big) = \mathbf{E}[z_1],$$

*and more generally for $m = 2, \ldots, k'$,*

$$\mathrm{TF}^{(k')}\big(\mathbf{E}[x_1], \ldots, \mathbf{E}[x_d], \mathbf{E}[\mathrm{EOS}], \mathbf{E}[z_1], \ldots, \mathbf{E}[z_{m-1}]; \mathbf{W}^\star\big) = \mathbf{E}[z_m],$$

*where each $\mathbf{E}[z_m] = [\, \boldsymbol{\mu}^{z_m}, \mathbf{p}_{d+m}\,]^\top$ is the concatenation of the token embedding $\boldsymbol{\mu}^{z_m}$ and its positional embedding $\mathbf{p}_{d+m}$. Finally, at step $k'+1$ the model deterministically outputs the $\mathrm{EOS}$ embedding:*

$$\mathrm{TF}^{(k')}\big(\mathbf{E}[x_1], \ldots, \mathbf{E}[x_d], \mathbf{E}[\mathrm{EOS}], \mathbf{E}[z_1], \ldots, \mathbf{E}[z_{k'}]; \mathbf{W}^\star\big) = \mathbf{E}[\mathrm{EOS}].$$

*Thus, $\mathrm{TF}^{(k')}(\cdot; \mathbf{W}^\star)$ exactly realizes the chain-of-thought subtask decomposition of $f_{S^{k'}}$ via next-token prediction.*

**Theorem 14** (`PART` Behavior of Parity Class)**.** *Consider a base model with Uniform Ordered Transition Probability at each node, where "legal" means both non-repeating and maintaining strictly increasing order ($i_1 < i_2 < i_3 < \cdots$), and where $k = o(d) < \lfloor d/2 \rfloor$ satisfying $d - k = \Theta(d)$.*

*Under this assumption, at each parent node, the model uniformly selects among all legal children nodes. Specifically:*

- *At the Root node: $d$ variables $\{x_1, x_2, \ldots, x_d\}$ are available, each with probability $1/d$*

- *At node $x_i$ in the tree: only variables $\{x_{i+1}, x_{i+2}, \ldots, x_d\}$ and EOS are legal, each with probability $1/(d - i + 1) = \Theta(d^{-1})$.*

- *EOS is always a legal choice at any node, ensuring termination*

*For a given $d$ and $k < \lfloor d/2 \rfloor$, the number of legal CoT sequences of length $k' + 1$ (including the EOS token) that compute parity functions $f_{S^{k'}} \in \mathcal{P}_{d,k'}$ is:*

$$|\mathcal{L}_{k'+1}| = \binom{d}{k'} \tag{68}$$

*This counts the number of ways to choose $k'$ distinct variables from $d$ available variables while maintaining strictly increasing order, followed by an EOS token.*

*Under the Uniform Ordered Transition Probability assumption, the probability of any specific legal CoT sequence of length $k' + 1$ is:*

$$P(\text{specific sequence of length } k' + 1) = O\big(\frac{(\log d)^{k'}}{d^{k'+1}}\big) \tag{69}$$

*When $k' = O\left(\frac{\log d}{\log\log d}\right)$, it holds that $P(\text{specific sequence of length } k' + 1) \leq O(d^{-k'})$*

*The probability of generating any CoT sequence of length $k' + 1$ is:*

$$P(\text{length } k' + 1) = O\big(\binom{d}{k'} \frac{(\log d)^{k'}}{d^{k'+1}}\big) \tag{70}$$

*This probability distribution over CoT lengths exhibits a systematic bias that depends on the relationship between $k'$ and $d$.*

**Proof.** *We analyze the structure of legal CoT sequences systematically:*

***Step 1: Legal Sequence Structure and Length Constraint.*** *A legal CoT sequence that computes a parity function $f_{S^{k'}}$ must have length $k' + 1$ and be structured as:*

$$Root \rightarrow x_{i_1} \rightarrow x_{i_2} \rightarrow \cdots \rightarrow x_{i_{k'}} \rightarrow EOS \tag{71}$$

*where $\{i_1, i_2, \ldots, i_{k'}\}$ are distinct indices from $[d]$ with strictly increasing order ($i_1 < i_2 < \cdots < i_{k'}$), satisfying our constraints for parity problems. Every legal CoT must terminate with EOS to indicate completion of the parity computation. The constraint $k < \lfloor d/2 \rfloor$ ensures that at the maximum length $k + 1$, the only legal next prediction is EOS, preventing unbounded tree growth.*

**Step 2: Counting Legal Sequences of Length** $k' + 1$**.** *The number of legal sequences of length $k' + 1$ (including EOS) that compute parity functions $f_{S^{k'}}$ is:*

$$|\mathcal{L}_{k'+1}| = \binom{d}{k'} \tag{72}$$

*This counts:*

- $\binom{d}{k'}$ *ways to choose $k'$ distinct variables from $d$ available variables*

- *The ordering constraint $i_1 < i_2 < \cdots < i_{k'}$ eliminates the need for permutations*

- *Each sequence must end with EOS to be valid, making the total length $k' + 1$*

**Step 3: Transition Probability Calculation.** *Under the Uniform Ordered Transition Probability assumption, the probability of any specific legal sequence is the product of transition probabilities at each step:*

*For a sequence Root $\to x_{i_1} \to x_{i_2} \to \cdots \to x_{i_{k'}} \to$ EOS ($1 \le i_1 < i_2 < \cdots < i_{k'} \le d$):*

- *Root $\to x_{i_1}$: probability $1/d$ (choosing from $d$ variables)*

- *$x_{i_1} \to x_{i_2}$: probability $d^{-(d-i_1+1)}$ (choosing from remaining $d - i_1 + 1$ variables)*

- *$x_{i_2} \to x_{i_3}$: probability $d^{-(d-i_2+1)}$ (choosing from remaining $d - i_2 + 1$ variables)*

- $\vdots$

- *$x_{i_{k'-1}} \to x_{i_{k'}}$: probability $d^{-(d-i_{k-1}+1)}$ (choosing from remaining $d - i_{k'} + 1$ variables)*

- *$x_{i_{k'}} \to$ EOS: probability $d^{-(d-i_k+1)}$.*

*Therefore:*

$$P(\text{specific sequence}) := \mathbb{E}[F] = \mathbb{E}\left[\frac{1}{d(d - i_1 + 1) \cdots (d - i_{k'} + 1)}\right] \tag{73}$$

*For $1 \le k' \le d$, we here serve to show that*

$$\mathbb{E}[F] = O\left(\frac{(\log d)^{k'}}{d^{k'+1}}\right).$$

*By Lemma 9,*

$$\mathbb{E}[F] = \frac{e_{k'}\left(1, \frac{1}{2}, \ldots, \frac{1}{d}\right)}{d\binom{d}{k'}}.$$

*Recall Maclaurin's inequality (monotonicity of symmetric means): for any $a_1, \ldots, a_d \ge 0$, if*

$$S_k(a) := \frac{e_k(a_1, \ldots, a_d)}{\binom{d}{k}} \quad (k = 1, \ldots, d),$$

*then the sequence $\{S_k(a)^{1/k}\}_{k=1}^d$ is nonincreasing; in particular*

$$S_{k'}(a) \le \left(S_1(a)\right)^{k'} \quad \text{for all } k' \ge 1.$$

*Apply this with $a_t = 1/t$ $(t = 1, \ldots, d)$. We get*

$$\frac{e_{k'}\left(1, \frac{1}{2}, \ldots, \frac{1}{d}\right)}{\binom{d}{k'}} \leq \left(\frac{e_1\left(1, \frac{1}{2}, \ldots, \frac{1}{d}\right)}{\binom{d}{1}}\right)^{k'} = \left(\frac{H_d}{d}\right)^{k'},$$

*where $H_d = \sum_{t=1}^{d} \frac{1}{t}$ is the d-th harmonic number. Hence*

$$\mathbb{E}[F] \leq \frac{1}{d}\left(\frac{H_d}{d}\right)^{k'}.$$

*Using the standard bound $H_d \leq 1 + \log d$ for all $d \geq 1$, we obtain*

$$\mathbb{E}[F] \leq \frac{1}{d}\left(\frac{1 + \log d}{d}\right)^{k'} = O\left(\frac{(\log d)^{k'}}{d^{k'+1}}\right),$$

*which proves the claim.*

*In particular, when $k' = O\left(\frac{\log d}{\log \log d}\right)$, the logarithmic factor in the bound above never exceeds a fixed polynomial in d. Indeed, observe that*

$$(\log d)^{k'} \leq (\log d)^{C \cdot \frac{\log d}{\log \log d}} = \exp\left(C \cdot \frac{\log d}{\log \log d} \cdot \log \log d\right) = d^C,$$

*for some absolute constant $C > 0$. Therefore*

$$\mathbb{E}[F] \leq \frac{(\log d)^{k'}}{d^{k'+1}} \leq \frac{d^C}{d^{k'+1}} = O(d^{-k'}),$$

*which shows that in the regime $k' = O\left(\frac{\log d}{\log \log d}\right)$ the expectation decays at least on the order of $d^{-k'}$.*

**Step 4: Probability of Length $k' + 1$.** *Since there are $\binom{d}{k'}$ different sequences of length $k' + 1$, and each has the same probability structure (though with different specific values), the total probability of generating any CoT sequence of length $k' + 1$ is:*

$$P(\text{length } k' + 1) = \sum_{\substack{\text{all sequences of length } k'+1}} P(\text{specific sequence}) \tag{74}$$

*This can be expressed as:*

$$P(\text{length } k' + 1) = \Theta\left(\binom{d}{k'} \cdot d^{-k'}\right) \tag{75}$$

$\square$

**Lemma 9** (Expected reciprocal product over a random $k'$-subset). *Fix integers $d \geq 1$ and $0 \leq k' \leq d$. Choose indices*

$$1 \leq i_1 < i_2 < \cdots < i_{k'} \leq d$$

*uniformly at random among all $\binom{d}{k'}$ $k'$-subsets of $\{1, \ldots, d\}$, and define*

$$F(i_1, \ldots, i_{k'}) := \frac{1}{d \prod_{j=1}^{k'}(d - i_j + 1)}.$$

*Let the degree-$k'$ elementary symmetric polynomial be*

$$e_{k'}(y_1, \ldots, y_d) := \sum_{1 \leq t_1 < \cdots < t_{k'} \leq d} y_{t_1} \cdots y_{t_{k'}}, \qquad (e_0 \equiv 1),$$

*and write the (generalized) harmonic numbers*

$$H_d := \sum_{t=1}^{d} \frac{1}{t}, \qquad H_d^{(m)} := \sum_{t=1}^{d} \frac{1}{t^m} \quad (m \geq 2).$$

*Then the expectation of $F$ admits the exact closed form*

$$\boxed{\mathbb{E}[F] = \frac{e_{k'}\left(1, \frac{1}{2}, \ldots, \frac{1}{d}\right)}{d\binom{d}{k'}}}$$

*and, for fixed $k'$ as $d \to \infty$, the asymptotic expansion*

$$\boxed{\mathbb{E}[F] = \frac{1}{d\binom{d}{k'}}\left(\frac{(\log d)^{k'}}{k'!} + \frac{\gamma(\log d)^{k'-1}}{(k'-1)!} + O\big((\log d)^{k'-2}\big)\right) = \frac{(\log d)^{k'}}{d^{k'+1}}\left(1 + O\big(\tfrac{1}{\log d}\big)\right),}$$

*where $\gamma$ is the Euler–Mascheroni constant. In particular,*

$$\mathbb{E}[F] \asymp \frac{(\log d)^{k'}}{d^{k'+1}}.$$

*Moreover, the first few exact instances are*

$$k' = 1: \quad \mathbb{E}[F] = \frac{H_d}{d\binom{d}{1}} = \frac{H_d}{d^2},$$

$$k' = 2: \quad \mathbb{E}[F] = \frac{H_d^2 - H_d^{(2)}}{2d\binom{d}{2}},$$

$$k' = 3: \quad \mathbb{E}[F] = \frac{H_d^3 - 3H_d H_d^{(2)} + 2H_d^{(3)}}{6d\binom{d}{3}}.$$

**Proof.** **Step 1: Reparametrization and identification with an elementary symmetric sum.** *Define $t_j := d - i_j + 1$ for $j = 1, \ldots, k'$. Because $i_1 < \cdots < i_{k'}$, we have*

$$1 \le t_{k'} < \cdots < t_1 \le d,$$

*so $\{t_1, \ldots, t_{k'}\}$ is exactly a $k'$-subset of $\{1, \ldots, d\}$ and, since the $(i_1, \ldots, i_{k'})$ are chosen uniformly, the unordered set $\{t_1, \ldots, t_{k'}\}$ is also uniformly distributed over all $\binom{d}{k'}$ $k'$-subsets. With this change of variables,*

$$\prod_{j=1}^{k'}(d - i_j + 1) = \prod_{j=1}^{k'} t_j, \qquad F(i_1, \ldots, i_{k'}) = \frac{1}{d} \cdot \frac{1}{\prod_{j=1}^{k'} t_j}.$$

*Taking expectation over all $\binom{d}{k'}$ subsets and recalling the definition of $e_{k'}$,*

$$\mathbb{E}[F] = \frac{1}{d\binom{d}{k'}} \sum_{1 \le t_1 < \cdots < t_{k'} \le d} \frac{1}{t_1 \cdots t_{k'}} = \frac{e_{k'}\left(1, \frac{1}{2}, \ldots, \frac{1}{d}\right)}{d\binom{d}{k'}}.$$

*This proves the exact formula.*

**Step 2: A generating-function expression for $e_{k'}\left(1, \frac{1}{2}, \ldots, \frac{1}{d}\right)$.** *Consider*

$$G_d(x) := \prod_{t=1}^{d}\left(1 + \frac{x}{t}\right) = \sum_{k=0}^{d} e_k\left(1, \frac{1}{2}, \ldots, \frac{1}{d}\right) x^k.$$

*Taking logarithms and expanding,*

$$\log G_d(x) = \sum_{t=1}^{d} \log\left(1 + \frac{x}{t}\right) = \sum_{m \ge 1} \frac{(-1)^{m+1}}{m} x^m \underbrace{\sum_{t=1}^{d} \frac{1}{t^m}}_{H_d^{(m)}} = \sum_{m \ge 1} \frac{(-1)^{m+1}}{m} H_d^{(m)} x^m.$$

*Hence*

$$G_d(x) = \exp\left(\sum_{m \geq 1} \frac{(-1)^{m+1}}{m} H_d^{(m)} x^m\right),$$

*so the coefficient*

$$e_{k'}\left(1, \tfrac{1}{2}, \ldots, \tfrac{1}{d}\right) = [x^{k'}] G_d(x)$$

*is a degree-$k'$ polynomial in the variables $H_d^{(1)}, \ldots, H_d^{(k')}$ with leading term*

$$\frac{(H_d)^{k'}}{k'!} \qquad (H_d = H_d^{(1)}),$$

*and without an $H_d^{k'-1}$ term (this is the standard consequence of Newton's identities / exponential formula; concrete examples for $k' = 2, 3$ are listed in the statement).*

***Step 3: Asymptotics of the ingredients and of the ratio.*** *We use the classical asymptotics*

$$H_d = \log d + \gamma + o(1), \qquad H_d^{(m)} \to \zeta(m) \ (m \geq 2), \qquad \binom{d}{k'} = \frac{d^{k'}}{k'!}\left(1 + O(\tfrac{1}{d})\right), \quad (d \to \infty, \ k' \text{ fixed}).$$

*Plugging these into the polynomial expression for $e_{k'}$ from Step 2 gives*

$$e_{k'}\left(1, \tfrac{1}{2}, \ldots, \tfrac{1}{d}\right) = \frac{(\log d + \gamma + o(1))^{k'}}{k'!} + O\left((\log d)^{k'-2}\right) = \frac{(\log d)^{k'}}{k'!} + \frac{\gamma(\log d)^{k'-1}}{(k'-1)!} + O\left((\log d)^{k'-2}\right).$$

*Therefore*

$$\frac{e_{k'}(1, 1/2, \ldots, 1/d)}{\binom{d}{k'}} = \frac{(\log d)^{k'}}{d^{k'}}\left(1 + O\left(\frac{1}{\log d}\right)\right),$$

*and multiplying by the prefactor $1/d$ yields*

$$\mathbb{E}[F] = \frac{(\log d)^{k'}}{d^{k'+1}}\left(1 + O\left(\frac{1}{\log d}\right)\right),$$

*which is exactly the asserted asymptotic expansion in the lemma.*

***Step 4 (Optional sanity check).*** *By Maclaurin's inequalities,*

$$\frac{e_{k'}(1, 1/2, \ldots, 1/d)}{\binom{d}{k'}} \leq \left(\frac{e_1}{\binom{d}{1}}\right)^{k'} = \left(\frac{H_d}{d}\right)^{k'} \sim \frac{(\log d)^{k'}}{d^{k'}},$$

*so $\mathbb{E}[F] \leq (\log d)^{k'}/d^{k'+1}$ up to lower-order factors, matching the leading term above.*

$\square$

**Proof.** *Representation for* `PART` *of parity. We fix the first $d_\mathcal{X}$ columns of $\mathbf{K}, \mathbf{Q}$ to zero so that the attention scores are determined by only the positional encodings. This ensures that the transformer focuses on learning which positions contribute to the parity at each step. $\mathbf{K}, \mathbf{Q}$ are then reparametrized by a single matrix $\mathbf{W} \in \mathbb{R}^{(d_\mathrm{E} - d_\mathcal{X})^2}$; conversely, the value matrix is set to only preserve the $\mathbf{x}$ component, as follows.*

$$\mathbf{K}^\top \mathbf{Q} = \begin{pmatrix} \mathbf{0}_{d_\mathcal{X} \times d_\mathcal{X}} & \mathbf{0}_{d_\mathcal{X} \times (d_\mathrm{E} - d_\mathcal{X})} \\ \mathbf{0}_{(d_\mathrm{E} - d_\mathcal{X}) \times d_\mathcal{X}} & \mathbf{W} \end{pmatrix}, \quad \mathbf{V} = \begin{pmatrix} \mathbb{I}_{d_\mathcal{X} \times d_\mathcal{X}} & \mathbf{0}_{d_\mathcal{X} \times (d_\mathrm{E} - d_\mathcal{X})} \end{pmatrix}.$$

*This type of reparametrization is common in the literature to make dynamical analysis tractable (Kim & Suzuki, 2024; 2025). Then denote $x_{m+d+1} = z_m$, $m \in [k]$, we have $\hat{\mathbf{z}}_m = \sum_{j=1}^{m-1+d} \sigma_j(\mathbf{w}_m) \boldsymbol{\mu}^{x_j}$ where the softmax scores $\sigma_j(\mathbf{W}_m) = e^{w_{j,m}} / \sum_{\alpha=1}^{m-1+d} e^{w_{\alpha,m}}$.*

***W Matrix for Uniform Ordered Transition Probability.*** *For $0 < m \leq k$, $i_m \neq d + 1$*

$$\mathbf{p}_{d+1+m} = \mathbf{p}_{i_m}$$
$$\mathbb{E}[x_{d+1+m}] := [\boldsymbol{\mu}^{z_m}, \mathbf{p}_{i_m}]^\top$$
$$(\mathbf{p}_{j \leq i_m})\mathbf{W}(\mathbf{p}_{i_m}) = -\infty \tag{76}$$
$$(\mathbf{p}_{i_m < j \leq d+1})\mathbf{W}(\mathbf{p}_{i_m}) = c_{i_m}$$

where $z_m = z_{m-1} \oplus x_{i_m}$.

For $\mathbb{E}[x_{d+1}] = [\boldsymbol{\mu}^{\mathrm{EOS}}, \mathbf{p}_{d+1}]^\top$,

$$(\mathbf{p}_{j<d+1})\mathbf{W}(\mathbf{p}_{d+1}) = c_{d+1} \tag{77}$$

Then, denote $x_{m+d+1} = z_m, m \in [k]$, we have

$$\boldsymbol{\mu}^{x_{i_m}} = \hat{\boldsymbol{\mu}}^{x_{i_m}} \sim \mathrm{softmax}\left(\left(\sum_{j<m-1+d} \frac{e^{(\mathbf{p}_j)\mathbf{W}(\mathbf{p}_{i_{m-1}})}}{\sum_{j<m-1+d} e^{(\mathbf{p}_j)\mathbf{W}(\mathbf{p}_{i_{m-1}})}} \boldsymbol{\mu}^{x_j}\right)^\top \mathbf{U}/\beta\right). \tag{78}$$

Then $\mathbb{E}[z_m] = \mathbb{E}[x_{m+d+1}] = [\phi(\hat{\boldsymbol{\mu}}^{x_{i_m}} + \boldsymbol{\mu}^{z_{m-1}}), \mathbf{p}_{i_m}]^\top$. $\qquad\square$

