# OpenReview forum: "Provable Sample Efficiency of Curriculum Post-Training for Transformer Reasoning"
_ICML.cc/2026/Conference — ICML 2026 regular_

### Official Review · Reviewer_gJFb · 2026-03-12

**Soundness:** 3
**Presentation:** 3
**Significance:** 3
**Originality:** 3
**Overall Recommendation:** 5
**Confidence:** 5

**Summary:**

The paper provides a rigorous theoretical framework for understanding when curriculum learning is beneficial in the post-training of Large Language Models. Curriculum learning has recently shown promise in post-training, particularly for smaller language models, so developing a theory that explains when and why it helps is both timely and valuable.

**Compliance With Llm Reviewing Policy:**

Affirmed.

**Final Justification:**

The authors addressed my concerns, which led me to increase my confidence after rebuttal.

**Key Questions For Authors:**

1. Could the authors clarify why the proof cannot be extended to settings such as Blocksworld? For instance, could one not model the configuration of block positions as a single state and analyze the process within the same framework?

2. Another emerging paradigm for transformer-based reasoning is Masked Diffusion Language Models, which have recently shown reasoning performance competitive with standard LLMs. Since the paper models reasoning as a Markov process, it would be interesting to know whether the theory could also be extended to DLMs, and whether it might similarly suggest that curriculum learning is beneficial in that setting.

3. Could the authors also clarify how the attention maps help determine whether curriculum learning is effective, and more broadly how Figure 3 should be interpreted? At present, the connection between these visualizations and the paper’s main claims is not entirely clear.

**Limitations:**

Yes

**Strengths And Weaknesses:**

Strengths:
1. The paper is well written, and the proofs and theorems are presented in a clear and coherent manner.
2. The work builds on curriculum learning theory developed in prior literature, including Foster and Foerster as well as Parashar et al. This grounds the contribution in a relevant and active line of research, especially one that has already shown promising empirical results. The connections drawn to prior work also improve the overall clarity and positioning of the paper.
3. In addition, the paper discusses two forms of curriculum learning that are commonly used in LLM post-training, namely depth-focused and hint-focused curricula. This makes the contribution more thorough and helps connect the theoretical analysis to practical settings of interest.

Weaknesses:
1. I felt that the empirical evaluation was somewhat rushed and underexplained. A tighter connection between the theoretical analysis and the empirical findings would have significantly strengthened the paper.

---

> ### Author Rebuttal · Authors · 2026-03-31
>
> We are truly grateful for the reviewer’s careful reading and constructive feedback. We especially appreciate the reviewer’s recognition that our paper provides a unified theoretical framework for curriculum post-training. We are also glad that the reviewer found the writing, theorems, and proofs clear and coherent.
>
> > **W1 & Q3**: Room for intepretation of Figures.
>
> **A1:** In our transformer realization of 2S-ART, **attention** is responsible for selecting the relevant earlier token/state, while the **FFN** performs the corresponding atomic operation (e.g., XOR $⊕$) based on the current state and the retrieved token. For Fig. 3(b), consider the target parity task with $d=8$ and secret indices $[0,1,2]$. The corresponding parent-child relation is $8\to 0\to 1\to 2\to 8$, where $8$ denotes the final EOS token in the input. In other words, starting from the EOS token, the model should attend to the secret indices step by step, and terminate the CoT by correctly outputting $x_1⊕x_2⊕x_3$ followed by EOS. From this perspective, the attention map directly reflects whether the model has recovered the intended reasoning chain.
>
> With this in mind, Fig. 3(b), which shows the attention map after successful fine-tuning, should be interpreted as follows. Both curriculum strategies concentrate sharply on the correct transition chain, with only minimal leakage to spurious edges. By contrast, direct fine-tuning spreads substantial probability mass over incorrect parent-child links even with significantly more samples, reflecting the intrinsic difficulty of direct training. The accuracy curves in Fig. 3(a,c) show the sample/query complexity separation between curriculum and non-curriculum strategies. Together, these results corroborate the predictions of Thm. 2 and Thm. 3, namely that curriculum learning can yield exponential improvement. In the appendix, the same qualitative pattern persists in broader settings. We'll revise the paper to make the interpretation of Fig. 3 and the appendix figures much more explicit.
> > **Q1**: Extensions to Blocksworld.
>
> **A2:** Thank you for this insightful suggestion. We believe this is a very reasonable perspective, and it indeed suggests that our previous discussion can be stated more carefully. Our original hesitation about Blocksworld was that, unlike parity, each intermediate configuration is not naturally a token from a small fixed dictionary (e.g., for parity, each reasoning state $z_k⊕x_{k+1}$ is $0$ or $1$). However, as the reviewer suggests, one can instead treat the full block configuration as a single state, namely as an element in a much larger state dictionary containing all feasible configurations. Under this view, Blocksworld can in principle still fit the same high-level autoregressive framework.
>
> From the perspective of our transformer realization, this is also compatible with the intended mechanism: attention associates the current state with a relevant earlier state or contextual position, and the FFN implements the local transition operator that maps one valid configuration to the next. So at a high level, we agree that Blocksworld appears extendable into the same framework by enlarging the state space. We thank the reviewer for pointing this out, and we'll clarify this in the revision.
>
> > **Q2**: Extension to DLMs.
>
> **A3:** We agree with this perspective. As noted briefly in our conclusion section, several recent curriculum-style strategies for DMs appear closely aligned with the principle behind our Cor. 1, even though the current formal developments in the paper are specialized to autoregressive transformer reasoning.
>
> For MDLMs, Chen et al. (2502.06768), Kim et al. (2509.01025) and Bai et al. (2602.01842) suggest that **test-time decoding guided by the model’s highest-confidence positions** can improve reasoning performance. If confidence is viewed as a proxy for easiness (difficulty), then decoding guided by confidence are naturally interpretable as an easy-to-hard *inference curriculum*: the model first resolves easier subproblems, and then compositionally tackles harder ones. Relatedly, Kim et al. (2602.10314) provides a training-time analogue of this idea: its progressive unmasking strategy constructs training states according to model's top-k confidence. Viewing confidence as a proxy for difficulty, this can likewise be viewed as a curriculum, in which learning is concentrated on progressively more challenging intermediate states rather than being spread uniformly over an exponentially large space of states.
>
> From this viewpoint, both adaptive decoding and progressive training can be seen as high-level supporting instances of Cor. 1: structuring a sequence of easier-to-harder intermediate subproblems may reduce the effective complexity needed to achieve a target accuracy. We agree that extending the theory more explicitly to DLMs is an exciting direction, and we'll clarify this broader connection in the revision.
>
> Thank you and we welcome further exchange.

---

> > ### Author Rebuttal · Reviewer_gJFb · 2026-03-31
> >
> > The authors answer my concerns comprehensively. My questions around Blocksworld and DLMs were sufficiently answered and I feel this theoretically grounded paper has clear merits.

---

> > > ### Author Response · Authors · 2026-04-07
> > >
> > > Dear Reviewer gJFb,
> > >
> > > Thank you very much for raising your confidence to 5. Your recognition is truly encouraging, and we are especially grateful for your thoughtful engagement with the discussions on Blocksworld and DLMs. We also deeply appreciate your positive assessment of the paper as a theoretically grounded work with clear merits.
> > >
> > > We will incorporate these insightful discussions into the revision. Thank you again for your time and trust!
> > >
> > > Best regards,
> > > Authors of Submission 30760

---

### Official Review · Reviewer_6Ssi · 2026-03-12

**Soundness:** 2
**Presentation:** 1
**Significance:** 3
**Originality:** 2
**Overall Recommendation:** 4
**Confidence:** 3

**Summary:**

This paper studies the setting when curriculum post training can improve sample efficiency for sparse terminal rewards. They propose a framework where they introduce a two state conditioned autoregressive reasoning tree (2S-ART) for chain of thought. Their main observation is that native post training on hard tasks produce highly sparse reward signals. Then, a curriculum strategy basically provides denser signals sequentially. They prove that with curriculum-like strategy, they can go from exponential complexity under native post training to polynomial sample complexity under curriculum applied post training.

**Compliance With Llm Reviewing Policy:**

Affirmed.

**Final Justification:**

Author's rebuttal convinced me to increase my score to 4.

**Key Questions For Authors:**

1. The experiments are all on parity. How do you think that your results are applicable to tasks like countdown. Or more importantly, how do they connect to real llm post training?
2. Is there any possible impossibility result for the Corollary 1 when the conditions of that fails?

**Limitations:**

yes.

**Strengths And Weaknesses:**

### Strengths

1. At this time, an analysis post-training and specifically curriculum based post-training would be interesting to the overall ML community.
2. I think 2S-ART formulation is quite good as a starting point in realistically developing a theory.
3. I liked the reward hacking example as it highlights the difficulties in RL based post training with sparse terminal rewards.
4. Exponential vs polynomial sample complexity result is interesting.

### Weaknesses

1. The exposition in this paper is quite bad. The paper has many notational inconsistencies and typos that actually change the meaning. Overall, it makes the job of following along the paper almost impossible. For example, in Definition 2, the paper refers again to Definition 2. But I guess that should be definition 1.  Or $K$ is used in many places for different things.
2. Apart from the stylistic issues, paper is also quite difficult to follow. I spent considerable amount of time to be able to verify theorems in the paper but with the issues of notation/definition inconsistencies, it is quite difficult to verify the main claims in the paper.
3. The experiments are all on parity. Why is there no countdown experiments while it is used as a motivating example in many places in the paper?
4. Overall the paper would benefit from better presentation. It is in its current form one of the bottlenecks of the paper.
5. Corollary 1 conditions are too strong and under that regime, the result is not that interesting.

---

> ### Author Rebuttal · Authors · 2026-03-31
>
> Thank you for recognizing our theory, model, and results as interesting to the community.
>
> > **Room for Presentation**
>
> **A1:** We highly value the reviewer's careful feedback. We agree that the presentation have room to improve.
> Concretely, we'd correct the self-reference in Def.2 to Def.1, rename the input vocabulary size in Def.1 to $V$, reserve $K$ exclusively for pass@K evaluation, and use $L$ for the maximum reasoning depth / number of curriculum stages.
>
> We also sincerely appreciate the reviewer’s effort in checking. To improve readability, we **already** included proof sketches for both Thm.1 (lines 330-342, right column) and Thm.2 (lines 400-414, left column). In the revision, we'd further clean up the notation throughout and add notation tables (see Tab.5-6 in the anonymous link in **A3**) in the appendix to make the paper easier to follow and verify.
>
> At the same time, we note that other reviewers (e.g., Reviewer gJFb) found the theorems and proofs already clear and coherent. This suggests that the underlying logical structure, together with the proof sketches, is **already reasonably clear**, but can be made easier to follow through cleaner notation and exposition in the appendix. Thank you and the presentation would be revised accordingly.
> > **Role of Cor.1**:
> > 1. Cor.1 conditions are too strong.
> > 2. Is there any possible impossibility result for the Cor.1 when the conditions of that fails?
>
> **A2:** Thank you for this question. The purpose of Cor.1 is to identify a **natural sufficient condition** under which curriculum post-training can provably help exponentially, namely, the existence of a $L/p_{\max}$-root-complexity curriculum $(p_{\max}\ll L)$. We then show that this condition arises *naturally* in our autoregressive reasoning tree model (2S-ART, Def.1) for both Depth-Increasing and Hint-Decreasing curricula, including the **theoretically interesting** parity setting.
>
> When this condition fails, curriculum need not help. In particular, if each stage remains too hard, e.g.,
> $N_{ε}(\pi_ℓ^⋆|\pi_{ℓ-1}^⋆)
> = \Omega(N_{ε}(\pi^⋆ |\pi_{ref})/L),\forall ℓ \in [L]$
> then using $L$ stages can even be more expensive—than learning the target policy directly from $\pi_{ref}$. So while Cor.1 is a positive result stated under a sufficient condition, its failure naturally points to a limitation: curriculum is beneficial only when the intermediate stages are genuinely easier than the original task.
>
> > **Broader Experiments**:
> > 1. The experiments are all on parity. Why is there no countdown experiments while it is used as a motivating example in many places in the paper?
> > 2. How do you think that your results are applicable to tasks like countdown. Or more importantly, how do they connect to real llm post training?
>
> **A3:** Thank you for this constructive feedback. As noted in the paper, our abstract 2S-ART (Def.1) is motivated by the autoregressive tree reasoning process of LLMs and covers a broader class of theoretical reasoning problems, including Markov-chain reasoning, induction-head-style associative recall, and causal-graph reasoning (see the discussion around lines 217–219 on the left and 183–187 on the right). Among these, the parity task is the most **theoretically-interesting** one, since its minimax-rate is well-known to be $d^{k^*}$, suffering from an exponential bottleneck. This makes it especially suitable for demonstrating our theorems to show the benefit of curriculum in both fine-tuning and test-time scaling.
>
> For real-world LLM post-training, however, a rigorous sample-complexity analysis is much more challenging. Deep neural networks introduce substantial nonlinearity, and existing theory remains limited, mostly to relatively shallow or highly structured models/algorithms (see lines 360-382 on the left). Notably, there exists some non-curriculum SGD that replicates Gaussian elimination and achieve polynomial complexity (see lines 355-358 on the left). Taken together, these gaps make a fully realistic theory of complexity separation for LLM post-training intractable at present.
>
> For this reason, we view our work as a first theoretical step: it abstracts an **empirically-motivated** transformer tree-reasoning setting in which curriculum benefit could be prvably shown in theoretically-attractive problems. Our goal is to highlight the theoretical potential of curriculum post-training, whose empirical benefits have **already been widely observed** in real LLMs (Parashar et al., 2025; Liu et al., 2025a). That is to say, we also appreciate the reviewer’s suggestion to include broader empirical evidence. To this end, we have conducted experiments on **Countdown**, **MATH** and **Blockworlds** with **Qwen2.5-1.5B** to further support our practical relevance (see Tab.1-4 and Fig.12-14 in https://anonymous.4open.science/r/submission-materials-0B17/supplementary_figures_tabs.pdf); these results and discussions will be included in the revision.
>
> Thank you and we welcome further exchange.

---

> > ### Author Rebuttal · Reviewer_6Ssi · 2026-04-03
> >
> > 1. For the notation and clarity, I suggested concrete examples, where the clarity is nonexistent. That another reviewer found it clear does not address my concern. The role of the reviewer is to check the soundness/significance of the work. When the exposure is limited, it is not possible.
> >
> > 2.  What does **natural** sufficient condition mean? The issue is that, in the later parts of the paper, the authors also present conditions that essentially give rise to Corollary 1 such as "when the base policy assigns comparable probability mass to the children at each parent state". Why should we think that such a condition in 2S-ART setting is flexible and not limiting? Furthermore, namings such as **natural** or canonical is highly ambitious and there should be a very good justification for the use of it.
> >
> > Also, I thank for the experiments. If authors address my concerns, I will increase my score.

---

> > > ### Author Response · Authors · 2026-04-04
> > >
> > > Thank you for the follow-up. We appreciate the reviewer's acknowledgement of our additional experiments.
> > >
> > > > For the notation and clarity, I suggested concrete examples, where the clarity is nonexistent. That another reviewer found it clear does not address my concern. The role of the reviewer is to check the soundness/significance of the work. When the exposition is limited, it is not possible.
> > >
> > > We fully understand this concern, and we agree that improving notation and exposition is very important. Our point in mentioning another reviewer was not to dismiss your concern, but only to indicate that the core logical structure, together with the proof sketches, is present; we agree that your concrete suggestions are helpful and valuable.
> > >
> > > Concretely, we'll implement the notation fixes listed above in our **A1**, including correcting the Definition 2 self-reference, removing symbol collisions, and standardizing the selector/update notation throughout. As mentioned, we have also prepared notation tables (Tabs. 5--6 in https://anonymous.4open.science/r/submission-materials-0B17/supplementary_figures_tabs.pdf) to make the symbol roles and dependencies explicit. In addition, the main text already contains proof sketches for both Theorem 2 (lines 330--342, right column) and Theorem 3 (lines 400--414, left column), which, together with the notation fixes and tables, should make the paper substantially easier to follow.
> > >
> > > At a high level, both theorems analyze how many samples are needed to reliably distinguish the relevant probability gap via Bernstein-type arguments in the presence of reward hacking. For example, in parity, choosing a wrong secret index can still produce the correct output token with probability $0.5$ (also see our **A1** to Reviewer CgMU). While the gradient/signal ranges are bounded by the same order, curriculum strategies provide stage-wise information and thereby enlarge the correct-vs.-incorrect selection margin to order $\Theta(d^{-1})$, whereas the no-curriculum counterpart only has a much smaller margin of order $\Theta(d^{-(k^\star+c)})$, where $c$ is a constant. Distinguishing these gaps information-theoretically requires $\Theta(d^2)$ and $\Theta(d^{2(k^\star+c)})$ samples, respectively.
> > >
> > > The notation density in the appendix comes from three sources: (i) the nonlinearity of the gradient, which requires tracking different components separately; (ii) our goal of covering a general setting beyond parity, with parity as a subcase, where reward-hacking effects are captured by spurious-acceptance parameters; and (iii) the need to classify probability events carefully and locally for the different regimes of RL fine-tuning and test-time scaling.
> > >
> > > Together with the existing proof sketches, the revised notation, and the notation tables, we believe the paper has become satisfactorily reader-friendly. Thank you again for these helpful suggestions.
> > >
> > > > What does natural sufficient condition mean? The issue is that, in the later parts of the paper, the authors also present conditions that essentially give rise to Corollary 1 such as "when the base policy assigns comparable probability mass to the children at each parent state". Why should we think that such a condition in 2S-ART setting is flexible and not limiting? Furthermore, namings such as natural or canonical is highly ambitious and there should be a very good justification for the use of it.
> > >
> > >
> > > Thank you for this valuable question.
> > >
> > > Our theory is motivated by reasoning datasets such as parity, Countdown and Blockworlds, where reasoning depth naturally correlates with instance difficulty. In Blockworlds, for example, more operations in planning is inherently more challenging. The assumption of comparable probability mass over legal children means that the pretrained base model has comparable capability across tasks *at the same difficulty level*, which is plausible when pretraining is **not particularly skewed** toward some specific subset of tasks.
> > >
> > > In parity, this means that, conditioned on any prefix, the base model has roughly comparable prior knowledge over all legal next positions, **rather than systematically neglecting or favoring** any particular secret index. Likewise, for Countdown, it is reasonable to assume that a pretrained Qwen model does not systematically favor a particular number or arithmetic operation among equally legal choices. This is the sense in which we referred to the condition as "natural": it corresponds to a non-skewed pretraining regime.
> > >
> > > That said, we agree with the reviewer that terms such as "natural" can sound too strong without sufficient qualification. In the revision, we will therefore revise the wording with clarifications above, to make clear that this is one interpretable and plausible regime.
> > >
> > > Thank you once again for your valuable time, thoughtful feedback, and your kind willingness to reconsider the score based on our response.

---

### Official Review · Reviewer_CgMU · 2026-03-13

**Soundness:** 3
**Presentation:** 3
**Significance:** 2
**Originality:** 3
**Overall Recommendation:** 4
**Confidence:** 3

**Summary:**

This paper investigates why the curriculum techniques in the post-training stage of LLMs could be sample efficient, especially when the rewards for post-training are outcome-only. Their analysis shows that curriculum strategies could help reinforcement learning achieves high accuracy with polynomial sample complexity. On the other hand, non-curriculum is exponential complexity. They also do some analysis on test-time scaling and indicates that curriculum strategies reduced the number of required queries. They validate their theory with an experiment on a parity task.

**Compliance With Llm Reviewing Policy:**

Affirmed.

**Final Justification:**

The author resolved my score with more experiments. I will raise my score

**Key Questions For Authors:**

- I think the main question would be parity task might be a toy task
- I also expect explanations on the difference between the theory and practical training.

**Limitations:**

They mentioned some limitations in the conclusion

**Strengths And Weaknesses:**

Strengths:
- Explain well the benefits of curriculum learning
- Not only talk about the curriculum strategies in RL but also test-time scaling

Weakness:
- The theory could explain the exponential bottleneck. But in practical rl training, the mistakes could stem from multiple places(e.g skip steps, hallucinate, etc)
- Parity task is a kind of toy task, experiments on some popular reasoning benchmarks, like ARC, MMLU, GSM8k would be more helpful.
- Might need some citations regard recent works about curriculum learning in the post training. E.g. bread, core, etc

---

> ### Author Rebuttal · Authors · 2026-03-31
>
> Thank you for acknowledging our theoretical contribution on both curriculum finetuning and test-time scaling.
>
> > **W1**: The theory could explain the exponential bottleneck.But in practical rl training,the mistakes could stem from multiple places(e.g skip steps, hallucinate, etc)
>
> **A1:** Thank you for the comment. Our theory does not **pre-assume a specific source of mistakes**. In particular, it already allows **spurious success** (see line 268-274 on the right), where an incorrect reasoning process still potentially produces the correct final answer and receives reward. This covers both examples you mentioned:
>
> 1. **Skipped steps.** For example, in the parity case, if the correct answer depends on two secret indices $x_1,x_2$ but the reasoning uses only $x_1$, then the model effectively skips a necessary step $x_1⊕x_2$ and outputs the parity of $x_1$ alone.
>
> 2. **Hallucination.** In the same example, although reasoning with only $x_1$ is incorrect, its parity can still coincide with $x_1⊕x_2$ by 0.5 chance. This is exactly the kind of spurious success captured by our 2S-ART model.
>
> That is to say, our autoregressive tree model per Def.1 is abstract, but it **already** covers both types of errors in a *high-level* sense, as well as a broader class of theoretically meaningful problems beyond parity (see the left of line 217-219; the right of line 183-187). We agree that some practical error sources are outside the current scope, such as a noisy reward oracle that potentially rejects a correct answer. Still, we believe the current model is already general enough in theory for our main theoretical question.
>
> > **W2 & Q1**: Parity task is a kind of toy task, experiments on some popular reasoning benchmarks...would be more helpful; I think the main question would be parity task might be a toy task.
>
> **A2:** Thank you for the feedback. To clarify, our theory is not limited to parity. Our autoregressive tree reasoning model (2S-ART in Def.1) also covers broader theoretical reasoning settings, including Markov-chain reasoning, induction-head associative recall, and causal-graph reasoning (see the left of line 217-219; the right of line 183-187). Among these, the parity task is the most **theoretically-interesting** one, since its minimax-rate is well-known to be $d^{k^*}$, suffering from an exponential bottleneck. This makes it especially suitable for demonstrating our theoretical results to show the theoretical benefit of curriculum post-training, whose empirical success have **already been widely observed** in LLMs (see lines 10-20 on the right).
>
> That is to say, we appreciate the chance to provide experiments on celebrated benchmarks. We conducted experiments on **Countdown**, **MATH** and **Blockworlds** with **Qwen2.5-1.5B**, see Tab.1-4 and Fig.12-14 in https://anonymous.4open.science/r/submission-materials-0B17/supplementary_figures_tabs.pdf; these'll be included in the revision with discussions.
> >**W3**: Might need some citations regard recent works about curriculum learning in the post training. E.g. bread, core, etc
>
> **A3:** Thank you and we'll add these citations in the revision. **BREAD** uses branched rollouts with partial expert prefixes: when the model fails, it is given a short expert hint and trained to complete the remaining reasoning steps. As training progresses, the rollouts become increasingly self-generated. This is a clear instance of the **Hint-Decreasing Curriculum** in our paper, though implemented in a more practically refined way. **CORE** is different in form but closely aligned with our perspective. It introduces concept-guided intermediate supervision, so the model first learns important concepts and then reasons from them. In our view, this can be interpreted as constructing an easier intermediate policy $\pi^\star_{concept}$ before learning the final target policy. From the perspective of Cor.1, it follows the same general principle: a easier intermediate subtask can reduce the overall learning complexity.
> >**Q2**: Explanations on the difference between the theory and practical training.
>
> **A4:** Thank you for the feedback. At a high level, both theoretical and practical curriculum-based post-training share the same goal: to improve learning by introducing intermediate subtasks.
> The main difference lies in **how these stages are implemented** for real tasks. In our theory, the stages are explicitly defined, and each stage comes with a theoretically characterized sample complexity. In practice, however, curriculum stages are usually implemented more smoothly and heuristically. For example, UFT and E2H use scheduled reductions in hints and increase in data difficulty, while BREAD adapts the hint lengths based on the model's current rollout behavior. We believe this is natural: smoother transitions adaptively match the model's continuously changing capability in real-training. We'd clarify this theory-practice distinction in the revision.
>
> Thank you and we welcome further exchange.

---

> > ### Author Rebuttal · Reviewer_CgMU · 2026-04-02
> >
> > The author fully resolved my concerns. I will raise my score.

---

> > > ### Author Response · Authors · 2026-04-07
> > >
> > > Dear Reviewer CgMU,
> > >
> > > Thank you very much for raising your score to 4. We are glad that our rebuttal successfully addressed all of your concerns. We especially appreciate your questions about error sources, your suggestions on additional citations, and your request for broader empirical evidence and clarification of practical relevance. The clarifications we prepared have helped us strengthen the final version of the paper.
> > >
> > > We deeply value your support and will ensure that the revision incorporates the proposed revisions. Thank you again for your time, trust, and constructive feedback.
> > >
> > > Best regards,
> > > Authors of Submission 30760

---

### Official Review · Reviewer_msRe · 2026-03-13

**Soundness:** 3
**Presentation:** 3
**Significance:** 2
**Originality:** 2
**Overall Recommendation:** 4
**Confidence:** 2

**Summary:**

This paper studies when curriculum-style post-training can provably improve the sample efficiency of reasoning in transformers. It uses the notion of coverage, where task difficulty is tied to the rarity of correct reasoning trajectories under the base policy, and shows that if curriculum stages are sufficiently easier than the final task, curriculum learning can turn exponential dependence into polynomial dependence. The paper then instantiates this with a synthetic autoregressive reasoning-tree model (2S-ART / PART) and proves polynomial-vs-exponential separations for RL fine-tuning and test-time scaling. The empirical section is limited to parity-style simulations, where curriculum variants do converge much faster than no-curriculum baselines.

**Compliance With Llm Reviewing Policy:**

Affirmed.

**Final Justification:**

The rebuttal addressed all of my concerns as indicated in my response.
However,  I do not feel confident advocating for acceptance too strongly, since this work is outside my area of expertise. My initial review leaned towards acceptance based on my understanding of the contributions of this work, and I will maintain my score.

**Key Questions For Authors:**

- How should one think about the curriculum oracle $R_{F_{S*}}(\cdot,\ell)$ in realistic tasks where only the final answer is verifiable?
- How sensitive are the conclusions to the one-step-per-stage setup and the very restricted trainable parameterization?

**Limitations:**

The paper does acknowledge that its 2S-ART/PART abstraction is restrictive, but they can be made more complete. In particular, the paper should more explicitly highlight that the theory depends on a specific reasoning-tree model, strong architectural and optimization assumptions, and curriculum oracles that provide richer intermediate feedback than ordinary final-answer-only supervision. It should also mention the limited scope of empirical validation.

**Strengths And Weaknesses:**

### Strengths

- The paper asks a clear theoretical question and provides a general bottleneck corollary for when curriculum can help.
- The framework is reasonably unified: it covers both RL fine-tuning and test-time scaling, and gives exponential-to-polynomial improvements in both settings under the paper’s assumptions.
- The synthetic experiments match the claimed qualitative trend and support the theory in the parity setting.

### Weaknesses

- The work relies on several strong assumptions: the base model assigns uniform mass to legal children, reasoning primitives already exist in the FFN and stay stable under fine-tuning, and only a restricted matrix is trainable. While some assumptions are needed to enable theoretical analysis, but some of these seem too detached from standard LLM post-training.
- The “outcome-only” framing seems a bit misleading for the curriculum results. The curriculum oracle $R_{F_{S*}}(\cdot,\ell)$ gives stage-specific feedback about the subtask at depth $\ell$. It should be stated more clearly that the curriculum result relies on richer supervision than just a single final verifier.
- The empirical evidence is very narrow relative to the title and claims, since the paper only validates the theory on synthetic parity tasks with small dimensions, while examples like Countdown or more realistic reasoning benchmarks are discussed but not actually evaluated here. That is enough to show consistency with the theory, but not enough to support a broad practical claim about transformer reasoning more generally.

---

> ### Author Rebuttal · Authors · 2026-03-31
>
> Thank you for recognizing our theoretical results as explanatory for both RL fine-tuning and test-time scaling within a unified framework. We are also glad that the reviewer found the experiments supportive of the theory.
>
> > **W1 & Q2**:The work relies on several strong assumptions: the base model assigns uniform mass to legal children, reasoning primitives already exist in the FFN and stay stable under fine-tuning, and only a restricted matrix is trainable. While some assumptions are needed to enable theoretical analysis, but some of these seem too detached from standard LLM post-training. How sensitive are the conclusions to the one-step-per-stage setup and the very restricted trainable parameterization?
>
> **A1:** Our theory is motivated by reasoning datasets in LLM post-training such as Countdown and Blocksworld, where reasoning length naturally correlates with instance difficulty. The assumption of equal probability over legal children means that the pretrained base model has comparable capability across tasks of the same difficulty level, which is natural when **pretraining is not skewed** toward some particular subset of tasks. For example, it is natural to assume that a pretrained Qwen model does not systematically favor any particular number or arithmetic operation.
>
> It is true that, in LLMs, representations are jointly formed by attention and FFN layers. However, for a sufficiently pretrained model, it is natural to assume that the relevant atomic skills already exist before RL fine-tuning—for example, grammar, the arithmetic operations in Countdown, or the XOR operation in parity. Otherwise, one would typically rely on SFT before RL fine-tuning. In our 2S-ART parameterization, the FFN plays the role of implementing these atomic skills, so keeping it fixed is a natural abstraction for post-training. For instance, in the parity case, the pretrained FFN is fixed as in Eq.(8) (lines 318-320). If one also fine-tunes the FFN, then $W_1$ and $W_2$ may become biased toward certain XOR operations while weakening their ability to represent others.
>
> In practice, curriculum algorithms usually adopt smoother transitions between stages, such as cosine scheduling (e.g., UFT in 2505.16984 or E2H in 2506.06632), or adapt to the model’s current status (e.g., BREAD in 2506.17211). But at a high level they are still stage-wise curricula, so a step-wise analysis is a natural conceptual abstraction for theoretical purpose (see lines 360-363 on the left).
> > **W2 & Q1**:The “outcome-only” framing seems a bit misleading for the curriculum results. The curriculum oracle gives stage-specific feedback about the subtask at depth.It should be stated more clearly that the curriculum result relies on richer supervision than just a single final verifier.How should one think about the curriculum oracle in realistic tasks where only the final answer is verifiable?
>
>
> **A2:** Thank you for this helpful comment. The reward oracle $\mathbf{R}^{\mathcal{F}\_{S\_{⋆}}}\_{\mathbf{x}}(⋅,ℓ)$ explicitly depends on the stage index $ℓ$ because the subtask family $\mathcal{F}\_{S\_{⋆}}=\{f\_{S\_{⋆}^1},\ldots,f_{S\_{⋆}^{k^{⋆}}}\}$ contains subtasks of different lengths. If $ℓ$ were not specified, and reward $1$ were given whenever the outcome matched *any* subtask in $\mathcal{F}\_{S\_{⋆}}$, then the oracle would become excessively noisy. Therefore, for the current target stage $ℓ$, we reward only whether the pre-EOS outcome token matches the correct final token of the current subtask $f\_{S\_{⋆}^{ℓ}}(\mathbf{x})$, and do not reward outcomes corresponding to easier or harder subtasks.
>
> Importantly, this is still **outcome-only supervision for the current subtask**: the oracle checks only whether the final outcome of the stage-$ℓ$ subproblem is correct. The purpose of including $ℓ$ is simply to avoid severe reward noise across subtasks of different difficulty. For example, in parity or Countdown, when training a depth-$2$ subtask, the model should not be rewarded for just completing the previous depth-$1$ subtask instead. For broader tasks' curricula, this corresponds to the control of the current stage's data to be the level of current stage. In the revision, we'll state this more explicitly: the curriculum oracle is outcome-only at the subtask level, but it does depends the stage index $ℓ$ without rewarding too many subtasks simultaneously.
> > **W3**: Broader Empirical Relevance such as Countdown.
>
> **A3:** Thank you for recognising that our current empirical results already consistent with our theory enough. We appreciate the chance to include broader empirical evidence. To this end, we have conducted experiments on **Countdown**, **MATH** and **Blockworlds** with **Qwen2.5-1.5B** to further support our practical relevance (see Tab.1-4 \& Fig.12-14 in https://anonymous.4open.science/r/submission-materials-0B17/supplementary_figures_tabs.pdf); these results and discussions'll be included in the revision.
>
> Thank you and we welcome further exchange.

---

> > ### Author Rebuttal · Reviewer_msRe · 2026-04-02
> >
> > I thank the authors for their response and for clarifying my questions.
> > I will admit this paper is outside my core expertise (hence the low confidence). I will maintain my score and am inclined to accept this paper based on my understanding of its contributions.

---

> > > ### Author Response · Authors · 2026-04-07
> > >
> > > Dear Reviewer msRe,
> > >
> > > Thank you very much for your encouraging feedback and thoughtful engagement throughout the review process. We especially appreciate your questions on practical relevance, including the plausibility of uniform mass over legal children, the reasoning primitives, and the trainable parameter regime. These comments helped us further clarify the motivation and interpretation of our assumptions.
> > >
> > > We also appreciate your suggestions on broader empirical relevance. The corresponding clarifications and additional experiments will be incorporated into the final version and would help strengthen the paper.
> > >
> > > Thank you again for your time, trust, and constructive feedback.
> > >
> > > Best regards,
> > > Authors of Submission 30760

---

### Decision · Program_Chairs · 2026-04-30

**Decision:**

Accept (regular)

**Comment:**

This paper introduces a theoretical framework for curriculum post-training in LLM reasoning, proving exponential-to-polynomial sample complexity reductions for depth-increasing and hint-decreasing strategies in both RL and test-time scaling settings.

After sufficient discussion, I summarize the primary merits of this work as :(i) Provides a unified theory for both RL post-training and test-time scaling; (ii) offers rigorous proof for the efficiency of curriculum strategies; (iii) validates qualitative trends through simulations and empirical results on benchmarks like Countdown, MATH, and Blocksworld.

Some limitations can be further added in the final version: (i) Simplified theoretical assumptions regarding model dynamics and trainable parameters [**Reviewer msRe, CgMU**]; (ii) reliance on a stage-indexed oracle for feedback rather than simple outcome verification [**Reviewer msRe**].

**Overall Consideration:** This technically solid paper provides valuable theoretical insights into why curriculum-based post-training is effective for complex reasoning tasks. Hence, I recommed an acceptance